# Photostimulation of brain lymphatics in male newborn and adult rodents for therapy of intraventricular hemorrhage

Dongyu Li[1,2,11], Shaojun Liu[1,11], Tingting Yu [1] ✉, Zhang Liu[1], Silin Sun[1], Denis Bragin[3,4], Alexander Shirokov [5,6], Nikita Navolokin [6,7], Olga Bragina[3], Zhengwu Hu[1,2], Jürgen Kurths [6,8,9,10], Ivan Fedosov[6], Inna Blokhina[6], Alexander Dubrovski [6], Alexander Khorovodov[6], Andrey Terskov[6], Maria Tzoy[6], Oxana Semyachkina-Glushkovskaya [6,8] ✉ & Dan Zhu [1] ✉

Intraventricular hemorrhage is one of the most fatal forms of brain injury that is a common complication of premature infants. However, the therapy of this type of hemorrhage is limited, and new strategies are needed to reduce hematoma expansion. Here we show that the meningeal lymphatics is a pathway to remove red blood cells from the brain's ventricular system of male human, adult and newborn rodents and is a target for non-invasive transcranial near infrared photobiomodulation. Our results uncover the clinical significance of phototherapy of intraventricular hemorrhage in 4-day old male rat pups that have the brain similar to a preterm human brain. The course of phototherapy in newborn rats provides fast recovery after intraventricular hemorrhage due to photo-improvements of lymphatic drainage and clearing functions. These findings shed light on the mechanisms of phototherapy of intraventricular hemorrhage that can be a clinically relevant technology for treatment of neonatal intracerebral bleedings.

Intraventricular hemorrhage (IVH) is a bleeding into the brain's ventricular system, where the cerebrospinal fluid (CSF) is produced. The IVH is one of the most common types of brain injury in preterm infants born before the 30th gestational week[1–3]. The IVH in preterm infants occurs when a germinal matrix hemorrhage ruptures through the ependyma into the lateral ventricle[4]. The incidence of IVH in such infants is approximately 25–30%[5,6]. Most IVH occurs within the first 72 h after birth and progresses rapidly within 1 week[7]. Despite survival rates increasing to approximately 70%, about 45–85% of premature infants with moderate-to-severe IVH develop significant cognitive deficits, and approximately 75% of such babies need special education in school[8,9]. In adults, about 30% of IVHs primarily result from trauma, and 70% are secondary, i.e., originate from spontaneous intracranial hemorrhage (ICH)[10–12]. The common associations include hypertension, arteriovenous malformations, aneurysms, moyamoya disease, coagulopathy, and arteriovenous fistula[13]. It is established that IVH

[1]Britton Chance Center for Biomedical Photonics - MoE Key Laboratory for Biomedical Photonics, Wuhan National Laboratory for Optoelectronics - Advanced Biomedical Imaging Facility, Huazhong University of Science and Technology, 430074 Wuhan, Hubei, China. [2]School of Optical Electronic Information, Huazhong University of Science and Technology, 430074 Wuhan, Hubei, China. [3]Lovelace Biomedical Research Institute, Albuquerque, NM 87108, USA. [4]Department of Neurology University of New Mexico School of Medicine, Albuquerque, NM 87131, USA. [5]Institute of Biochemistry and Physiology of Plants and Microorganisms, Russian Academy of Sciences, Prospekt Entuziastov 13, Saratov 410049, Russia. [6]Saratov State University, Astrakhanskaya str., 83, Saratov 410012, Russia. [7]Saratov State Medical University, B. Kazachya str., 112, Saratov 410012, Russia. [8]Physics Department, Humboldt University, Newtonstrasse 15, 12489 Berlin, Germany. [9]Potsdam Institute for Climate Impact Research, Telegrafenberg A31, 14473 Potsdam, Germany. [10]Sechenov First Moscow State Medical University, Bolshaya Pirogovskaya 2, building 4, 119435 Moscow, Russia. [11]These authors contributed equally: Dongyu Li, Shaojun Liu. ✉ e-mail: yutingting@hust.edu.cn; glushkovskaya@mail.ru; dawnzh@mail.hust.edu.cn

extension is independently associated with high mortality and poor functional outcome in adult patients[13].

Blood in the ventricular system contributes to morbidity in a variety of ways. Pressure from the leaked blood damages brain cells, disabling the proper function of the injured area. Besides, red blood cell (RBC) lysis after IVH results in a release of blood breakdown products (hemoglobin, iron, and bilirubin). Such products have been implicated in post-hemorrhagic hydrocephalus and increased intracerebral pressure (ICP) due to impaired CSF circulation and brain drainage system[14,15]. Therefore, the conventional therapy for IVH, including surgery and fibrinolysis in combination with extraventricular drainage, is aimed at the blood evacuation from the ventricles, both to reduce ICP induced by mass effects of blood clots on the ventricular walls and the secondary damage caused by blood cell lysis[16,17]. However, the existing therapy directed at ameliorating intraventricular clot has been limited, and new strategies are needed to reduce hematoma expansion and improve the drainage system of the brain[16,17].

Photostimulation (PS) can be an innovative technology targeted for the therapy of IVH. Transcranial PS is a non-pharmacological and non-invasive therapy for stroke[18–20] and traumatic brain injuries[20–23]. It is believed that PS increases the metabolic activity of brain tissues and microcirculation, which increases recovery resources[20,23]. There is strong evidence that PS can regulate the relaxation and permeability of the lymphatic vessels (LVs), activate the movement of immune cells in the lymph, and effectively manage lymphedema[24–26]. Our preliminary work demonstrated that near-infrared PS (1267 nm) stimulates the clearance of tracers from the brain via modulation of lymphatic tone and contraction[24,25,27]. We also showed that PS (1267 nm) effectively stimulates lymphatic clearance of beta-amyloid from the brain[28].

In this pilot study performed on adult mice and newborn rats, we found that meningeal lymphatic vessels (MLVs) drain RBCs from the right lateral ventricle into the deep cervical lymph nodes (dcLNs). Using the meninges of humans which died due to IVH, we found RBCs in MLVs. Our animal and human results provide strong support for the lymphatic pathway of RBCs evacuation from the brain. We have in particularly uncovered that PS has therapeutic effects on IVH in both adult and newborn rodents, improving neurological outcomes, accelerating the RBCs evacuation from the ventricles, providing faster recovery of permeability of the blood-brain barrier (BBB) and the brain's drainage. These findings shed light on our fundamental knowledge about the effects of low-level infrared PS on mature and neonatal brain recovery after IVH. The PS-acceleration of RBCs clearance from the brain via MLVs can be clinically significant for the therapy of brain hemorrhages in newborns, where PS can be applied through the fontanelles and where MLVs are localized along the sagittal and transversal sinuses. If our preclinical results will be confirmed in further clinical trials, non-invasive PS-mediated stimulation of lymphatic clearance of RBCs can become a readily applicable, and commercially viable technology for the effective routine treatment of IVH and other types of brain bleeding.

## Results

### Lymphatic clearance of RBCs from the adult mouse and human brain

It has been shown that RBCs can be cleared from the brain[29] and the subdural and subarachnoid space[30,31] into dcLNs. We, therefore, examined the lymphatic pathway of RBCs clearance from the adult mouse brain into dcLNs after IVH. The confocal colocalization analysis of dcLNs stained for two classical markers of the lymphatic (LYVE-1, Lymphatic vessel endothelial hyaluronan receptor 1 and PROX-1, Prospero homeobox protein 1) and the blood endothelium cells (CD-31) revealed the presence of RBCs in dcLNs of mice with IVH (Fig. 1b). The RBCs were observed inside LVs of dcLNs 1 h after IVH. There were no RBCs in LVs of dcLNs in intact mice (Fig. 1a).

Theoretically, following IVH, RBCs can be evacuated to the subarachnoid space via CSF movement and penetrate into MLVs, which drain them to dcLNs (Supplementary Fig. 1). Therefore, we analyzed the RBCs presence in MLVs stained with LYVE-1/PROX-1; the blood vessels were labeled with CD-31. The confocal colocalization analysis demonstrated that RBCs were observed in MLVs of mice with IVH but not in intact mice (Fig. 1c, d). In addition, we analyzed the human meninges stained with LYVE−1 (LEC marker), CD-31 (the blood endothelium marker) and anti-glycophorin A (GPA, marker for RBCs). The human data indicated that RBCs were also presented in MLVs after IVH (Fig. 1e).

These results suggest the lymphatic pathway of RBCs clearance from the adult mouse brain, which is consistent with the oldest[32,33] and latest[29–31] animal data on the lymphatic efflux of RBCs from the brain and the meninges. Our human results on the meninges are supported by the data presented by Caversaccio et al.[34], who found an intense accumulation of iron, as an essential element of RBCs, in dcLNs in patients died due to an intracranial hemorrhage that indicates on a lymphatic pathway of clearance of blood products from the brain to the peripheral lymphatics.

### PS stimulates lymphatic clearance of RBCs and macromolecules from the brain of adult mice

It is well known that irradiation wavelength range starting from 1100 nm and longer begin to be absorbed by water[35] and can cause significant heating effects on the water content of biological objects. As demonstrated[36,37], brain functions are so sensitive to temperature changes that an increase of 0.5 °C can cause significant alterations in the cellular processes. 1 °C and above can induce profound effects on neural network functioning. Therefore, we measured the temperature on the skull (the scalp removed) and on the top of the cortex for five single laser doses of 3–6–9–18–27 J/cm² as well as we analyzed morphological changes in the brain tissues after the 7 days PS-course with different laser doses in healthy mice (see a design of experiment in Supplementary Fig. 10c).

The temperature on the external skull surface was increased by 0.11–0.14–0.20–0.70–1.83 °C after PS 3–6–9–18–27 J/cm² ($n = 10$ in each group, Supplementary Table 1). The application of these small PS doses was not accompanied by any changes in the cortex surface temperature (Supplementary Table 1). Thus, the laser doses of 3–6–9–18–27 J/cm² did not significantly increase the brain tissue's temperature. The histological analysis did not reveal any changes in the brain tissues after the PS course for 7 days, suggesting no effects of PS on the brain morphology (Supplementary Fig. 2).

To find the effective PS dose for stimulation of lymphatic functions, we studied the clearance of gold nanorods (GNRs) from the right lateral ventricle for 60 min in healthy mice (Supplementary Table 2). Using in vivo optical coherence tomography (OCT), we found that the rate of GNRs accumulation in dcLNs was minimal in the control group (no PS) and was not changed after PS 3 J/cm² and 6 J/cm² (Supplementary Table 2). However, the rate of GNRs accumulation in dcLN was higher after PS 9 J/cm² (Fig. 2c). There was no further increase in the intensity of GNRs accumulation in dcLNs using higher PS doses 18 J/cm² and 27 J/cm² (Supplementary Table 2).

To confirm these findings, we performed a quantitative analysis of the GNRs level in dcLNs using atomic absorption spectroscopy (Supplementary Table 3). The ex vivo results confirmed the OCT data and showed that the level of GNRs in dcLNs was not changed after PS 3 J/cm² and 6 J/cm² and was significantly increased after PS 9 J/cm². The use of higher PS doses of 18 J/cm² and 27 J/cm² was not accompanied by a more significant increase in the level of GNRs in dcLNs than the PS dose of 9 J/cm² (Supplementary Table 3).

Additionally, we used fluorescent microscopy for in vivo imaging of Evans Blue dye (EBD) clearance from the right ventricle before and after PS with the laser doses of 3–6–9–18–27 J/cm² in healthy mice.

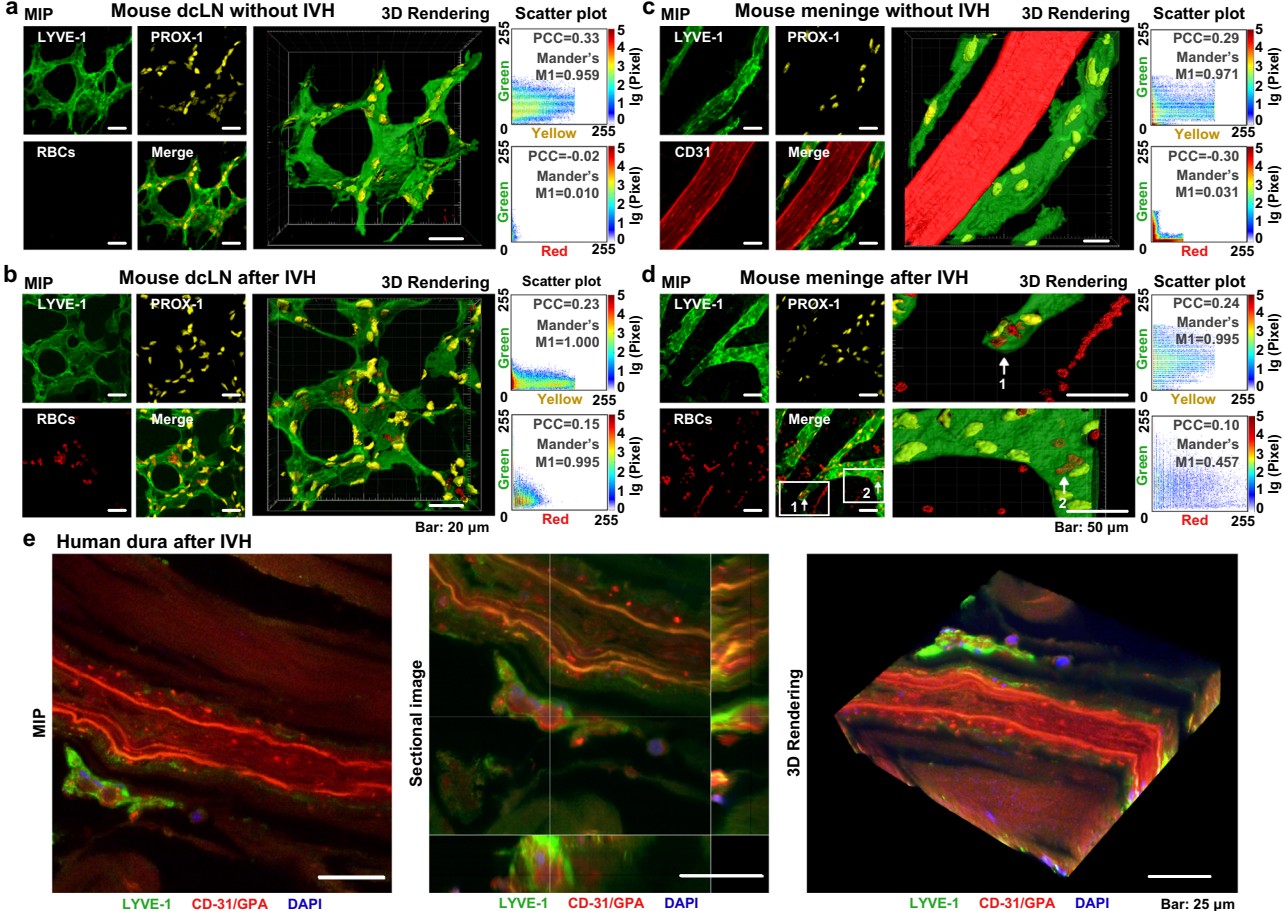

**Fig. 1 | Lymphatic clearance of RBCs from mouse brain into dcLNs.**
**a**, **b** Representative confocal images (from >3 replicates) of LVs of dcLN stained for LYVE-1 (green color), PROX-1 (yellow color) and RBCs (red color, autofluorescence) without (**a**) and after IVH (**b**). Scale bar: 20 μm. 3D rendering images illustrate LVs of dcLN in intact mice on the top in (**a**) and LVs filled with RBCs in mice with IVH on the bottom in (**b**). MIP: Maximum intensity projection. PCC: Pearson's correlation coefficient, which is between 1 and −1. 1 represents perfect correlation, −1 represents an entirely negative correlation, and 0 represents a random relationship. Manders' M1: The proportion of the red/yellow fluorescence regions co-located with green fluorescence. Scatter plots indicate that the distribution of PROX-1 is positively correlated with LYVE-1, and almost all PROX-1 coincide with LYVE-1. In addition, the red signals, which represent RBCs are negatively correlated with the distribution of LYVE-1 without IVH, suggesting there were no RBCs inside LVs of dcLNs in intact mice. **c**, **d** Representative confocal images (from >3 replicates) of MLVs stained for LYVE-1 (green color) and PROX-1 (yellow color), the blood vessels stained with CD31 (red color) and RBCs (red color, autofluorescence) in intact mice (**c**) and in mice with IVH (**d**). Scale bar: 50 μm. 3D rendering images illustrate MLVs in intact mice on the top in (**c**) and MLVs filled with RBCs (red color, autofluorescence) in mice with IVH on the bottom in (**d**). In (**d**) 3D rendering images are larger view of the frame areas in (1) and (2), which clearly show that RBCs (red color) around MLVs (1) or inside MLVs (2). **e** Representative confocal images (from 3 replicates) of LVs in the human meninges stained for LYVE-1 (green color), the cerebral vessels stained for CD-31 (red color), RBCs labeled with GPA (red color), DAPI (blue color). Scale bar: 25 μm.

The effective time of dye elimination was initially established in mice without PS using white-light imaging. Supplementary Fig. 3 illustrates that 60 min after EBD intraventricular injection, dcLNs become blue due to dye accumulation. Therefore, 60 min of observation was selected as the time for monitoring EBD accumulation in dcLNs. The fluorescence images demonstrate that the intensity of EBD fluorescence in dcLNs during 60 min was not changed after PS 3 J/cm² and 6 J/cm² and was increased after PS 9 J/cm² (Supplementary Table 4 and Fig. 2a, b). There was no further increase in the intensity of EBD accumulation in dcLNs using higher PS doses of 18 J/cm² and 27 J/cm² (Supplementary Table 4).

Thus, our results on healthy mice demonstrate that PS doses 3 J/cm² and 6 J/cm² were insufficient for PS-stimulation of lymphatic clearance of GRNs and EBD. In contrast, PS doses 9–18–27 J/cm² caused similar effects on GNRs and EBD accumulation in dcLNs. However, PS doses of 18 J/cm² and 27 J/cm² were accompanied by an increase in the skull temperature that can induce the heating effects on red blood cells[38]. Therefore, PS 9 J/cm² on the skull (3 J/cm² on the brain surface, Supplementary Fig. 4) has been selected as optimal for further

experiments. Supplementary Fig. 5 demonstrates that PS in single dose 9 J/cm² did not change the cerebral blood flow (CBF) on macro- and microcirculation, suggesting direct PS effects on LVs.

Using the established protocol for PS on healthy mice, we studied PS 9 J/cm² effects on the RBCs clearance from the brain in mice with IVH. The results presented in Fig. 2d–j demonstrate that the number of RBCs in dcLNs was significantly higher in the group IVH + PS vs. the IVH group without PS ($(5.68 \pm 0.75) \times 10^5$ per mm³ vs. $(1.76 \pm 0.42) \times 10^5$ per mm³, $p = 0.000136$, $n = 6$). There were no RBCs in the sham group.

### The mechanisms of PS effects on the lymphatic vessels

Recent work has shown that the basal MLVs have the valves as the peripheral collecting lymph vessels and are hotspots for the clearance of CSF macromolecules[39]. In our previous works on the mesenteric lymphatic vessels, we discovered that PS increases lymphatic drainage via the PS-mediated modulation of LVs tone and the permeability of LVs[24,25,27]. Here, we tested our hypothesis that PS can cause dilation of the basal MLVs providing faster lymphatic clearance of RBCs from the brain. Our results revealed that the development of IVH was not

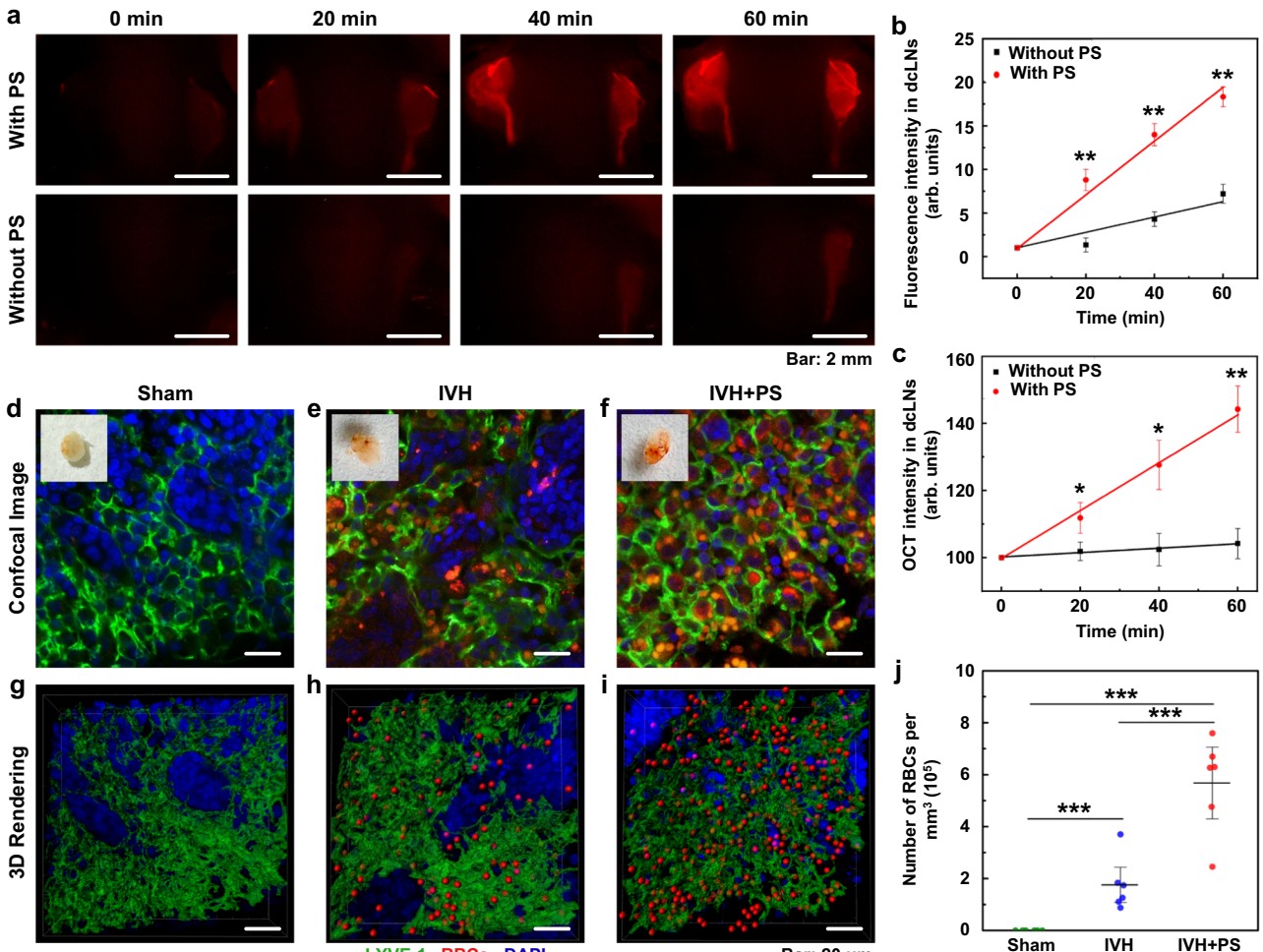

**Fig. 2 | The PS 9 J/cm² stimulation of lymphatic clearance of RBCs and macromolecules from mouse brain. a** Representative fluorescent images (from >3 replicates) of EBD clearance from the right lateral ventricle into dcLNs with and without PS in healthy adult mice. Scale bar: 2 mm. **b** Quantitative analysis of fluorescence intensity of EBD accumulation in dcLNs in healthy adult mice with and without PS. $n = 10$ mice in each group, pooled from 3 independent experiments ($P_{20 min} = 0.007$; $P_{40 min} = 0.005$; $P_{60 min} = 0.005$). **c** The OCT data of GNRs accumulation in dcLNs after its intraventricular injection with and without PS in healthy mice. $P_{20 min} = 0.047$; $P_{40 min} = 0.022$; $P_{60 min} = 0.005$. $n = 10$ mice in each group. **d–f** Representative confocal images (from >3 replicates) of dcLN 1 h after the injection of saline (**d**), blood without PS (**e**) and blood+PS (**f**) into the right later

ventricle. The insets in (**d–f**) represent photos of dcLN; and (**g–i**) 3D rendering of dcLN 1 h after the injection of saline (**g**), blood without PS (**h**) and blood+PS (**i**) into the right lateral ventricle (the volume of dcLN was $135 \times 135 \times 40$ μm³). Scale bar: 20 μm. **j** The number of RBCs in dcLN 1 h after the blood injection into the right later ventricle with and without PS. The LVs were labeled by LYVE-1 (green color) and PROX-1 (yellow color), RBCs were imaged by their autofluorescence (red color). $P(\text{Sham vs IVH}) < 0.001$; $P(\text{IVH vs IVH+PS}) < 0.001$; $P(\text{Sham vs IVH+PS}) < 0.001$. $n = 6$ mice in each group. All values are presented as Mean ± SEM. Wilcoxon tests were used for (**b, c**); and one-way ANOVA with Turkey's multiple-comparison tests were used for (**j**) ($F_{(2, 15)} = 34.720$). The statistical tests involved two-sided analyses. *$P < 0.05$, **$P < 0.01$ and ***$P < 0.001$. Source data are provided as a Source Data file.

accompanied by statistically significant changes in the diameter of basal MLVs ($20.23 \pm 1.07$ μm for the IVH group and $19.32 \pm 1.29$ μm for the sham group, $n = 6$ in each group, $p = 0.985$). However, PS (single dose 9 J/cm²) significantly dilated the basal MLVs in the IVH + PS compared with the IVH group without PS ($28.39 \pm 1.37$ μm vs. $20.23 \pm 1.07$ μm, $n = 6$ in each group, $p = 0.000739$) (Fig. 3a, b).

The mechanism by which PS affects lymphatics has not been sufficiently explored. There is evidence that it might be via PS-mediated stimulation of nitric oxide (NO) production[40–43]. NO is a vasodilator that modulates the lymphatic vessel contractility and subsequent lymph flow via multiple points in the Ca²⁺-contraction pathway[44,45].

Considering these facts, we hypothesized that PS-mediated lymphatic removal of RBCs from the mouse brain might be due to PS-stimulation of the NO production in the lymphatic endothelium and PS-mediated regulation of the alternation of phases of relaxation and contraction of LVs. The MLVs express all of the molecular hallmarks of lymphatic proteins typical for the peripheral lymphatic vessels[46]. There

are no real-time technologies for monitoring MLVs functions because they are extremely thin and small transparent vessels. Therefore, as a lymphatic function model, we used the mesenteric lymphangion to monitor the PS effects on the lymphatic contractility and the NO production in the isolated lymphatic endothelial cells (LECs). Our OCT data revealed that PS 3 J/cm² (PS dose was similar to those on the brain surface) significantly increased lymphangion contraction (Fig. 3c, Supplementary Movies 1 and 2). In in vitro experiments on LECs from the mesenteric tissue, we showed a significant increase in the 24 h accumulation of $NO_2^-$ in the cell culture medium after PS 3 J/cm² compared with the accumulation of $NO_2^-$ produced by LECs without PS (Fig. 3d).

Thus, these findings suggest the essential role of NO in the PS-mediated modulation of cleansing functions of the basal MLVs.

Additionally, to confirm the NO-ergic mechanism of PS-related stimulation of cleansing functions of MLVs, we studied the effects of blockage of NO-synthases (NOS) using L-NAME on PS-mediated changes in the diameter of MLVs and the RBCs accumulation in

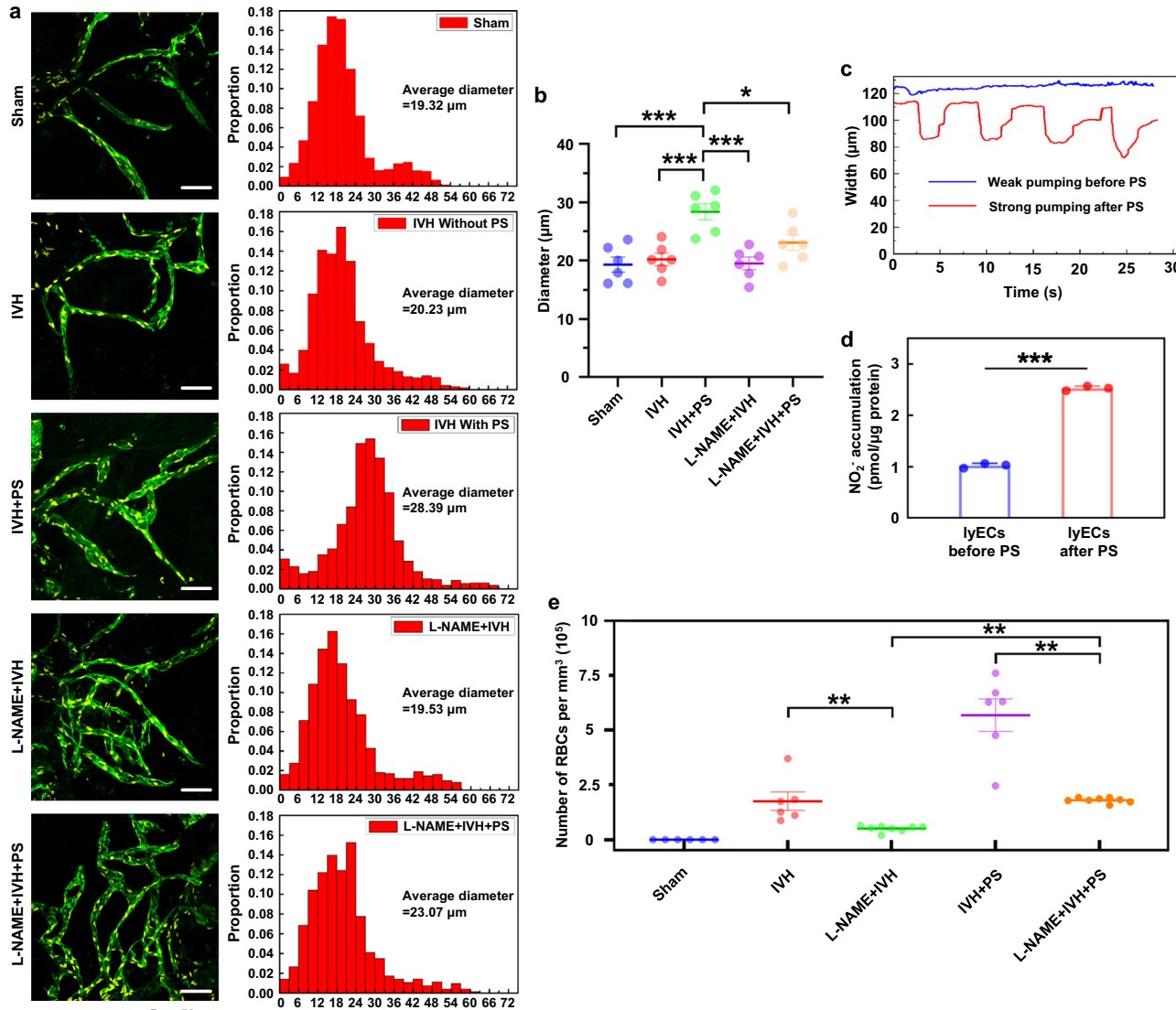

**Fig. 3 | The mechanisms of the PS effects on the LVs. a** Representative fluorescent images (from 2 replicates) of basal MLVs (labeled with LYVE-1, green and PROX-1, yellow) in each group. Scale bar: 20 μm. **b** Quantitative analysis (from 2 replicates) of the diameter of MLVs in each group. $P$(Sham vs IVH+PS)<0.001; $P$(IVH vs IVH+PS) <0.001; $P$(IVH+PS vs L-NAME+IVH)<0.001; $P$(IVH+PS vs L-NAME+IVH+PS) = 0.039. $n$ = 6 mice in each group. **c** The OCT in vivo assay of contractility of the mesenteric lymphangion before and after PS. $n$ = 3 mice in each group. **d** Total nitrite and nitrate content, measured by NO analyzer, in the medium of primary LECs. P (before PS vs after PS)<0.001. $n$ = 3 mice in each group. **e** Quantitative analysis of the number of RBCs in dcLNs in each group. $P$(IVH vs L-NAME+IVH) = 0.008; $P$(IVH+PS vs L-NAME+IVH+PS) = 0.007; $P$(L-NAME+IVH vs L-NAME+IVH+PS) = 0.002. $n$ = 6 mice in the Sham, the IVH and the IVH+PS, and $n$ = 8 mice in the L-NAME+IVH groups and the L-NAME+IVH+PS groups. All values are presented as Mean ± SEM. One-way ANOVA with Turkey's multiple-comparison tests were used for (**b**) ($F$(4,25) = 9.607) and (**e**) ($F$(4.29) = 40.244); and the independent-samples T tests were used for (**d**). The statistical tests involved two-sided analyses. *$P$ < 0.05, **$P$ < 0.01 and ***$P$ < 0.001. Source data are provided as a Source Data file.

dcLNs in mice with IVH (Fig. 3a, b). We found that the blockade of NOS significantly reduced the PS-related relaxation of MLVs in the L-NAME + IVH + PS group vs. the IVH + PS without L-NAME (23.08 ± 1.33 μm vs. 28.39 ± 1.37 μm, $n$ = 6, $p$ = 0.039). This led to the fact that the diameter of MLVs was not different between the L-NAME + IVH + PS group and the IVH group (23.08 ± 1.33 μm for the L-NAME + IVH + PS and 20.23 ± 1.07 μm for the IVH, $n$ = 6, $p$ = 0.490), i.e., PS did not affect the diameter of MLVs in IVH mice treated with L-NAME. The L-NAME also reduced the PS-simulation of removal of RBCs from the right lateral ventricle into dcLN (Fig. 3e). Indeed, the number of RBCs in dcLNs was significantly lesser in the L-NAME + IVH + PS group vs. IVH + PS group ((1.80 ± 0.04) × 10⁵ per mm³ vs. (5.68 ± 0.75) × 10⁵ per mm³, $n$ = 8 for L-NAME + IVH + PS and $n$ = 6 for IVH + PS, $p$ = 0.007).

Interestingly, despite the fact that there were no changes in the diameter of MLVs between the L-NAME + IVH and the IVH groups

(19.53 ± 1.07 μm for the L-NAME + IVH and 20.23 ± 1.07 μm for the IVH, $n$ = 6, $p$ = 0.994), the NO-blockade significantly reduced the number of RBCs in dcLNs in the L-NAME + IVH vs. the IVH ((0.51 ± 0.05) × 10⁵ per mm³ vs. (1.76 ± 0.42) × 10⁵ per mm³, $n$ = 8 for L-NAME + IVH group and $n$ = 6 for IVH group, $p$ = 0.008) that was improved after PS ((1.80 ± 0.04) × 10⁵ per mm³ vs. (0.51 ± 0.05) × 10⁵, $n$ = 8, $p$ = 0.002) (Fig. 3e).

These observations allowed us to assume that the PS effects on the lymphatic RBCs clearance can be realized mainly through NO-ergic tone regulation and contractility of the basal MLVs, which, however, does not exclude the involvement of other systemic and cellular mechanisms of RBCs clearing from the brain.

To answer the question of whether there is a pathway of RBCs clearance from the brain independent of lymphatics, we studied the effects of photodynamic impairment of MLVs on the accumulation of

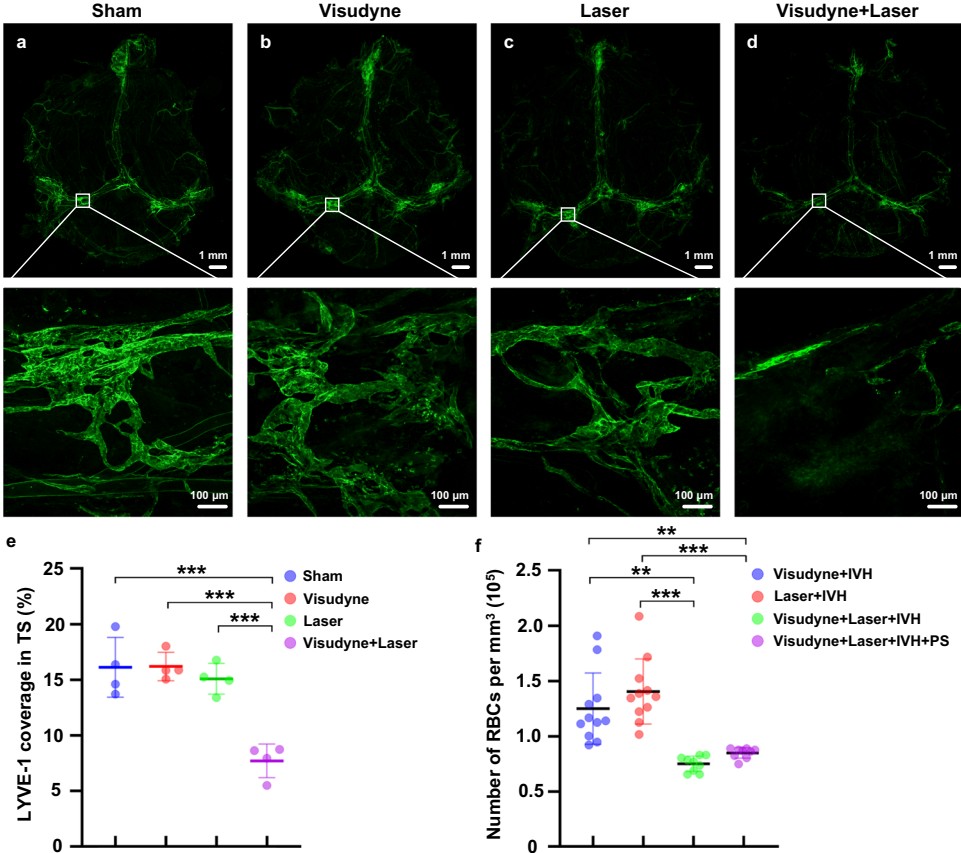

**Fig. 4 | The effects of ablation of MLVs on the lymphatic removal of RBCs from mouse brain. a–d** Representative confocal images (from 4 replicates) of MLVs labeled with LYVE-1 (green) surrounding the transverse sinus in the Sham group (**a**- without ablation of MLVs), in the Visudyne group (**b**), in the Laser group (**c**) and after ablation of MLVs in the Visudyne+Laser group (**d**). Scale bar: 20 µm.
**e** Quantitative analysis of MLVs covering the transverse sinus in each group. *P*(Sham vs Visudyne+Laser)<0.001; *P*(Visudyne vs Visudyne+Laser)<0.001; *P*(Laser vs Visudyne+Laser)<0.001. *n* = 4 mice in each group. **f** Quantitative analysis of the RBCs number in dcLNs in each group. *P*(Visudyne+IVH vs Visudyne+Laser+IVH) = 0.002;

*P*(Visudyne+IVH vs Visudyne+Laser+IVH+PS) = 0.003; *P*(Laser+IVH vs Visudyne+Laser+IVH)<0.001; *P*(Laser+IVH vs Visudyne+Laser+IVH+PS)<0.001. *n* = 11 mice in the Visudyne+IVH and the Laser+IVH groups, and *n* = 9 mice in the Visudyne+Laser+IVH and the Visudyne+Laser+IVH+PS groups. All values are presented as Mean ± SEM. One-way ANOVA with Turkey's and Dunnett's T3 multiple-comparison tests were used for (**e**) ($F_{(3,12)}$ = 20.533) and (**f**) ($F_{(3,36)}$ = 17.882), respectively. The statistical tests involved two-sided analyses. **P* < 0.05, ***P* < 0.01 and ****P* < 0.001. Source data are provided as a Source Data file.

RBCs in dcLNs (Fig. 4). The ablation of MLVs was performed by the intracisternal magna injection of Visudyne and its excitation by 689-nm laser light. Fig. 4a−e evidently show that 7 days after ablation of MLVs, the lymphatic coverage of the transverse sinus was significantly decreased (the Sham group vs. the Laser+Visudyne group, *p* = 0.000128; the Laser group vs. the Laser + Visudyne group, *p* = 0.000425; the Visudyne vs. the Laser + Visudyne group, *p* = 0.000117, *n* = 4 in each group). There were no changes in the MLV network in the Visudyne only and the Laser only groups (Fig. 4e). No difference in the CBF was observed between sham mice and the Laser+Visudyne group (Supplementary Fig. 6). Thus, photodynamic ablation induced the effective loss of MLV covering the transverse sinus (MLVs in this region of the meninges plays an important role in the brain drainage[39]) that is consistent with the findings presented by Chen et al.[31].

On the seventh day of MLVs ablation, IVH was induced, and the RBCs accumulation in dcLNs was examined 1 h after the intraventricular injection of blood. The ablation of MLVs caused a dramatic decrease n the number of RBCs in dcLNs (the Laser+IVH group vs. the Laser+Visudyne+IVH group, *p* = 0.000089; the Visudyne+IVH group vs. the Laser+Visudyne+IVH group, *p* = 0.002, *n* = 11 for Laser+IVH and Visudyne+IVH, and *n* = 9 for Visudyne+IVH). The PS did not improve the removal of RBCs from the brain into dcLNs after MLV ablation (the Laser+Visudyne+IVH + PS group vs. the Laser+Visudyne+IVH group, *n* = 9 in each group, *p* = 0.797) (Fig. 4f).

Notice that the photodynamic impairment of MLVs only reduced the RCBs clearance from the brain but did not entirely block it, as evidenced by the presence of a small number of RBCs in dcLNs (Fig. 4f). This fact suggests alternative pathways for the RBCs clearance from the brain, probably through the CSF drainage via the cranial nerves[47–49]. However, PS does not influence on such pathways.

## Study of the clinical significance of PS-mediated lymphatic clearance of RBCs from the brain

The main obstacle to PS application is limited laser penetration into the brain due to the scattering effects of the skull. However, PS-acceleration of the RBCs clearance from the brain via MLVs can be clinically significant for the therapy of intracranial hemorrhages in newborns, where PS can be applied through the fontanelles in the neonatal skull, which is also a window into MLVs localized along the sagittal and transverse sinuses[39].

Importantly, the lymphatic system can only be one pathway for the RBCs clearance from the brain during the first days of life[50,51]. There are two pathways for CSF drainage, including the extracerebral lymphatics and the arachnoid villi of the cerebral venous system[48]. However, the arachnoid villi or granulations do not exist prenatally. They only start to become visible in the dura in infants, increasing their density with age, and in adults, they exist in abundance[50,51]. In recent rodent studies has been discovered that MLVs appear very early in

embryos[52]. After postnatal days (PD) 4, MLVs grow along the middle meningeal artery reaching the transversal and the sagittal sinuses to the PD 8 and 28, respectively. At PD3, MLVs are presented in dcLNs.

Rats are commonly used as experimental models of neonatal studies due to the similarity of brain maturation in newborn rat pups and human neonates, based on gross anatomical analysis and brain function studies[53–57]. Since IVH is common in preterm infants born before the 30th gestational week[1–3], newborn rat pups PD4 were selected for the study of PS effects on the brain lymphatic functions. The newborn rats PD4 have been suggested to reach a brain maturation level that is similar to a preterm human brain of 23–28 weeks gestation[53–57].

Since there is no information about the anatomical and functional maturation of the lymphatic drainage and clearing in newborn rats, we performed the study of lymphatic removal of FITC-dextran 70 kDa (FITCD) from the right lateral ventricle to dcLNs of PD4 pups using the published protocol[58]. Suppression of the brain lymphatic functions after the development of subdural and subarachnoid hemorrhages as well as the traumatic brain trauma in adult mice has been reported in several publications[29–31,59]. Based on these facts, we hypothesized that IVH will be accompanied by a decrease in lymphatic drainage and clearing of the brain tissues in newborn rats. Indeed, Fig. 5a–e show that the IVH group vs. the control (Sham) group demonstrated suppression of lymphatic evacuation of FITCD from the right lateral ventricle leading to reduced spreading of tracer in the dorsal and ventral aspects of the brain as well as causing a decrease of its accumulation in dcLNs. The quantitative analysis showed that the intensity of fluorescent signal from FITCD in the IVH group vs. the control group was significantly decreased in the dorsal and ventral aspects of the brain as well as in dcLNs ($0.29 \pm 0.02$ arb. units vs. $0.55 \pm 0.12$ arb. units, $p = 0.01587$, in the dorsal part of the brain; $0.35 \pm 0.05$ arb. units vs. $0.61 \pm 0.11$ arb. units, $p = 0.03175$, in the ventral part of the brain; $1.19 \pm 0.27$ arb. units vs. $8.58 \pm 1.10$ arb. units, $p = 0.02857$, in dcLNs; $n = 5$ in each group). These results confirm our hypothesis that IVH causes significant suppression of lymphatic functions in newborn rats.

Thus, PD4 rat pups demonstrate the clearance of FITCD from the brain into the peripheral lymphatic system suggesting that they have enough effective lymphatic mechanisms of drainage and clearance of the brain that is suppressed after the development of IVH.

## PS stimulates lymphatic clearance of RBCs and macromolecules from the brain of newborn rats

Since, in previous experiments on adult mice, the PS dose 9 J/cm² was found as optimal, we used this PS dose in newborn rats that was 4 J/cm² after adaptation of the PS dose to their thin skull transparency (See Session "*Laser radiation scheme and dose calculation*" in Methods). There were no changes in the temperature on the cortex surface (single PS dose) and no morphological changes (the PS course) in the brain tissues after PS 4 J/cm² (Supplementary Table 5, Supplementary Fig. 7).

In in vivo experiments on healthy pups, we demonstrated that PS application in a single PS dose of 4 J/cm² significantly stimulated the removal of EBD, GNRs and RBCs from the right lateral ventricle into dcLNs (Fig. 5f–o). Indeed, the intensity of EBD fluorescence in dcLNs on 60 min of observation was higher in pups after PS compared with pups without PS ($2.36 \pm 0.14$ arb. units vs. $1.66 \pm 0.10$ arb. units, $n = 5$ in each group, $p = 0.037$) (Fig. 5f, g). The rate of GNRs accumulation in dcLNs was also higher in pups after PS vs. pups without PS ($0.14 \pm 0.03$ arb. units vs. $-0.09 \pm 0.05$ arb. units, $n = 5$ in each group, $p = 0.005$) (Fig. 5h).

The results presented in Fig. 5i–o show that the number of RBCs in dcLNs was higher in the IVH + PS group vs. the IVH group without PS ($(0.23 \pm 0.01) \times 10^5$ per mm³ vs. $(0.16 \pm 0.01) \times 10^5$ per mm³, $n = 6$ in each group, $p = 0.000052$).

In the next step, we answered the question whether PS could improve lymphatic functions in PD4 pups after IVH. Our results clearly demonstrate that the FITCD removal from the brain to dcLNs was significantly improved by PS in newborn rats with IVH and increased in the control group (Fig. 5a–e). So, in the IVH + PS vs. the IVH no PS groups, the intensity of signal from FITCD was significantly higher in the dorsal and ventral parts of the brain as well as in dcLNs ($0.53 \pm 0.07$ arb. units vs. $0.29 \pm 0.02$ arb. units, $p = 0.00793$, in the dorsal part of the brain; $1.04 \pm 0.22$ arb. units vs. $0.35 \pm 0.05$ arb. units, $p = 0.00793$, in the ventral part of the brain; $7.77 \pm 1.07$ arb. units vs. $1.19 \pm 0.27$ arb. units, $p = 0.01587$, in dcLNs; $n = 5$ in each group). In the control group +PS vs. the control group no PS, the intensity of signal from FITCD was higher in the dorsal and ventral parts of the brain as well as in dcLNs ($1.47 \pm 0.35$ arb. units vs. $0.55 \pm 0.12$ arb. units, $p = 0.03175$, in the dorsal part of the brain; $1.46 \pm 0.21$ arb. units vs. $0.61 \pm 0.11$ arb. units, $p = 0.01587$, in the ventral part of the brain; $19.87 \pm 3.23$ arb. units vs. $8.58 \pm 1.10$ arb. units, $p = 0.02857$, in dcLNs; $n = 5$ in each group).

Thus, these series of in vivo and ex vivo experiments demonstrated that PS effectively improves lymphatic removal of RBCs from the brain of PD4 pups with IVH and increases lymphatic clearance of macromolecules (GNRs, FITCD and EBD) from the brain of healthy newborn rats.

## Therapeutic effects of the PS-course on recovery of newborn rats and adult mice after IVH

The PS course significantly reduced the intraventricular hemorrhage size in PD4 pups (Fig. 6a). Indeed, the size of intraventricular hemorrhage 11 days vs. 3 days after IVH was decreased by 72.5% the IVH + PS ($0.09 \pm 0.01$ mm² vs. $0.29 \pm 0.02$ mm², $p = 0.00037$, $n = 7$ in each group) and was decreased only by 34.5% in the IVH without PS ($0.22 \pm 0.01$ mm² vs. $0.29 \pm 0.02$ mm², $p = 0.1608$, $n = 7$ in each group). Thus, the size of hematoma was reduced 2.3 times in the IVH + PS group vs. the IVH without PS on the 11th days after IVH ($0.09 \pm 0.01$ mm² vs. $0.22 \pm 0.01$ mm², $p = 0.0014$, $n = 7$ in each group).

The PS-course promoted the entire recovery of the ventricles, which were dilated after IVH (Fig. 6b and Supplementary Fig. 8). Indeed, 3 days after IVH, the size of the right lateral ventricle was higher in the IVH vs. the sham group ($0.98 \pm 0.03$ mm² vs. $0.21 \pm 0.01$ mm², $p = 0.00001$, $n = 7$ in each group). After the PS course, the size of the right lateral ventricle returned to the normal values and did not differ between the IVH + PS and the sham+PS groups ($0.23 \pm 0.01$ mm² and $0.22 \pm 0.01$ mm², respectively, $p = 0.9992$, $n = 7$ in each group). However, in the IVH group without PS, the size of the right lateral ventricle remained large vs. the sham group ($0.43 \pm 0.06$ mm² vs. $0.18 \pm 0.01$ mm², $p = 0.00007$, $n = 7$ in each group), despite the trend towards recovery compared with pups 3 days after IVH ($0.43 \pm 0.06$ mm² vs. $0.98 \pm 0.12$ mm², $p = 0.00037$, $n = 7$ in each group) (Fig. 6b and Supplementary Fig. 8). Thus, PS contributed the effective recovery of the ventricular system of brain after IVH.

After the PS course, rat pups demonstrated complete recovery from the vasogenic edema, which was observed in pups 3 days after IVH. Three days after IVH, the size of PVS increased and was higher in the IVH group vs. the sham group ($0.0199 \pm 0.0017$ μm vs. $0.0016 \pm 0.0003$ μm, $p = 0.00006$, $n = 7$ in each group) (Fig. 6c, d, g). Eleven days after the IVH, the size of PVS was completely restored after the PS course ($0.0014 \pm 0.0003$ μm in the IVH + PS group and $0.0013 \pm 0.0002$ μm in the sham+PS group, $p = 0.9999$, $n = 7$ in each group) (Fig. 6c, f, i). However, the size of PVS remained large on 11th day of observation in the IVH without PS and was higher compared with the sham group ($0.0099 \pm 0.0008$ μm vs. $0.0017 \pm 0.0004$ μm, $p = 0.0002$, $n = 7$ in each group) (Fig. 6c, e, h).

The PS-mediated improvement of brain drainage was associated with better recovery of the blood-brain barrier (BBB) after IVH (Fig. 6j–o). Indeed, the FITCD leakage was observed 3 and 11 days after

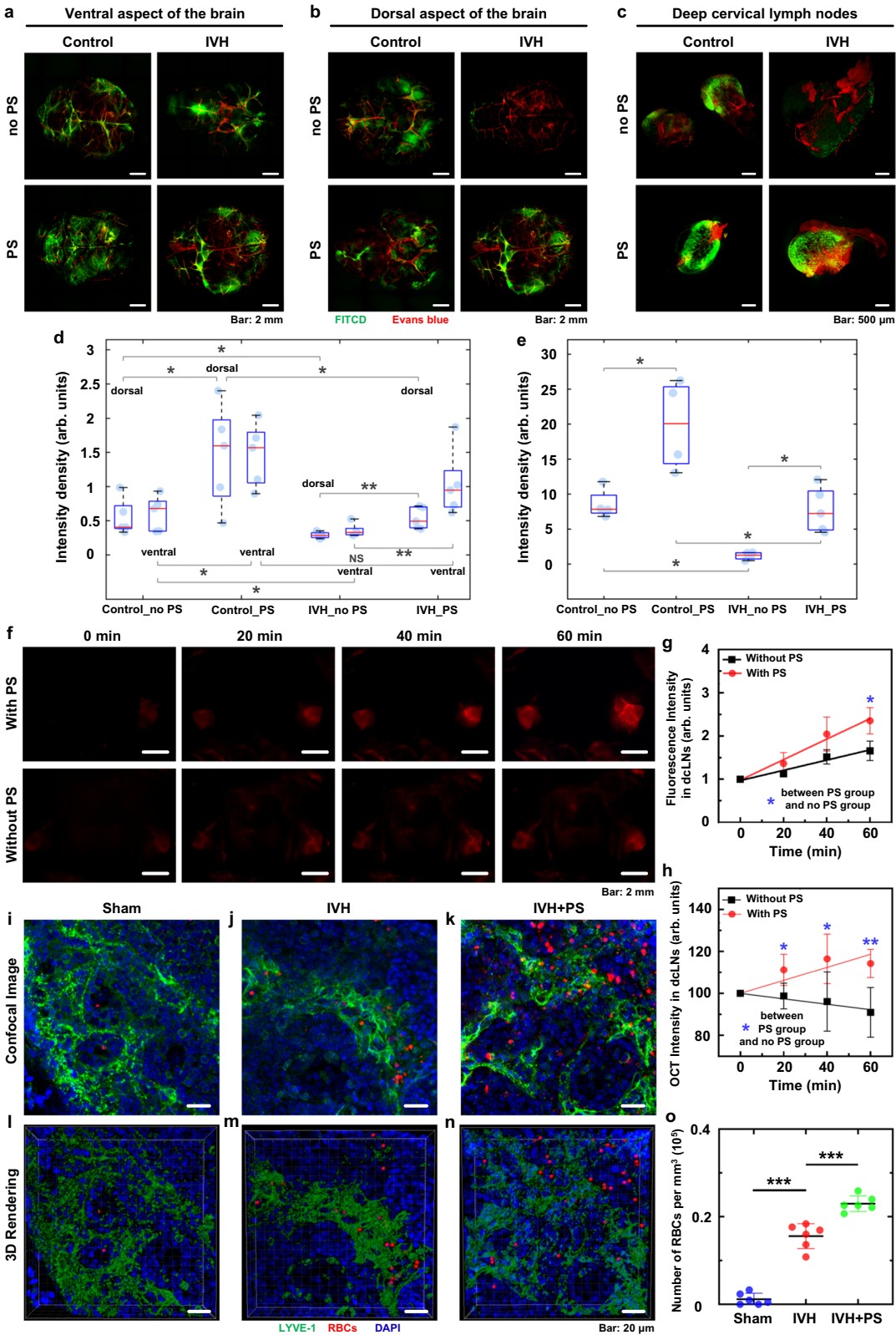

IVH vs. the sham groups and was quantified by measuring the fluorescence intensity of FITCD in the perivascular area (Fig.6j, k and m, n) (1.073 ± 0.016 arb. units vs. 0.069 ± 0.007 arb. units, 3 days after IVH, $p = 0.000583$ and 0.763 ± 0.090 arb. units vs. 0.057 ± 0.007 arb. units, 11 days after IVH, $p = 0.000583$; $n = 7$ in each group). However, there was no the FITCD leakage in the IVH + PS group and the Sham+PS group (Fig. 6l, o) (0.080 ± 0.010 arb. units in the IVH + PS group and

0.071 ± 0.005 arb. units in the Sham+PS group, $n = 7$ in each group, $p = 0.037879$).

The PS-course 4 J/cm² improved the neurological status after IVH in newborn rats. The timeline of motor and neurodevelopment reflex testing is presented in Supplementary Fig. 9. Indeed, the score of the motor tests, such as hind-limb and front-limb suspension, decreased in the IVH group vs. the sham group

**Fig. 5 | The PS 4 J/cm² stimulation of lymphatic clearance of RBCs and macromolecules from the brain of PD4 pups. a, b** Representative fluorescent images (from 2 replicates) of the distribution of FITCD (green color) in the ventral (**a**) and dorsal (**b**) aspects of the brain without and after PS in the control and IVH groups, respectively. The cerebral blood vessels were filled by 1% Evans Blue (red color). Scale bar: 2 mm. **c** Representative fluorescent images (from 2 replicates) of the accumulation of FITCD in dcLNs without and after PS in the control and IVH groups. Scale bar: 500 μm. **d, e** The quantitative analysis of the intensity signal from FITCD in the brain (**d**) and in dcLNs (**e**). Ventral aspect of the brain: $P$(Control_no PS vs Control_PS) = 0.01587; $P$(Control_no PS vs IVH_no PS) = 0.03175; $P$(Control_PS vs IVH_PS) = 0.22222; $P$(IVH_no PS vs IVH_PS) = 0.00793. Dorsal aspect of the brain: $P$(Control_no PS vs Control_PS) = 0.03175; $P$(Control_no PS vs IVH_no PS) = 0.01587; $P$(Control_PS vs IVH_PS) = 0.03175; $P$(IVH_no PS vs IVH_PS) = 0.00793. dcLNs: $P$(Control_no PS vs Control_PS) = 0.02857; $P$(Control_no PS vs IVH_no PS) = 0.02857; $P$(Control_PS vs IVH_PS) = 0.01587; $P$(IVH_no PS vs IVH_PS) = 0.01587. $n = 5$ rats in each group. The box indicates the upper and lower quantiles, the thick line in the box indicates the median and whiskers indicate 2.5th and 97.5th percentiles. **f** Representative fluorescent images (from 2 replicates) of EBD clearance from the right lateral ventricle into dcLNs with and without PS in healthy newborn rats. Scale bar: 2 mm. **g** Quantitative analysis of fluorescence intensity of EBD accumulation in dcLNs in healthy newborn rats with and without PS. $P_{60\ min} = 0.037$. $n = 5$ rats in each group. **h** The OCT data of GNRs accumulation in dcLNs after its intraventricular injection with and without PS in healthy newborn rats. $P_{20\ min} = 0.001$; $P_{40\ min} = 0.001$; $P_{60\ min} = 0.005$. $n = 5$ rats in each group. **i–k** Representative confocal images (from 2 replicates) of dcLN 1 h after the injection of saline (**i**), blood without PS (**j**) and blood+PS (**k**) into the right later ventricle; and (**l–n**) 3D rendering of dcLN 1 h after the injection of saline (**l**), blood without PS (**m**) and blood+PS (**n**) into the right later ventricle (the volume of dcLN was $135 \times 135 \times 40$ μm³). Scale bar: 20 μm. **o** The number of RBCs in dcLN 1 h after the blood injection into the right later ventricle with and without PS. The LVs were labeled by LYVE-1 (green color), RBCs were imaged by their autofluorescence (red color), and nucleus were labeled by DAPI (blue color). $P$(Sham vs IVH)<0.001; $P$(IVH vs IVH+PS) < 0.001. $n = 6$ rats in each group. All values are presented as Mean ± SEM. Wilcoxon tests were used for (**d, e, g, h**); and one-way ANOVA with Turkey's multiple-comparison tests were used for (**o**) ($F(2,15) = 169.597$). The statistical tests involved two-sided analyses. *$P < 0.05$, **$P < 0.01$ and ***$P < 0.001$. Source data are provided as a Source Data file.

suggesting the impairment of motor function after IVH ($n = 7$ in each group, $p = 0.009$ for the front-limb test and $p = 0.002$ for the hind-limb test). However, the scores of these tests were essentially increased after PS compared with pups without the PS therapy ($n = 7$ in each group, $p = 0.080$ for the front-limb test and $p = 0.009$ for the hind-limb test) (Fig. 6p, q). The neurodevelopment reflexes were impaired in pups with IVH as evidenced by a decrease in score of the cliff avoidance and the gait tests in the IVH group compared with the sham group ($n = 7$ in each group, $p = 0.002$ for the cliff avoidance test and $p = 0.002$ for the gait test). After the PS course, pups performed these tests much better than animals without the PS therapy ($n = 7$ in each group, $p = 0.002$ for the cliff avoidance test and $p = 0.002$ for the gait test) (Fig. 6r, s).

Our results demonstrate that no animals died in the IVH + PS group (10 of 10) and the sham group (10 of 10), while 3 of 10 pups died in the IVH no PS ($p = 0.092$, X2 test Log Rank (Mantel−Cox) = 2.832, Kaplan−Meier method).

In the final step, we studied the therapeutic effects of PS on adult mice with IVH. The schematic diagram of the time points of experiments and the number of animals in the experimental groups are presented in Supplementary Fig. 10c. The adult mice (2-3 months old) did not show statistically significant differences in the mortality between the IVH and IVH + PS group ($p = 0.161$, X2 test Log Rank (Mantel−Cox) = 1.961, Kaplan−Meier method) because only 2 mice died (2 of 20) in the IVH group and no dead mice were in the IVH + PS group ($n = 20$ in each group) (Supplementary Fig. 10a). No animals died in the sham groups ($n = 20$). However, despite the fact that mortality among adult mice was low, they demonstrated significant IVH-mediated changes in the brain morphology and behavior that were improved after the PS course.

The IVH was accompanied by the formation of periventricular hematoma, which was reduced after 11 days of recovery. The PS course significantly improved brain recovery after IVH. Indeed, the size of periventricular hematoma 11 days vs. 3 days after IVH reduced by 60.3% in the IVH + PS group ($0.27 \pm 0.05$ mm² vs. $0.68 \pm 0.12$ mm², $n = 10$ in each group, $p = 0.000001$), while this parameter was decreased only by 36.6% in the IVH group without PS ($0.43 \pm 0.12$ mm² vs. $0.68 \pm 0.12$ mm², $n = 10$ in each group, $p = 0.000299$, the Welch's test) (Supplementary Fig. 10b, l).

Supplementary Fig. 10f, g clearly demonstrate that the number of RBCs in dcLNs gradually decreased from 1 to 7 days of observation that was more pronounced in the IVH + PS group vs. the IVH group without PS (($5.34 \pm 0.05$) × 10⁵ per mm³ vs. ($2.04 \pm 0.21$) × 10⁵ per mm³, for 1 h after IVH, $p = 0.000008$, $n = 6$; ($3.12 \pm 0.17$) × 10⁵ per mm³ vs. ($2.32 \pm 0.14$) × 10⁵ per mm³, for 1 day after IVH, $p = 0.004$, $n = 6$; ($1.65 \pm 0.06$) × 10⁵ per mm³ vs. ($2.63 \pm 0.07$) × 10⁵ per mm³, for 3 days after IVH, $p = 0.000001$, $n = 6$; ($0.94 \pm 0.10$) × 10⁵ per mm³ vs. ($1.61 \pm 0.12$) × 10⁵ per mm³, $p = 0.0013$, for 7 days after IVH, $n = 6$).

The PS course effectively reduced the vasogenic edema observed in mice after IVH (Supplementary Fig. 10h–k, m). So, we observed the enlarged PVS in mice 3 days after IVH vs. sham mice ($0.027 \pm 0.007$ μm vs. $0.008 \pm 0.001$ μm, $n = 10$ in each group, $p = 0.0001$) that was followed by gradually recovery of PVS by day 11 of recovery. In this recovery time compared with 3 days after IVH, the PVS size was decreased to 55.6% in the IVH + PS group and to only 29.6% in the IVH group without PS-course ($0.012 \pm 0.001$ μm vs. $0.027 \pm 0.007$ μm, $n = 10$ in each group, p = 0.000009 and $0.019 \pm 0.004$ μm vs. $0.027 \pm 0.007$ μm, $n = 10$ in each group, $p = 0.006$, respectively).

The PS course promoted a faster recovery of the ventricles, which dilated after IVH (Supplementary Fig. 10n). Indeed, the right lateral ventricle size was significantly enlarged in mice 3 days after IVH vs. sham mice ($0.91 \pm 0.02$ mm² vs. $0.26 \pm 0.06$ mm², $n = 10$ in each group, $p = 0.00001$). By 11 days, the ventricle size was decreased (Supplementary Fig. 10n). However, during this time, the size of the right later ventricle was decreased to 53.8% in the IVH + PS group and 30.8% in the IVH group without the PS course ($0.42 \pm 0.01$ mm² vs. vs. $0.91 \pm 0.02$ mm², $n = 10$ in each group, $p = 0.000008$ and $0.63 \pm 0.01$ mm² $0.91 \pm 0.02$ mm², $n = 10$ in each group, $p = 0.005$, respectively).

The development of IVH was accompanied by a neurologic deficit with changes in locomotor and memory functions that were significantly improved after the PS course (Supplementary Fig. 11a–e). All behavior tests were performed 11 days after the injection of blood (the IVH and the IVH + PS groups) or saline (the sham groups) into the right later ventricle (Supplementary Fig. 10c). The locomotor deficits were evaluated by a 24-point neurologic scoring system, where higher scores indicate a more significant deficiency, and by the wire-hanging test, which measures gripping and forelimb strength (Supplementary Fig. 11c, d). Mice in the IVH group had a substantial locomotor deficit vs. the sham group ($n = 10$ in each group, $p = 0.0056$). Gripping and forelimb strength were significantly impaired in the IVH group compared with the sham group ($n = 10$ in each group, $p = 0.0058$). After the PS course, locomotor functions ($n = 10$ in each group, $p = 0.0059$) as well as gripping and forelimb strength were improved compared with the IVH group without the PS course ($n = 10$ in each group, $p = 0.0058$) (Supplementary Fig. 11c and d).

The immobility time in the forced swim and the tail suspension tests was longer in the IVH than in the sham group ($n = 10$ in each group, $p = 0.0059$ for the forced swim and $p = 0.0020$ for the tail suspension tests) (Supplementary Fig. 11a, b). Mice after the PS course improved the time of immobility in both tests ($n = 10$ in each group,

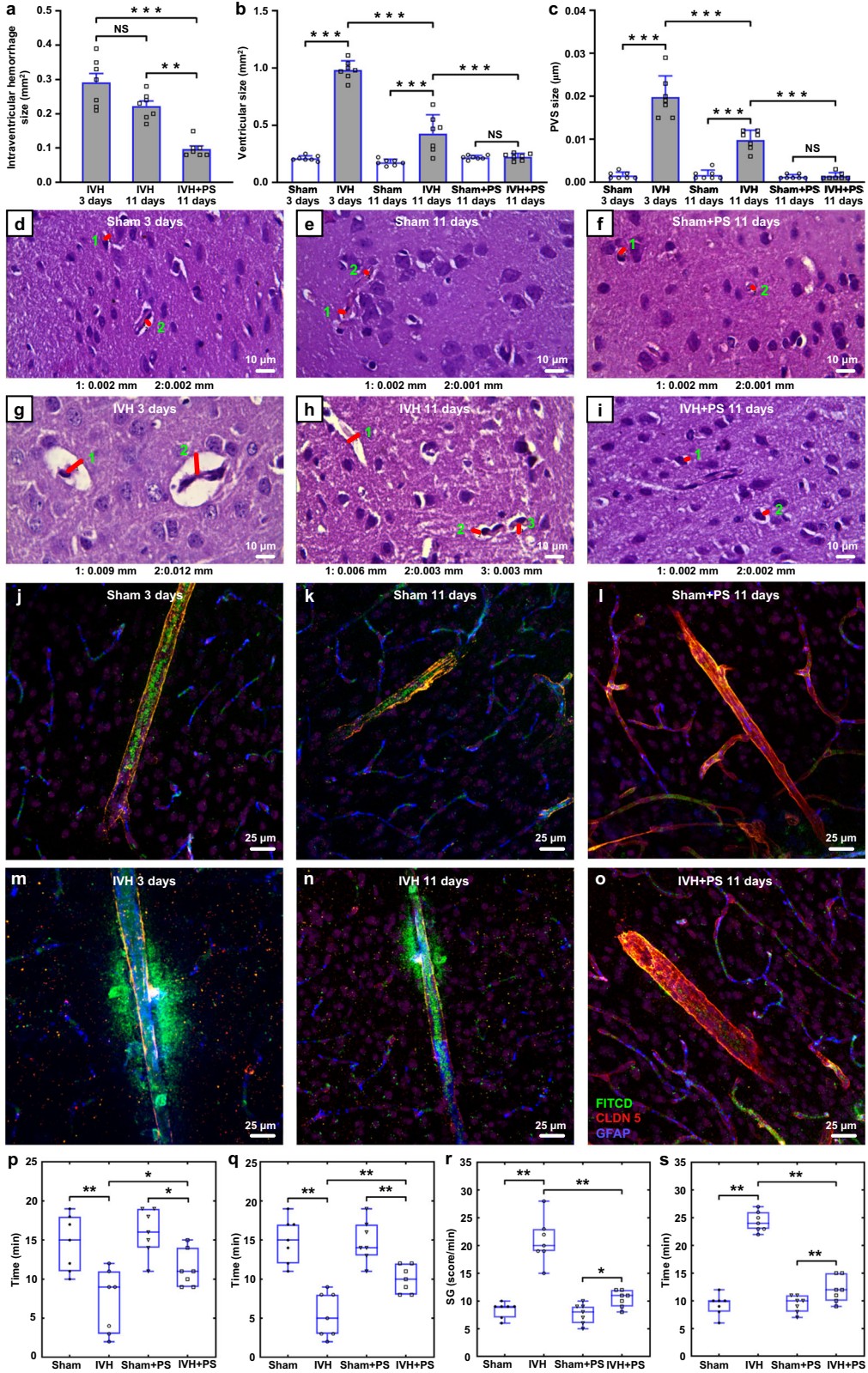

$p = 0.0059$ for both the forced swim and the tail suspension tests (Supplementary Fig. 11a, b).

In the novel object recognition test, sham mice spent more time exploring the novel object than the old ones ($n = 10$ in each group, $p = 0.0056$) (Supplementary Fig. 11e), but mice with IVH spent the similar time exploring the new and old objects ($n = 10$ in each group, $p = 0.1983$) (Supplementary Fig. 11e) suggesting the recognition memory deficit. After the PS course, mice performed this test similarly to the sham group and spent more time exploring the novel object than the old object, suggesting improvement of memory function.

## Discussion

In this pilot study performed on adult mice and newborn rats, we found that MLVs drain RBCs from the right lateral ventricle into dcLNs.

**Fig. 6 | Therapeutic effects of the PS 4 J/cm² course in PD4 newborn rats with IVH. a** The quantitative analysis of intraventricular hemorrhage size (mm²) formed on 3 days and 11 days after IVH with and without the PS-course. $P$(IVH_3 days vs IVH_11 days) = 0.1608; $P$(IVH_3 days vs IVH + PS_11 days)<0.001; $P$(IVH_11 days vs IVH + PS_11 days) = 0.0014. $n$ = 7 rats in each group. **b** The quantitative analysis of the right lateral ventricle size (mm³) on 3 days and 11 days after IVH with and without the PS-course. $P$(Sham_3 days vs IVH_3 days) < 0.001; $P$(Sham_11 days vs IVH _11 days) < 0.001; $P$(Sham + PS_11 days vs IVH + PS_11 days) = 0.9992; $P$(IVH_3 days vs IVH_11 days)<0.001; $P$(IVH_11 days vs IVH + PS_11 days)<0.001. $n$ = 7 rats in each group. **c** The quantitative analysis of the PVS size (μm) on 3 days and 11 days after IVH with and without PS-course. $P$(Sham_3 days vs IVH_3 days)<0.001; $P$(Sham_11 days vs IVH _11 days)<0.001; $P$(Sham + PS_11 days vs IVH + PS_11 days) = 0.9999; $P$(IVH_3 days vs IVH_11 days) = 0.0012; $P$(IVH_11 days vs IVH + PS_11 days) < 0.001. $n$ = 7 rats in each group. **d–i** Representative histological images (from >3 replicates) illustrating the normal brain tissues without vasogenic edema in the sham group 3 (**d**) and 11 (**e**) days after the injection of physiological saline into the right lateral ventricle as well as in the sham group treated by the PS course (**f**); the significant increase of the PVS size 3 days after IVH (**g**); the reducing the PVS size 11 days after IVH without the PS course (**h**) and the completely recovery of PVS in

the IVH + PS group (**i**). Scale bar: 10 μm. **j–o** Representative confocal images (from >5 replicates) of the intact BBB in the sham group 3 (**j**) and 11 (**k**) days after the injection of physiological saline into the right lateral ventricle as well as in the sham group treated by the PS course (**l**); the FITCD leakage in mice 3 (**m**) and 11 (**n**) days after IVH without PS; completely recovery of BBB in mice with IVH 11 days after IVH with the PS course (**o**) (green: FITCD; red: CLDN 5; blue: GFAP). Scale bar: 25 μm. $n$ = 7 rats in each group. **p, q** The score of motor tests, including the forelimb test (**p**) and hindlimb test (**q**). $P$(Sham vs IVH) = 0.009; $P$(IVH vs IVH+PS) = 0.000; $P$(Sham + PS vs IVH + PS) = 0.012 for (**p**), and $P$(Sham vs IVH) = 0.002; $P$(IVH vs IVH + PS) = 0.009; $P$(Sham + PS vs IVH + PS) = 0.005 for (**q**). $n$ = 7 rats in each group. **r, s** The score of neurodevelopment reflex tests, including the cliff avoidance test (**r**) and the gait test (**s**). $P$(Sham vs IVH) = 0.002; $P$(IVH vs IVH + PS) = 0.002; $P$(Sham + PS vs IVH + PS) = 0.038 for (**r**), and $P$(Sham vs IVH)=0.002; $P$(IVH vs IVH + PS) = 0.002; $P$(Sham + PS vs IVH + PS) = 0.007 for (**s**). $n$ = 7 rats in each group. The box indicates the upper and lower quantiles, the thick line in the box indicates the median and whiskers indicate 2.5th and 97.5th percentiles. All values are presented as Mean ± SEM. Welch's tests were used for (**a–c**); and Wilcoxon tests were used for (**p–s**). The statistical tests involved two-sided analyses. NS represents not significant, *$P$ < 0.05, **$P$ < 0.01 and ***$P$ < 0.001. Source data are provided as a Source Data file.

Using the meninges of humans died due to IVH, we found RBCs in MLVs. Our animal and human results provide strong support for the classic theory of the RBCs evacuation from the brain via lymphatic efflux[32,33,60]. Our data are also consistent with the newest discovery of the meningeal lymphatic pathway of clearance of RCBs after subdural[30], subarachnoid hemorrhage[31], and intracerebral hemorrhage[29]. In experiments with photodynamic ablation of MLVs, we clearly showed that the injury of MLVs significantly reduces the lymphatic clearance of RBCs indicating that MLVs are an essential pathway of the RBCs evacuation from the brain. However, the impairment of MLVs does not entirely block the RBCs clearance from the brain, as evidenced by the presence of a small number of RCBs in dcLNs. This fact suggests the existence of alternative pathways for RBCs removal from the brain[47–49]. Interestingly, dcLNs can serve as the anatomical station for both the efflux and influx of different macromolecules from dcLNs into the meninges that can be modulated by PS[27,61]. It can be assumed that PS can also stimulate the influx of RBCs from dcLNs into the meninges. However, Oehmichen et al. discovered in rabbits that only non-phagocytized RBCs arrive from the brain into dcLNs, where the extremely rapid ingestion and digestion of RBCs by lymph node macrophages[60]. Furthermore, our experiments clearly demonstrate that the PS effects on lymphatic clearance of RBCs do not appear in mice with ablation of MLVs. These observations lead to conclusions about direct evidence for the relationship between PS and lymphatic clearance of RBCs in the post-IVH period.

Since we found that MLVs are an essential route of RBCs evacuation from the brain into the peripheral lymphatics, it is rational to assume that augmentation of the lymphatic evacuation of RBCs from the brain may offer therapeutic approaches to alleviate IVH severity.

Here we investigated the effects of infrared PS (1267 nm) on the lymphatic removal of RBCs from the brain and the outcome after IVH. The infrared light of 800–1100 nm is widely used for the PS therapy of brain diseases[19,20,62]. However, the infrared PS has significant limitations, such as limited penetration into the brain due to light scattering and heating effects[63,64]. The light wavelength of 1300 nm has less scattering and can penetrate deeper into the brain[65]. Therefore, we selected PS (1267 nm) with the optimal dose for the experiments studying the light transmission, the changes in the temperature on the skull, and the surface of the brain of adult mice after PS 3–6–9–18–27 J/cm² as well as the morphological changes in the brain tissues after the PS-course 9 J/cm². Our data demonstrate that PS 9 J/cm² vs. other PS doses was most effective for stimulating lymphatic clearance of GNRs and EBD without the heating effect and any morphological changes in the brain tissues that

determined our choice of PS 9 J/cm² as the optimal dose for further investigations.

In in vivo experiments on adult mice and rat pups, we uncover that PS accelerates the RBCs evacuation from the ventricles. The seven-day PS course has therapeutic effects on adult mice, and particular, in newborn rats improving behavioral outcomes and providing fast reduce of hematoma size that is associated with full recovery of ventricular size and the permeability of BBB leading to disappearance of vasogenic edema in PD4 rats. Our results align with other animal and human data suggesting a greater clinical significance of transcranial near-infrared laser phototherapy for stroke and brain trauma[18–23].

To study the mechanisms of therapeutic effects of PS, we investigated PS influences on the diameter of basal MLVs and the lymphatic contractility as well as on the NO-mediated regulation of LVs tone. Our findings demonstrate that PS dilates MLVs and increases the NO production in LECs, associated with an increase in lymphangion contraction. These results suggest that PS-mediated activation of the NO synthesis in LECs and the lymphatic contractility might be the possible mechanisms responsible for PS-stimulation of lymphatic clearance of RBCs and macromolecules from the brain. Indeed, using L-NAME, we demonstrated that NO blocker significantly attenuates a PS-mediated increase of the MLVs diameter and the RBCs evacuation from the brain into dcLNs suggesting an essential role of NO in the PS effects on MLVs and lymphatic clearance of RBCs.

The described effects can be related to a PS-mediated increase in the activity of endothelial NOS[40]. The NO is a vasodilator that acts via stimulation of soluble guanylate cyclase to form cyclic-GMP, which activates protein kinase G causing the opening of calcium-activated potassium channels and reuptake of $Ca^{2+}$. The decrease in the concentration of $Ca^{2+}$ prevents myosin light-chain kinase from phosphorylating the myosin molecule, leading to the relaxation of lymphatic vessels[66]. There are several other mechanisms by which NO could control the lymphatic tone and contractility: 1) the activation of an iron-regulatory factor in macrophages[67], 2) the modulation of proteins such as ribonucleotide reductase[68] and aconitase[69]; the stimulation of the ADP-ribosylation of glyceraldehyde-3-phosphate dehydrogenase[70] and protein-sulfhydryl-group nitrosylation[71]; the hemoglobin as a byproduct of blood degradation can stimulate NO generation[72–75]. In our previous work, we revealed that PS causes relaxation of the mesenteric lymphatic endothelium associated with an increase in the permeability of lymphatic walls and a decrease in the expression of tight junction proteins[25]. We assume that the PS effects on the tone of MLVs facilitate drainage in fluid spaces of the brain that

drains RBCs/macromolecules from the ventricles along with the sub-arachnoid space, where RBCs partly penetrate into MLVs.

The PS has been shown to produce highly pleiotropic biological effects (other than lymphatic engagement), with many of them having high relevance to the pathobiology of IVH—both systemic and cellular[23,76]. The most well-studied mechanism of action of PS centers around cytochrome c oxidase (CCO), a unit four of the mitochondrial respiratory chain, responsible for the final reduction of oxygen to water using the electrons generated from glucose metabolism[77]. The NO may inhibit CCO enzyme activity, which can be dissociated by photons of light absorbed by CCO (contains two heme and two copper centers with different absorption spectra)[78]. When NO is dissociated, the mitochondrial membrane potential increases, more oxygen is consumed, more glucose is metabolized, and more ATP is produced by the mitochondria. It has been shown that there is a transient increase in reactive oxygen species (ROS) produced in the mitochondria when they absorb the photons delivered during PS. The idea is that this burst of ROS may trigger some mitochondrial signaling pathways leading to cytoprotective, antioxidant, and anti-apoptotic effects in the cells[79]. The NO that is released by photodissociation acts as a vasodilator as well as a dilator of LVs. Moreover, NO is also a potent signaling molecule and can activate a number of beneficial cellular pathways[80]. There is evidence that the beneficial effects of PS on the brain can be explained by stronger oxygen availability and oxygen consumption that improves the ATP production and the mitochondrial activity[22,81,82].

Our findings shed light on the therapeutic effects of low-level infrared PS on brain recovery after IVH and clearly demonstrate that MLVs drain RBCs from the ventricle into dcLNs that can be modulated by PS via activation of lymphatic drainage and clearing mechanisms. The main obstacle to PS application is limited laser penetration into the brain due to the scattering effects of the skull. However, a PS-acceleration of RBCs clearance from the brain via MLVs can be clinically significant for the therapy of brain hemorrhages in premature newborns, who are at highest risk for IVH and in whom PS can be applied through the fontanel. Indeed, we found that the PS course in PD4 rats (a brain maturation level in this age of rats is identical to a preterm human brain of 23–28 weeks gestation[53–57]) improves neurological outcomes, accelerates the RBCs evacuation from the ventricles and provides faster recovery of the BBB permeability and the brain drainage. We believe that a PS-related stimulation of the lymphatic clearance of RBCs and macromolecules from the brain may offer innovative therapeutic approaches to alleviate IVH severity in humans that needs further detailed clinical investigations. We found that PS also had the therapeutic effects in adult mice. We suggest that PS can be an effective treatment for IVH in adults, as has already been shown for treatment of stroke and traumatic brain injury[83,84]. However, given the loss of laser energy when passing through the skull, it can be assumed that that PS during deep sleep, when the brain's drainage system is activated itself[58,85,86], can significantly increase the therapeutic efficacy of PS, as has been demonstrated for the night therapy of Alzheimer's disease[58,87]. Further studies of optimal doses and wavelengths of PS, and advantages and limitations of PS-stimulation of the brain's drainage and MLVs in patients of different ages and with various brain hemorrhages could significantly help in the development of guidelines for the safe use of PS in humans. If our preclinical results are confirmed in further clinical trials, non-invasive PS-mediated stimulation of lymphatic clearance of RBCs can be a readily applicable, commercially attractive and viable technology for effective routine treatment of IVH and other types of brain bleedings.

## Methods
### Subjects
The experiments were performed on male BALB/c mice (25–28 g, 2–3 months) and male PD4 newborn Wistar rats (7–8 g), which during the study reached 14 days of age (18–20 g). All animals were maintained under specific pathogen-free conditions under controlled temperature (18–22 °C) and humidity (50–60%) and a 12-h dark/12-h light cycle (7 a.m. to 7 p.m.), with access to regular rodent's chow and sterilized tap water ad libitum. At the end of each experiment, animals were immediately sacrificed by cervical dislocation or by decapitation within seconds under deep anesthesia with isoflurane and using the guillotine (Stoelting Co, Wheat Lane, USA) and the DecapiCone® (Braintree Scientific Inc., Braintree, USA). All procedures were performed in accordance with the "Guide for the Care and Use of Laboratory Animals". The experimental protocols were approved by the Local Bioethics Commission of the Saratov State University (Protocol No. 7); Experimental Animal Management Ordinance of Hubei Province, P. R. China (No. 1000639903375); the Institutional Animal Care and Use Committee of the University of New Mexico, USA (#200247).

The human dura was collected in accordance with the Declaration of Helsinki as a statement of ethical principles for medical research involving human subjects, including research on identifiable human material and data[88]. The species of human meninges at autopsy were obtained from the Department of Pathological Anatomy at the Saratov Medical State University. The present study was performed according to a protocol approved by the Committee of Science and Research Ethics, the Saratov Medical State University. All personal data are stored in strict ethical control, and samples were coded before the analyses of tissue. In accordance with GOST R ISO 14155-2014 (Russian National Standard) of the Ministry of Health of the Russian Federation (Article 38 of the Federal Law dated November 21, 2011) and the Order of the Ministry of Health of the Russian Federation No. 354 dated June 6, 2013 "On the procedure for conducting pathological and anatomical studies" is not required the informed consent from died patients or their relatives for the analysis of the brains and organs to establish the cause of death. The study was carried out on material in the amount necessary to make a post-mortem diagnosis and nothing additional was taken from the corpse. All obtained samples were fixed and stored in a 10% formalin solution for prolonged periods. The human studies were performed on 3 male patients (average age 42) died from IVH (parenchymatous-ventricular hemorrhage in the right cerebral hemisphere with formation of subdural and intracerebral hematomas with blood rupture into ventricles and subarachnoid space).

### Intraventricular hemorrhage model
To produce a mouse IVH model, autologous blood (10 µL) was injected into the right lateral ventricle (AP = −0.5 mm; ML = −1.06 mm; DV = 2.5 mm) in adult mice. To produce the neonatal IVH model, we used the method of IVH in newborn rats[89] and injected autologous blood (5 µL) into the right lateral ventricle (AP = −1.0 mm; ML = −0.9 mm; DV = 1.2 mm). The blood was taken from the tail vein of the same mouse and collected in a sterile eppendorf pre-flushed with heparin to avoid coagulation during blood sampling and injection. Aseptic techniques were used in all surgical procedures. The disinfection with Betadine and 70% ethanol of the stereotactic apparatus and surgical tools were made prior to surgery. Throughout surgery and the experimental period, rectal temperature was monitored until the mouse completely recovered and displayed normal motor activity. The ketamine (100 mg/kg) and xylazine (10 mg/kg) was injected intraperitoneally for the anesthesia. The mouse was placed onto a thermal blanket and the scalp was shaved. The ophthalmic ointment to both eyes was applied. A 1 cm long midline incision of the scalp with a 10-scalpel blade was made. The Hamilton syringe (25 µL) was mounted onto the injection pump, and the needle (25 Gauge) over bregma was directed stereotaxically. Next, the needle was positioned according to the coordinates with the stereotactic manipulator. A small cranial burr hole was drilled through the skull using a variable speed drill with a 1 mm drill bit. The animal's tail was disinfected with 70% ethanol and the central tail vein was punctured with a sterile needle (25 Gauge).

After that, the arterial blood was collected into a sterile eppendorf. The blood was quickly transferred from eppendorf into the glass barrel of the Hamilton syringe and inserted into the plunger. The needle was inserted into the right lateral ventricle. The arterial blood was injected at a rate of 0.1 μL/min. The needle was left in the ventricle for 10 min and then removed at a rate of 1 mm/min to prevent the reflux of blood. The burr hole and scalp incision were closed with bone wax (Ethicon, Somerville, NJ) and with cyanoacrylate glue (Henkel Consumer Adhesive Inc. Scottsdale, Arizona), respectively. Sham control mice were injected with an equal volume of saline.

## Laser radiation scheme and dose calculation

A fiber Bragg grating wavelength locked high-power laser diode (LD-1267-FBG-350, Innolume, Dortmund, Germany) emitting at 1267 nm was used as a source of irradiation. The laser diode was pigtailed with a single-mode distal fiber ended by the collimation optics to provide a 5 mm beam diameter at the specimen. The mice with shaved heads were fixed in a stereotaxic frame under inhalation anesthesia (1% isoflurane at 1 L/min $N_2O/O_2$−70/30 ratio) and irradiated in the area of the Sagittal sinus using a single laser dose (3-6-9-18-27 J/cm$^2$) or the PS course 63 J/cm$^2$ during 7 days with the sequence of 17 min−irradiation, 5 min−pause, 61 min in total). For the PS course, the mice were treated daily by PS for 7 days under inhalation anesthesia (1% isoflurane at 1 L/min $N_2O/O_2$−70:30) 3 days after the surgery procedure of blood injection into the right lateral ventricle (Supplementary Fig. 1).

For adult animals, the dose of 9 J/cm$^2$ at the skull surface and 3 J/cm$^2$ on the surface of the cortex was applied during a 51-min procedure (3 times 17-min irradiation separated with 5-min pauses). Respectively, the total PS dose for 7 days was 63 J/cm$^2$. To ensure the same irradiation dose at the brain surface for the newborn rats, the laser radiation dose at the skull surface was reduced by 2.3 times to compensate for 2.3 times higher optical transmission of the newborn's skull (0.8 vs. 0.35 for adults). Thus, the PS doses of 9 J/cm$^2$ at the adult skull surface and 4 J/cm$^2$ at the skull surface of newborn rats result in nearly the same dose of 3 J/cm$^2$ at the brain surface of both adults and newborn animals.

## Measurement of the PS' thermal impact

A type A-K3 thermocouple (Ellab, Hillerød, Denmark) was used to measure skull temperature. The thermocouple was placed subcutaneously 2 mm lateral to the bregma in the irradiated zone. A burr hole was drilled under inhalation anesthesia (1% isoflurane at 1 L/min $N_2O/O_2$−70:30). To measure the brain surface temperature under the 1267 nm laser irradiation, the medial part of the left temporal muscle was detached from the skull bone, a small burr hole was drilled into the temporal bone, and a flexible thermocouple probe (IT-23, 0.23 mm diam, Physitemp Instruments LLC, NJ, USA) was introduced between the parietal bone and brain into the epidural space. Brain surface temperature was measured before and during the laser stimulation in 5 min increments using a handheld thermometer (BAT-7001H, Physitemp Instruments LLC, NJ, USA).

## Immunohistochemistry (IHC) and confocal imaging

To visualize LVs, fluorescent markers were used to label specific structures using the immunohistochemical method[46]. Anti-LYVE-1 and anti-PROX-1 antibodies were used to label LVs; an anti-CD-31 antibody was used to label blood vessels.

At the end of each experiment, animals were immediately sacrificed by cervical dislocation or by decapitation within seconds under deep anesthesia with isoflurane and using the guillotine (Stoelting Co, Wheat Lane, USA) and the DecapiCone® (Braintree Scientific Inc., Braintree, USA). To collect the meninges, the skin was removed from the head, and the muscles were stripped from the bone. After removing the mandibles and the skull rostral to maxillae, the top of the skull was removed with surgical scissors. Whole-mount meninges were fixed while still attached to the skull cap in phosphate-buffered saline (PBS) with 4% paraformaldehyde (PFA) overnight at 4 °C. The meninges were then dissected from the skull.

For analysis of dcLNs, the lymph nodes were removed and fixed in PBS with 4% PFA overnight at 4 °C, and then fixed in 2% agarose, followed by sliced into 60 μm-thick sections using a vibratome (Leica VT1000, Germany). The whole mounts of meninges and the sections of dcLNs were firstly washed 3 times (5 min for each) with wash solution (0.2% Triton-X-100 in PBS), secondly incubated in the blocking solution (a mixture of 2% Triton-X-100 and 5% normal goat serum in PBS) for 1 h, followed by incubation with rat Alexa Fluor 488-conjugated anti-LYVE-1 antibody (1:500; Cat. No. FAB2125G, R&D Systems, Minneapolis, Minnesota, USA), rabbit anti-PROX-1 antibody (1:500; Cat. No. ab 101851, Abcam, Cambridge, United Kingdom), rat Alexa Fluor 647-conjugated anti-CD-31 antibody (1:500; Cat. No.102416, BioLegend, San Diego, USA) and rabbit anti-LYVE-1 antibody (1:500; Cat. No. ab 218535, Abcam, Cambridge, United Kingdom) for rat dcLNs overnight at 4 °C in PBS containing 0.2% Triton-X-100 and 0.5% normal goat serum. Next, the meninges were incubated at room temperature for 1 h and then washed 3 times, followed by incubation with goat anti-rabbit IgG (H + L) Alexa Fluor 555 (1:500, Cat. No. A21429, Invitrogen, Molecular Probes, Eugene, Oregon, USA), and goat anti-rabbit IgG (H + L) Alexa Fluor 488 (1:500, Cat. No. A11008, Invitrogen, Molecular Probes, Eugene, Oregon, USA) to visualize LVs in rat dcLNs.

The sections of whole meninges from mice as well as approximately 10 slices of dcLN per adult animal and 5 slices for PD4 newborn rats were imaged using a confocal microscope (LSM 710, Zeiss, Jena, Germany) with a ×20 objective (0.8 NA) or a ×60 oil immersion objective (1.46 NA). Alexa Fluor 488 and Alexa Fluor 555 were excited with excitation wavelengths of 488 nm and 561 nm, respectively. Alexa Fluor 647 and RBCs were excited with the same excitation wavelength of 647 nm. Three-dimensional imaging data were collected by obtaining images from the x, y, and z-planes. The resulting images were analyzed with Imaris software (Bitplane).

For confocal visualization of MLVs from patients (average age 42) died after IVH (n = 3), the meninges (5 cm × 5 cm) from the region of the junction of the anterior sagittal and transverse sinuses were fixed with 4% PFA. After fixation, the brains were cryoprotected using 20% sucrose in PBS (10 mL/brain mouse) for 48 h at 4 °C. The brains were frozen in hexane and cooled to −32 - −36 °C. Cryosections (14 μm) of the parietal cortex were collected on poly-L-Lys, Polysine Slides (Menzel-Glaser, Germany) using cryotome (Thermo Scientific Microm HM 525, Germany) and a liquid for fixing a Tissue-Tek sample (Sakura Finetek, USA). Brain sections were processed according to the standard IHC protocol with the corresponding primary and secondary antibodies. Confocal microscopy of human meninges sections was performed using a confocal microscope (Nikon A1R MP, Nikon Instruments Inc.). The nonspecific activity was blocked by 2-h incubation at room temperature with 10% BSA in a solution of 0.2% Triton X−100 in PBS. Solubilization of cell membranes was carried out during 1-h incubation at room temperature in a solution of 1% Triton X−100 in PBS. Incubation with primary antibodies in a 1:500 dilution was performed overnight at 4 °C: with rabbit anti-LYVE-1 antibody (1:500; Cat. No. ab219556; Abcam, Biomedical Campus Cambridge, Cambridge, UK); mouse anti-CD-31 antibody (1:500; Cat. No. ab187377; Abcam, Biomedical Campus Cambridge, Cambridge, UK) and mouse anti-glycophorin A (GPA) antibody (1:500; Cat. No. ab7503; Abcam, Biomedical Campus Cambridge, Cambridge, UK). At all stages, the samples were washed 3−4 times with 5-min incubation in a washing solution. Afterward, the corresponding secondary antibodies were applied (goat anti-rabbit IgG (H + L) Alexa Four 488 (1:500, Cat. No. A11008) and goat anti-mouse IgG (H + L) Alexa Four 555 (1:500, Cat. No. A21422); Invitrogen, Molecular Samples, Eugene, Oregon, USA). At the final stage, the sections were transferred to the glass and 15 μL of mounting liquid (50% glycerin in PBS with DAPI at a concentration of

2 μg/mL) was applied to the section. The preparation was covered with a cover glass and confocal microscopy was performed.

Sections of human meninges were visualized using a confocal microscope (Nikon A1R MP, Nikon Instruments Inc.) with a ×20 lens (0.75 NA) or a ×100 lens for immersion in oil (1.4 NA). DAPI, Alexa Fluor 488 and Alexa Fluor 555 were excited with excitation wavelengths of 405 nm, 488 nm and 561 nm, respectively. Three-dimensional visualization data was collected by obtaining images in the x, y and z planes. The images were obtained using NIS-Elements software (Nikon Instruments Inc.) and analyzed using Fiji software (Open-source image processing software) and Vaa3D (Open Source visualization and analysis software).

## The analysis of the BBB permeability to FITCD

FITCD (1 mg/25 g rat, 0.5% solution in saline, Sigma-Aldrich) was injected intravenously via the tail vein and allowed to circulate in the blood during 30 min in rat from the following group: 3 and 11 days after injection of saline/blood as well as after the PS course in sham mice and with IVH. Afterward, rats were decapitated, their brains were quickly removed and fixed in 4% paraformaldehyde (PFA) for 24 h. For confocal imaging of rat brain slices, we used the protocol for the IHC analysis with the markers for endothelial cell adhesion by Claudin 5 (CLDN 5) and for astrocytes by glial fibrillary acidic protein (GFAP). Brain sections were processed according to the standard IHC protocol with the corresponding primary and secondary antibodies. The brain tissues were fixed for 48 h in a 4% saline solution-buffered formalin, then sections of the brain with a thickness of 40–50 μm were cut on a vibrotome (Leica Microsystems GmbH, Germany). Confocal microscopy of brain sections was performed using a confocal microscope (Nikon A1R MP, Nikon Instruments Inc.). The nonspecific activity was blocked by 2-h incubation at room temperature with 10% BSA in a solution of 0.2% Triton X-100 in PBS. Solubilization of cell membranes was carried out during 1-h incubation at room temperature in a solution of 1% Triton X-100 in PBS. Incubation with primary antibodies in a 1:500 dilution was performed overnight at 4 °C with rabbit anti-CLDN 5 antibody (1:500; Cat. No. ab217316; Abcam, Biomedical Campus Cambridge, Cambridge, UK), mouse anti-GFAP antibody (1:500; Cat. No. ab279290; Abcam, Biomedical Campus Cambridge, Cambridge, UK). At all stages, the samples were washed 3–4 times with 5-min incubation in a washing solution. Afterward, the corresponding secondary antibodies were applied (goat anti-rabbit IgG (H + L) Alexa Four 555 (1:500, Cat. No. A21429); goat anti-mouse IgG (H + L) Alexa Four 647 (1:500, Cat. No. A21235); Invitrogen, Molecular Samples, Eugene, Oregon, USA). At the final stage, the sections were transferred to the glass and 15 μL of mounting liquid (50% glycerin in PBS with DAPI at a concentration of 2 μg/mL) was applied to the section. The preparation was covered with a cover glass and confocal microscopy was performed.

Approximately 10 slices per animal from cortical and subcortical (excepting hypothalamus and choroid plexus where BBB is leaky) regions were imaged. The FITCD leakage in arbitrary units was evaluated by measuring changes in the fluorescence intensity of FITCD in the perivascular area in confocal images of the cortex taken 50 and 150 μm depth in 30 min after FITCD intravenous injection using Fiji software (Open-source image processing software).

## Measurement of the meningeal lymphatic vessel diameter

To measure the diameter of LVs, the original program with Matlab was developed (Supplementary Fig. 12).

Procedure 1

This procedure was used to extract the profile of LVs from the initial image. First, Otsu's method[90] was utilized to decide the threshold and obtain the binary image (Supplementary Fig. 12-Step 1). Next, an image-closing operation was used to connect the broken edges of the image. Then, two Matlab functions, "imfill" and "bwareaopen",

were used to fill in the holes and remove the small connected domains of the image, respectively (Supplementary Fig. 12-Step 2). Finally, the obtained image was subtracted by itself after morphological corrosion, and the profile curve was then determined (Supplementary Fig. 12-Step 3).

Procedure 2

This procedure was used to calculate the diameter distribution of lymphatic vessels. As shown in Supplementary Fig. 12-Step 4, point A and point B represents any points on the outlines on both sides of the lymphatic vessel, and $l_1$ and $l_2$ are the tangent lines at point A and B, respectively. If $l_1$, $l_2$ and $l_{AB}$ follow both $l_1 \perp l_{AB}$ and $l_2 \perp l_{AB}$, i.e.

$$k_{AB} = k_B k_{AB} = -1 \tag{1}$$

where $k_A$, $k_B$ and $k_{AB}$ present the slope of lines $l_1$, $l_2$ and $l_{AB}$, respectively. In this case, $l_{AB}$ could be taken as the diameter at a certain position. However, points A and B were not always successfully found in all the images. Therefore, for every point A, point B was given by

$$\min(|k_A k_{AB} + 1| + |k_B k_{AB} + 1|) \tag{2}$$

Following the above rule, we could obtain a series of |AB| as the lymphatic vascular diameters at every position.

## Fluorescent microscopy monitoring of EBD accumulation in dcLN

Adult mice were anesthetized with ketamine (100 mg/kg) and xylazine (10 mg/kg), and newborn rats were anesthetized with isoflurane (1% isoflurane at 1 L/min N$_2$O/O$_2$−70/30 ratio) and fixed in a stereotactic apparatus, the skull exposed, and a small burr hole was made over the right lateral (AP = -0.5 mm; ML = −1.06 mm; DV = 2.5 mm for adult mice and AP = −1.0 mm; ML = −0.9 mm; DV = 1.2 mm for newborn rats). Afterward, 5 μL of 5% EBD (Sigma-Aldrich) was injected (0.5 mL/min) into the right later ventricle. 20 min later, the ventral skin of the neck was cut and dcLNs were exposed. The stereo fluorescence microscope (Axio Zoom. V16, Zeiss, Jena, Germany) working at 10× magnification was used for imaging of dcLNs for 60 min before and after PS (3-6-9-18-27 J/cm$^2$). After imaging, the fluorescence intensity of EBD in dcLNs (arb. units) was measured using FIJI software.

## OCT monitoring of GNRs accumulation in dcLN

The GNRs coated with thiolated polyethylene glycol (0.2 μL, the average diameter and length at 16 ± 3 nm and 92 ± 17 nm) were injected into the right lateral ventricle (AP = −0.5 mm; ML = -1.06 mm; DV = 2.5 mm for adult mice and AP = −1.0 mm; ML = −0.9 mm; DV = 1.2 mm for newborn rats). Afterward, OCT imaging of the dcLNs was performed during the next 1 h.

In this study, a commercial spectral domain OCT Thorlabs GANYMEDE (central wavelength 930 nm, spectral band 150 nm) was used. The LSM02 objective was used to provide a lateral resolution of about 13 microns within the depth of the field. The a-scan rate of the OCT system was set to 30 kHz. Each B-scan consists of 2048 A-scans to ensure appropriate spatial sampling.

Since lymph is optically transparent in a broad range of wavelengths, "empty" cavities exist in the resulting OCT image of the lymphatic node with a background signal-to-noise ratio inside. In order to visualize the dynamic accumulation of lymph within these cavities, suspensions of GNRs were used as a contrast agent and the OCT signal intensity is proportional to the GNRs concentration. By tracking the OCT signal temporal intensity changes inside a node's cavity, we could confirm the clearance pathways and calculate its relative speed. The OCT recordings were performed under anesthesia with ketamine (100 mg/kg, i.p.) and xylazine (10 mg/kg, i.p.).

The GNRs content in dcLNs was evaluated by atomic absorption spectroscopy on a spectrophotometer (Thermo Scientific Inc.,

Waltham, Massachusetts, USA). The atomic absorption spectroscopy of GNRs was performed 20–40–60 min after the start of OCT monitoring in the brain and in the dcLNs obtained from the same mice, which were used for OCT-GNRs measurements.

## Determination of the size of hemorrhagic injury, PVS and ventricular area

On day 3 after IVH, ten adult mice and newborn rats in each group were anesthetized and perfused intracardially with 4% PFA in 0.1 mol/L phosphate-buffered saline (pH 7.4). Afterward, animals were sacrificed and the brains were removed, kept in 4% PFA for 24 h, and then the entire brain of each animal was cut in 3-mm-think blocks in the three projections: 1) AP − 0.74 mm; 2) AP − 1.8 mm; AP − 5.8 mm in adult mice and 1) AP − 0.9 mm; 2) AP − 5.5 mm; AP − 9.0 mm. Each block was photographed (Supplementary Fig. 6 and 9) and the area of the lateral ventricles was determined using image-analysis software (Fiji, open-source image processing software[91]). The total ventricular area of the lateral ventricles (mm$^2$) in the blocks was expressed as a composite value[92]. To evaluate the size of hemorrhagic injuries and PVS, the entire brain of each animal was cut into 5-μm-thick sections and prepared using the histological protocol with eosin and hematoxylin. The size of hematoma (mm$^2$) and PVS (μm) was calculated by the sum of the hematoma areas or perivascular edema of each session, respectively[93]. Sections were analyzed by an investigator blinded to the experimental cohort.

## Behavioral tests

Mice were housed in a temperature- and humidity-controlled room that was maintained on a 12-h light/dark cycle. All behavioral tests were conducted during the light cycle phase in an enclosed behavior room. All behavioral tests were evaluated and analyzed by an investigator blinded to the groups. The same animals were used for behavior testing, performed on day 11 post-IVH.

## Neurologic deficit score

Each mouse was scored for neurologic deficits using a modified protocol[94] based six parameters: body symmetry, gait, climbing, circling behavior, front limb symmetry, and compulsory circling. Each test was graded from 0 to 4, establishing a maximum deficit score of 24.

## Novel Object Recognition Test (NORT)

The NORT is used for the evaluation of recognition memory[95,96]. The mice were presented with two similar objects (blue cubes) during the first (familiarization) session. Then one of the two cubes was replaced by a novel object (pink ball) during the second (test) session. It has been demonstrated that prior experience can alter the behavioral responses of mice in the NORT[97], animals were accustomed to being handled by experimenters twice a week for 1 min each session for 1 week before the beginning of the experiments. Following the protocol of Sik et al.[98], the habituation phase consisted of 5 min exposures to the testing arena per day, separated by 6 h during 3 days before the test phase. The test was conducted 10 days after surgery. Two identical cubes were placed in a cage for mice for 10 min. Then there was a second 10 min session when one cube was replaced with an unfamiliar earlier ball[95]. The exploration time for both objects during the test phase was 20 s[99]. For the experiments, we used a black wooden box (33 cm × 33 cm × 20 cm) and a video-tracking package. We used two asymmetric objects of the same size (cubes: 3 cm × 3 cm × 3 cm; ball: 3 cm in diameter) and odor. Since mice have difficulty in discriminating colors, we selected bright (dark blue and pink) objects. The weights of objects were heavy enough that the mice could not move them. Additionally, we used Patafix to hold the objects stuck on the floor. The mice were housed in a light-dark cycle and were tested in the dark phase (active phase between 08:00 and 20:00). The familiarization

session was carried out in the morning (at 09:00). The mice were placed in the testing room 30 min before testing. The exploration was defined as follows: directing the nose toward the object at a distance of less than or equal to 2 cm. We chose to score the object exploration whenever the mouse sniffed the object or touched the object while looking at it (i.e., when the distance between the nose and the object was less than 2 cm). Climbing onto the object (unless the mouse sniffs the object it has climbed on) or chewing the object does not qualify as exploration.

## Wire-hanging test

The wire hang test evaluates motor abnormalities, including balance and prehensile strength. For this procedure, mice were placed onto a 10-cm square grid that was flipped upside down 50 cm over a cushioned surface, and the time to fall off the grid was measured[100,101].

## Tail suspension and forced swim tests

The protocol for the tail suspension test was described previously[102]. Briefly, animals were suspended by their tails at the edge of a shelf 55 cm above a desk. Sticky tape (17 cm long) was used to fix the tail (approximately 1 cm from the tip) to the shelf. The recording of mouse mobility and immobility (lack of escape-related behavior when mice hung passively and completely motionless) was made for 360 sec. The forced swim test protocol was published in detail in Ref. 103. The cylindrical tanks (20 cm high, 22 cm in diameter) with water at 24 ± 1 °C (10 cm) were used for this test. Each mouse was placed individually in water. The swimming of mice was recorded for 360 sec. The immobility time (when the mouse remained floating motionless, making only small movements to keep its head above the water) was calculated during the last 4 min from the 240 s of test time[104].

Rat pups are altricial when born, and therefore too immature to undertake specific or complex motor, sensory and/or cognitive behavioral tasks. In this regard, their developmental immaturity relates to both their physical and organ development. With reference to brain development, substantial cortical maturation occurs postnatally. Newborn rat pups (PD4) have been suggested to reach a brain maturation level that is similar to a preterm human brain of 23 - 28 weeks gestation[52–56,105]. This correlation is based on gross anatomical analyses, however, other measures of brain maturation such as myelination and amplitude integrated electroencephalograms have also been described[53–57,106]. For these reasons, we performed neurodevelopment reflexes testing in PD4 pups. The battery of reflexes are adapted from studies by W.M. Fox and A. Lubics[106,107]. These reflexes include fore-and hindlimb suspension, cliff avoidance, and gait. Timeline for assessment of neurodevelopmental reflexes is presented in Supplementary Fig. 9.

## Fore-and hindlimb suspension[108]

This suspension test determines fore- and hindlimb strength. Pups were allowed to grasp a wire strung across a stable object and hang onto the wire with both fore- or hindpaws. The testing area was over a padded drop zone. Afterward, pups were released. Using a timer, we recorded the total time to fall, as well as paw weakness. Paw weakness was determined if a pup consistently falls from the wire with one paw before the other rather than releasing from the wire with both paws at the same time. Rats falling immediately when released or failure to grasp when placed on the wire are indicative of non-participation. The test was repeated three times.

For hindlipms suspension test, using a 50 mL conical, pups were placed gently face down into the tube with its hind legs hung over the rim. Afterward, pups were released and observed the hindlimb posture. Score posture according to the following criteria. Score of 4 indicates normal hindlimb separation with tail raised; score of 3 means weakness is apparent and hindlimbs are closer together but they seldom touch each other; score of 2 indicates hindlimbs are close to each

other and often touching; score of 1 shows a weakness is apparent and the hindlimbs are almost always in a clasped position with the tail raised; a score of 0 indicates constant clasping of the hindlimbs with the tail lowered or failure to hold onto the tube for any period of time. Using a timer, the latency to fall was recorded. The test was repeated 3 times.

### Cliff Avoidance

The pup was placed with the digits only of their forepaws and their snout positioned over the edge. Scoring was performed by counting the total time it takes the pup to turn away from the cliff and move its paws and snout away from the edge. If the pup does not move away from cliff within 30 sec, no score is given. If the pup falls off the edge, a single additional trial was performed. The test was repeated for a total of 3 trials. Score the cliff avoidance: 0 - for no movement or falling off the edge; 1 - for attempts to move away from the cliff but with hanging limbs; 2 - for successful movement away from the cliff.

### Gait

The pups were placed in the center of a 15 cm diameter circle and allowed to complete the task during 30 s[105]. A successful gait was performed when the rat pup was able to move both forepaws outside the circle in less than 30 s. Score the gait was the time in seconds, which the rat pup took to move both forepaws outside the circle.

### Isolation of lymphatic endothelial cells (LyECs) and NO measurement in LyECs

Freshly isolated primary LyECs were obtained from the mesentery of intact mice. Briefly, the atrium was cannulated and the vascular system was perfused with a normal saline solution. Mesenteric lymphatic tissue mucosa was harvested, placed on 35 mm plates containing ice-cold phosphate-buffered saline, and cut into small (1 mm) fragments. The fragments were incubated in 0.25% collagenase A (Roche Diagnostics, Basel, Switzerland) at 378 C. The suspension was passed through 100 mm nylon mesh and centrifuged at 1800 rpm for 4 min at 48 °C. The cell pellet was resuspended in Hank's balanced salt solution. The LyECs were isolated using rabbit antibody to rat podoplanin (Sigma Chemical, St Louis, Missouri, USA) in a 1:100 dilution as the primary antibody and microbeads coupled with a secondary goat anti-rabbit antibody (MACS system, Miltenyi Biotec, Bergisch-Gladbach, Germany). The cells were grown in Dulbecco's modified Eagle medium that was supplemented with 20% fetal calf serum, 50 U/mL penicillin and 50 mg/mL streptomycin.

The level of nitrite/nitrate (NOx) produced by LyECs was determined using the CLD88 NO analyzer (Ecophysics), as previously described[109]. Quantification of NOx accumulation was obtained by comparison with external standards and normalized to protein concentration, determined by the bicinchoninic acid protein assay.

### Measurement of contractility of lymphatic vessels

Video sequences of LVs were captured using transmitted light Axio Imager A1 microscope with 10×0.2 Epiplan Lens (Zeiss, Germany) and monochrome CMOS camera acA1920-40um (Basler AG, Germany). Image sequences were captured with a resolution of 1920 × 1200 pixels, 8 bit, 40 fps and stored in AVI video format. To measure lymphatic vessel diameter, the video sequence was processed with homemade software developed in LabVIEW (National Istruments Inc., USA). Walls of LVs were detected in each frame of the sequence along a line drawn across the vessel image. The IMAQ Edge Tool 3 VI (NI Vision, National Istruments Inc., USA) was used to get the position of both edges of the vessel. The resulting series of measured distances were then filtered with 4-point median filter to exclude spikes caused by detection errors related to occasional vessel movements.

### Monitoring of FITC-dextran 70 kDa distribution in the brain

Three days before experiments, a polyethylene catheter (PE-10, 0.28 mm ID × 0.61 mm OD, Scientific Commodities Inc., Lake Havasu City, Arizona, United States) was implanted into the right lateral ventricle (AP = −1.0 mm; ML = −0.9 mm; DV = 1.2 mm) according to the protocol reported by Devos et al.[110]. A small cranial burr hole was drilled through the skull using a variable-speed dental drill (with a 1 mm drill bit). An amount of 5 µL of FITC-dextran 70 kDa (Sigma-Aldrich, United States), at a rate of 0.1 µL/min, was injected into the right lateral ventricle. The cerebral vessels were filled by Evans Blue dye injected into the tail vein (Sigma Chemical Co., St. Louis, Missouri, 2 mg/ body weight, 1% solution in physiological 0.9% saline).

The ex vivo optical study of FITC-dextran distribution was performed 3 h after the intraventricular injection of tracer in PD4 rats. The imaging was performed using a confocal microscope (Nikon A1R MP, Nikon Instruments Inc.) with a ×20 objective (0.75 NA). Two lasers (488 nm and 561 nm, respectively) were used for the excitation of FITC dextran and Evans Blue fluorescent dyes, respectively, during the confocal imaging.

The Petri dishes, where samples were submerged in a buffer solution, were placed on a motorized stage below the objective of A1R MP. The top surface of each sample was covered with a 25 mm × 50 mm × 0.17 mm cover glass.

The 12-bit grayscale confocal images were acquired in a dark room. To cover larger regions of interest in the samples, each acquired confocal image was stitched out of smaller 512 px by 512 px confocal images with 15% overlap, as the field of view of the ×20 objective was restricted only to 0.5 mm by 0.5 mm area.

The images were obtained using NIS-Elements software (Nikon Instruments Inc.) and analyzed using Fiji software (Open-source image processing software). Image processing procedures were identical for each pair of images (control and laser-treated samples) for each channel to ensure an accurate comparison of the fluorescence intensity.

The quantitative analysis of tracer in the brain slices was carried out on a fluorescence microscopic system described above. For a quantitative analysis of the intensity signal from FITC-dextran, ImageJ was used for image data processing and analysis. The intensity of fluorescence for each slide was integrated over a rectangular region of interest, bounding the brain slice. The integral value was divided by slice area. The areas of brain slices were calculated using the plugin "Analyze Particles" in the "Analyze" tab, which calculates the total area of tracer fluorescence intensity tissue elements—the indicator "Total Area". In all cases, 10 regions of interest were analyzed.

### Photo-damages of MLVs

To ablate the MLVs, visudyne treatment was carried out according to a previous publication[31]. Briefly, mice from the visudyne+laser group were anesthetized with ketamine hydrochloride and visudyne (APExBIO, Cat. No. A8327, 5 µL) was injected into the cisterna magna at a speed of 1 µL/min. Fifteen minutes later, with a 689-nm wave-length laser (Changchun Laser Technology, a dose of 50 J/cm²) was applied through the skull in different places, including the cisterna magna, the left and right transverse sinuses, the superior sagittal sinus, and the junction of all sinuses. For the control groups (sham and visudyne only), mice were injected (5 µL, into the cisterna magna) by physiological saline or visudyne, respectively; the laser (control) group included mice treated by laser (689 nm) irradiation without visudyne. The eyes of mice were protected during photoablation of the MLVs. In all groups, 7 days after photoablation of the MLVs, blood (10 µL) was injected into the right lateral ventricle 1 h before removal of the meninges and confocal microscopy.

## Hematoxylin & Eosin (H&E) staining

To analyze the morphological changes in brain tissue, H&E staining was performed. The entire brain of each animal was fixed in 4% neutral paraformaldehyde for 24 h and embedded in paraffin. After that, the samples were cut into 5-μm-thick sections and stained with hematoxylin & eosin. Finally, the sections were magnified and scanned with white light.

## Laser speckle contrast imaging

Mice were anesthetized by isoflurane, an incision was done along the midline to separate the skin of the skull, and laser speckle contrast imaging (RWD Life Science Co., Ltd) was used to detect mice cerebral blood flow. Laser speckle blood flow images were recorded and used to identify the regions of interest (ROIs). Within these ROIs, the mean blood flow index was calculated in real time.

## L-NAME treatment

To block the release of NO, L-NAME (the blocker of NOS) treatment was carried out according to previous publications[111,112]. Briefly, mice were anesthetized with ketamine hydrochloride and their heads were fixed in a stereotactic instrument. L-NAME (Sigma, Cat. No. N5751) was reconstituted according to the manufacturer's instructions, and 5 μL (100 mg/mL) was injected into the cisterna magna at a speed of 1 μL/min. Four hours later, the mice were used to produce the IVH model.

## Statistical analysis

Statistical analysis was performed using the SPSS software. Data are presented as mean ± SEM. The Shapiro–Wilk test, a method for small sample sizes, was used to assess the normality of data distribution in each experiment. The heterogeneity of variance was evaluated using the Levene test, a stable method for both normally and nonnormally distributed data. The significance of differences between means was evaluated by unpaired Student's $t$ test (normality distribution, variance homogeneity), Welch's test (normality distribution, variance non-homogeneity) or non-parametric tests (Mann–Whitney–Wilcoxon test, non-normality distribution) for two independent group comparisons, and ANOVA with Turkey's multiple-comparison test (variance homogeneity) or Dunnett's T3 multiple-comparison test (variance non-homogeneity) was used for comparisons of more than two groups. In this study, $P < 0.05$ was considered significant (*$P < 0.05$, **$P < 0.01$, and ***$P < 0.001$).

## Reporting summary

Further information on research design is available in the Nature Portfolio Reporting Summary linked to this article.

## Data availability

The data supporting the findings of this study are available within the article, the Supplementary Information files and the Source Data files that accompany this article. Source data are provided with this paper.

## Code availability

The code used to analyze these data is freely available on Github [https://github.com/HUST-LAB/MLV-diameter].

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

## Acknowledgements

D.L. and S.L. contributed equally to this work. This work was supported by the National Natural Science Foundation of China (NSFC) (Grant Nos. 61860206009, 62375096, 82372012, 62375095, 82001877); Key Research and Development Project of Hubei Province (No. 2022BCA023); the Innovation Fund of WNLO; RF Governmental Grant No. 075-15-2022-1094; Russian Science Foundation No. 23-75-30001. D.B. was supported by NIH R01 NS112808. The authors also thank the Optical Bioimaging Core Facility of WNLO-HUST for support in data acquisition. We thank the research center "Symbiosis" and immu-nochemistry laboratory IBPPM RAS for their support with immuno-fluorescence analysis and confocal microscopy within Project No. GR 121031100266-3.

## Author contributions

D.L. was involved in the conceptualization, experiment setup, investi-gation, statistical analysis, writing and editing. S.L. was involved in the conceptualization, experiment setup and investigation. Z.L. was involved in statistical analysis. S.S. was involved in experimental setup and investigation. T.Y. was involved in conceptualization and writing. O.B. performed measuring of thermal effect of photostimulation. A.S. performed experiments on the LyECs. I.F. was involved in the mea-surement of the contractibility of LVs. N.N. and A.K. collected the human brain and performed ICH. Z.H. and A.D. performed the analysis of CBF. D.B., I.B., M.T., and A.T. performed the experiments with newborn rats. J.K. was involved in writing and editing. O.S. and D.Z. were involved in writing, editing, conceptualization and project management.

## Competing interests

The authors declare no competing interests.
