## [Peer Review File · Nature Communications]

REVIEWER COMMENTS

Reviewer #1 (Remarks to the Author):

Manuscript# NCOMMS-20-39996A-Z with Title: "Noninvasive technology of photostimulation of lymphatic clearance of red blood cells from the mouse brain after intraventricular hemorrhage" addresses the use of low-level laser photobiomodulation as a method to control Intraventricular hemorrhage (IVH) in the brain of rats and consequently, the level of cerebral edema and intracranial hypertension.

The research group is already studying the topic with a well-structured experimental model and used the established method to observe the effect of photobiomodulation, the most current name for photostimulation used by the authors, as a tool for non-invasive and easy-to-perform treatment for ventricular hemorrhage intervention, a clinical entity recognized as difficult to address and with a low rate of positive results when it appears in the clinical setting.

The work is convincing by the results presented, whose technology is really innovative for this type of application, but I believe that the conclusions should be adjusted, as there was only one analysis in experimental animals and there is no data related to its use in human beings, as suggested by the authors. Therefore, I believe it would be important to correct this detail so that it becomes consistent with the data presented.

I imagine that the article can actually influence other articles that may be published as a result of these initial data and become a therapeutic tool for this problem, which has a certain prevalence in the population.

The methods and statistical analyzes were carried out with competence, as well as the writing of the manuscript were well-matched, with small errors in the repetition of sentences, but without affecting the understanding and quality of the text.

From my point of view, the manuscript is capable of being published, if minor grammatical errors are corrected, and conclusions adjusted.

Reviewer #2 (Remarks to the Author):

The authors examined the effects of transcranial low-level infra-red photostimulation on clearance of red blood cells in a mouse model of intraventricular hemorrhage (IVH). IVH was induced by intraventricular injection of 10- μ l blood. The main weakness of this study is the model itself. The mortality rate should be extremely low after 10- μ l blood injection in mice. However, the authors

reported a mortality rate of 36.7%. The causes of death were unknown. If severe brain damage occurred in this IVH model, brain histology should be provided.

Reviewer #3 (Remarks to the Author):

In the manuscript by Li et al, the authors describe how photostimulation therapy is improving drainage of red blood cells by the meningeal lymphatic therefore facilitating recovery after subarachnoid hemorrhage. While the data presented in the manuscript is very interesting, some of the panels do not provide sufficient evidence or are misleading to really validate the observation.

Major points:

- Some of the nomenclature used by the authors does not match the data that is being presented which renders interpretation difficult. In Figure 2, the authors claim to stain the subarachnoid space. However, the stainings are performed on the meninges and not the subarachnoid space. Figure 1 and 2 could be consolidated into one figure as the data validate a recently published study (Chen et al, Nat Comm, 2019). In Figure 5, the authors write that they measured lymphatic diameter of the lymphatics of the skull base. However, the caption depicts the dorsal lymphatics adjacent to the confluence of the sinuses that are very different from the basal lymphatics.
- The authors select the dose of 9 J/cm³ as it significantly affects drainage without increasing skull temperature. Yet the authors discarded the potential direct effect of the PS to the brain because of lack of temperature change on the cortex. The author should provide some analysis of the brain at the different temperature to discard any temperature-independent effect PS may have on the brain.
- The author hypothesizes that the effect of PS on drainage is mediated by increase NO production by contractile lymphatics. However, valves are only present in the very basal region of the skull. With the PS being localized in the dorsal region of the skull, how do the author reconcile this discrepancy? Does PS have effects on initial lymphatics independent of NO production? Does it affect CSF dynamic independently of the lymphatics?
- It would be of great value if the author could demonstrate that the beneficial effect of PS is indeed mediated by the increase meningeal lymphatic drainage. A combination of PS with a method to ablate the meningeal lymphatics would greatly improve the manuscript and the claims made by the author.
- Recent studies have demonstrated that the skull bone marrow represents a major source of immune cells for the meningeal compartment. Do the author know how PS affects the skull bone marrow? May it induce increase of RBC in the meninges independently of IVH?

Minor points:

- The human data does not convincingly demonstrate the presence of RBCs in the meningeal lymphatics. Why are the rendering different in the control and IVH group? Better more annotated images would be required to validate the presence of RBC in human meninges
- In the survival graph, are the IVH and IVH+PS group significantly different? The authors should indicate for each graph what statistical test was used in order to provide clarity.
- This reviewer feels that the use of "emotional state" is anthropomorphic and misleading. It would be better if the authors just labelled it mouse behavior to not create meaning to these tasks.

Reviewer #4 (Remarks to the Author):

This is an interesting and potentially clinically relevant studies pertaining to intraventricular hemorrhage (IVH) cleanup, showing that the non-invasive transcranial low level infra-red (1267 nm) photostimulation help in RBC clearance from the cerebral ventricles, which coincides with lowering intracranial pressure (ICP) and reduced emotional distress. The overall hypothesis is that photostimulation (PS) by the mean of affecting brain lymphatic vessels and its draining systems, through (nitric oxide) NO involving mechanism, helps improving post IVH outcome. Overall paper is interesting and well written.

Nitric oxide (NO) is normally effectively sequestered by hemoglobin. Thus, in presence of high levels of local hemoglobin a byproduct of IVH, it is counterintuitive that NO may generate such strong effect on relaxation of lymphatic vasculature. Please consider discussing this issue. Also, the in vitro evidence suggesting increase in NO production by PS is indirect and weak. Additional in vivo data confirming the role of NO is needed.

The behavioral assays used by the investigators probe general sickness level and have limited specificity to measure IVH outcome. Broader range of behavioral outcome assessment should be used.

Presence of heparin in the blood used to generate SAH could influence interpretation of the outcome. E.g. heparinized blood used to produce experimental intracerebral hemorrhage dramatically inhibits edema formation as compared to non-heparinized blood. Could the heparin-inhibited RBC aggregation be essential for allowing the PS-facilitated cleanup process?

For decades and through hundreds of research papers, PS has been shown to produce highly pleiotropic biological effects (other than lymphatic engagement) with many of them having high relevance to pathobiology of IVH – both systemic and cellular. Thus, a stronger direct evidence for causal relationship between lymphatic system and PS in post IVH outcome (to exclude other beneficial effect of PS), could be highly beneficial.

Certainly, as mentioned by the Authors, photostimulation in humans may not be feasible due to scattering/extinction of trans-cranially applied energy. However, the method could be potentially useful

for newborns, as rightfully pointed by the Authors. As such it is unclear why experiment with model of IVH in newborn animals was not tested.

The Authors demonstrate the increased number of RBC in dLN only at one hour post ICV RBC injection. To understand the dynamic of the PS effect a timecourse (including RBC count in dLN at both earlier and later timepoint) could be useful. Also, to excluded possibility that the influx of RBC is increased by photostimulation it needs to be determined that the RBC efflux from the nodes is not influenced by the photostimulation.

Fig 1. Quantitative analysis for RBC in dLNs should be provided.

Are RBC present in lymph nodes free or insight of phagocytes? Lyve-1 is known to recognize some of the subtypes of macrophages. The images provided in their work do not exclude possibility that RBC are internalized by macrophages and as such are in the lymphatic system. Also, on representative images from panels 2 and 3, please include nuclear stain (e.g. DAPI) to show what is the overall population of cells on the micrograph.

Is the presence of RBC/hemoglobin remaining in the ventricles/CSF reduced significantly with PS? A total number of RBC cleared by the lymphatic system could be small and not essential in the overall blood clearance process. If so, it could mean that PS works through some other process.

Fig 5 legend – correct IHV to IVH.

Please indicate (especially for experiments described in Fig 4) when after IVH photostimulation was initiated. Also, regarding therapeutic implications, is the therapeutic window used in this study clinically relevant?

Fig 4. When after the IVH onset behavioral tests were conducted? Because of the high mortality in IVH groups how many animals were ultimately subjected to the behavioral evaluation. Was mortality and behavior evaluated in separate cohort of animals? Finally, line 190 – the authors indicate that the mortality was “1.57-fold lesser” in the treated group. Based on Fig 4b there is no statistical difference indicated for IVH vs. IVH-PS groups. Please clarified use of “lesser” and provide p value.

Some histological analyses comparing IVH vs. IVH+PS needs to be performed to further validate conclusions of the functional (behavioral) findings. Eg., hydrocephalous is one of the consequences of IVH. Was there any difference detected between ventricular sizes in animals from various treatment groups?

Response to Reviewers

Reviewer 1.

Manuscript# NCOMMS-20-39996A-Z with Title: "Noninvasive technology of photostimulation of lymphatic clearance of red blood cells from the mouse brain after intraventricular hemorrhage" addresses the use of low-level laser photobiomodulation as a method to control Intraventricular hemorrhage (IVH) in the brain of rats and consequently, the level of cerebral edema and intracranial hypertension.

The research group is already studying the topic with a well-structured experimental model and used the established method to observe the effect of photobiomodulation, the most current name for photostimulation used by the authors, as a tool for non-invasive and easy-to-perform treatment for ventricular hemorrhage intervention, a clinical entity recognized as difficult to address and with a low rate of positive results when it appears in the clinical setting.

I imagine that the article can actually influence other articles that may be published as a result of these initial data and become a therapeutic tool for this problem, which has a certain prevalence in the population.

The methods and statistical analyzes were carried out with competence, as well as the writing of the manuscript were well-matched, with small errors in the repetition of sentences, but without affecting the understanding and quality of the text.

From my point of view, the manuscript is capable of being published, if minor grammatical errors are corrected, and conclusions adjusted.

Comment: The work is convincing by the results presented, whose technology is really innovative for this type of application, but I believe that the conclusions should be adjusted, as there was only one analysis in experimental animals and there is no data related to its use in human beings, as suggested by the authors. Therefore, I believe it would be important to correct this detail so that it becomes consistent with the data presented.

Response: We would like to thank the reviewer for the constructive advice and for the positive assessment of our research. The conclusion has been modified in accordance with your recommendation (Page 26, Lines 561-574).

Reviewer 2.

Comment: The authors examined the effects of transcranial low-level infra-red photostimulation on clearance of red blood cells in a mouse model of intraventricular hemorrhage (IVH). IVH was induced by intraventricular injection of 10- μ l blood. The main weakness of this study is the model itself. The mortality rate should be extremely low after 10- μ l blood injection in mice. However, the authors reported a mortality rate of 36.7%. The causes of death were unknown. If severe brain damage occurred in this IVH model, brain histology should be provided.

Response: The authors thank the referee for the important remark. We used a modified model of IVH (1) and injected heparinized blood into the right lateral ventricle that was associated with penetration of blood into the brain tissues and with vasogenic edema 3 days after IVH suggesting an increase permeability of the cerebral blood vessels. We added additional histological data presented in Figures 3 d,e,h-n. The role of heparin in the permeability of the cerebral blood vessel is not fully understood (2-4). There is evidence that heparin increases the vascular permeability to the blood (2,3). Hemoglobin and heme are potent cytotoxic chemicals capable of causing death to many brain cells (5,6). Prominently, the mechanism of hemoglobin toxicity is via generating free radicals (mainly through the Fenton-type mechanism) and massive oxidative damage to proteins, nucleic acids, carbohydrates and lipids (5-9). This can explain the high mortality rate in mice with IVH.

We added this explanation to the manuscript (Page 8, Lines 201-209). References:

1. Belayev L, Saul I, Curbelo K, Busto R, Belayev A, Zhang Y, Riyamongkol P, Zhao W, Ginsberg MD. Experimental intracerebral hemorrhage in the mouse: histological, behavioral, and hemodynamic characterization of a double-injection model. *Stroke*. 2003 Sep;34(9):2221-7. doi: 10.1161/01.STR.0000088061.06656.1E.
2. Oschatz C, Maas C, Lecher B, Jansen T, Björkqvist J, Tradler T, Sedlmeier R, Burfeind P, Cichon S, Hammerschmidt S, Müller-Esterl W, Wuillemin WA, Nilsson G, Renné T. Mast cells increase vascular permeability by heparin-initiated bradykinin formation in vivo. *Immunity*. 2011 Feb 25;34(2):258-68. doi: 10.1016/j.immuni.2011.02.008.
3. Bentzer, P., Fisher, J., Kong, H.J. et al. Heparin-binding protein is important for vascular leak in sepsis. *ICMx* 4, 33 (2016). <https://doi.org/10.1186/s40635-016-0104-3>
4. Li X, Zhu J, Liu K, Hu Y, Huang K, Pan S. Heparin ameliorates cerebral edema and improves outcomes following status epilepticus by protecting endothelial glycocalyx in mice. *Exp Neurol*. 2020 Aug;330:113320. doi: 10.1016/j.expneurol.2020.113320.
5. Aronowski J, Zhao X. Molecular pathophysiology of cerebral hemorrhage: secondary brain injury. *Stroke*. 2011;42(6):1781-1786. doi:10.1161/STROKEAHA.110.596718
6. Wagner KR, Sharp FR, Ardizzone TD, Lu A, Clark JF. Heme and iron metabolism: Role in cerebral hemorrhage. *J Cereb Blood Flow Metab*. 2003;23:629–652.
7. Wu J, Hua Y, Keep RF, Nakamura T, Hoff JT, Xi G. Iron and iron-handling proteins in the brain after intracerebral hemorrhage. *Stroke*. 2003;34:2964–2969.
8. Wagner KR, Packard BA, Hall CL, Smulian AG, Linke MJ, De Courten-Myers GM, et al. Protein oxidation and heme oxygenase-1 induction in porcine white matter following intracerebral infusions of whole blood or plasma. *Dev Neurosci*. 2002;24:154–160.
9. Nakamura T, Keep RF, Hua Y, Hoff JT, Xi G. Oxidative DNA injury after experimental intracerebral hemorrhage. *Brain Res*. 2005;1039:30–36.

Reviewer 3.

In the manuscript by Li et al, the authors describe how photostimulation therapy is improving drainage of red blood cells by the meningeal lymphatic therefore facilitating recovery after subarachnoid hemorrhage. While the data presented in the manuscript is very interesting, some of the panels do not provide sufficient evidence or are misleading to really validate the observation.

Comment: Some of the nomenclature used by the authors does not match the data that is being presented which renders interpretation difficult. In Figure 2, the authors claim to stain the subarachnoid space. However, the staining are performed on the meninges and not the subarachnoid space. Figure 1 and 2 could be consolidated into one figure as the data validate a recently published study (Chen et al, Nat Comm, 2019).

Response: We improved the nomenclature and combined Figures 1 and 2 into one.

Comment: In Figure 5, the authors write that they measured lymphatic diameter of the lymphatics of the skull base. However, the caption depicts the dorsal lymphatics adjacent to the confluence of the sinuses that are very different from the basal lymphatics.

Response: We thank you for this advice. In accordance with the publication (Nature. 2019, 572(7767): 62-66. doi: 10.1038/s41586-019-1419-5), we conducted additional studies of the photo-effects on changes in the diameter of the basal meningeal lymphatic vessels presented in Figure 5.

Comment: The authors select the dose of 9 J/cm² as its significantly affects drainage without increasing skull temperature. Yet the authors discarded the potential direct effect of the PS to the brain because of lack of temperature change on the cortex. The author should provide some analysis of the brain at the different temperature to discard any temperature-independent effect PS may have on the brain.

Response: We measured the temperature on the skull (scalp removed) and the top of the cortex using different laser doses of 3-6-9-18-27 J/cm² (Page 5, Lines 125-138). The application of these small PS doses was not accompanied by any changes in the cortex surface temperature (Table S1). We also performed a histological analysis of the brain tissues before and after different the PS-course (3-6-9-18-27 J/cm²) presented in Figure S2. There were no changes in the brain morphological structures and in the cerebral vessels after PS-course with different PS doses that allows us to conclude that low level infra-red laser radiation cannot affect the brain tissues and functions.

Our results on healthy mice demonstrate that PS doses 3 J/cm² and 6 J/cm² were not sufficient for PS-stimulation of lymphatic clearance of GRNs and EBD. In contrast, PS doses 9-18-27 J/cm² caused similar effects on GNRs and EBD accumulation in dcLNs. However, PS doses 18 J/cm² and 27 J/cm² were accompanied by an increase in the skull temperature that can induce the heating effects on red blood cells [DOI:10.1039/b403127j]. Therefore, PS 9 J/cm² on the skull (3 J/cm² on the brain surface, Fig. S4) has been selected as an optimal for further experiments. Figure S5 demonstrates that PS in single dose 9 J/cm² did not change the cerebral blood flow (CBF) on macro- and microcirculation suggesting the direct PS effects on LVs (Page 6, Lines 167-169).

Brain tissue temperature elevation induced with infra-red laser radiation decreases exponentially with depth because of the exponential attenuation of the laser radiation power density related with the light scattering and absorption in the tissue [*Photochem. Photobiol. Sci.*, 2004,3, 981-989. doi: 10.1117/1.JBO.19.1.015009] and [*D. Galiakhmetova, et al. (2022) "Evaluation and modelling penetration depth of*

near-infrared irradiation generated by tunable ultra-short pulsed laser in ex vivo samples of mouse head”, SPIE Europe, Strasbourg, France, Paper 12147-10].

Therefore, we can assume that temperature elevation at brain depth is negligibly with respect to that measured at the cortex. We should note that direct measurements of laser induced temperature elevation in tissue is challenging, since a metal thermocouple absorbs more light radiation than the surrounding tissues and therefore the measured temperature is typically overestimated. Thus, the temperature elevation measured with a thermocouple at the brain cortex within illuminated area is actually the maximal possible temperature elevation induced with laser radiation over brain tissues at any point and at any depth.

Comment: The author hypothesize that the effect of PS on drainage is mediated by increase NO production by contractile lymphatics. However, valves are only present in the very basal region of the skull. With the PS being localized in the dorsal region of the skull, how do the author reconcile this discrepancies? Does PS have effects on initial lymphatics independent of NO production? Does it affects CSF dynamic independently of the lymphatics?

It would be of great value if the author could demonstrate that the beneficial effect of PS is indeed mediated by the increase meningeal lymphatic drainage. A combination of PS with a method to ablate the meningeal lymphatics would greatly improve the manuscript and the claims made by the author.

Response: The PS area was 5 mm, which is covered both the basal and dorsal parts of mouse brain and the brain of newborn rats.

We performed the additional experiments with the ablation of MLVs and discuss the new results in our manuscript (Pages 15-17, Lines 352-376; Page 23, Lines 479-494).

In experiments with photodynamic ablation of MLVs, we clearly show that the injuries of MLVs significantly reduces the lymphatic clearance of RBCs indicating that MLVs are an important pathway of the RBCs evacuation from the brain. However, photodynamic impairment of MLVs does not completely block of the RBCs removing from the brain, as evidenced by the presence of small amount of RCBs in dcLNs. This fact suggests existence alternative pathways of the RBCs removing from the brain, probably through the CSF drainage via the cranial nerves and the cribriform plate [*Nanophotonics*, vol. 10, no. 12, 2021, pp. 3215-3227. <https://doi.org/10.1515/nanoph-2021-0212>; *Fluids Barriers CNS*. 2022 Feb 3;19(1):9. doi: 10.1186/s12987-021-00282-z; *Fluids Barriers CNS* 11, 26 (2014); doi: 10.1186/2045-8118-11-26; *Cerebrospinal Fluid Res.* 2005;2:6. Published 2005 Sep 20. doi:10.1186/1743-8454-2-6; *Int. J. Mol. Sci.* 2022, 23(6), 2975; <https://doi.org/10.3390/ijms23062975>].

We discuss in the mechanism of PS-mediated improvement of recovery after IVH independent of NO production (Page 25, Lines 543-559).

Comment: Recent studies are demonstrated that the skull bone marrow represent a major source of immune cells for the meningeal compartment. Do the author know how PS affects the skull bone marrow? May it induce increase of RBC in the meninges independently of IVH?

Response: There are intrigue results presented in *Science*. 2021 Jul 23; 373(6553): eabf7844. doi: 10.1126/science.abf7844. There are findings demonstrating the therapeutic value of PS application on the bone marrow (BM), including PS-mediated stimulation of proliferation of bone marrow stem cells (*Lasers Surg Med*, 20, 56-63. [http://dx.doi.org/10.1002/\(SICI\)1096-9101\(1997\)20](http://dx.doi.org/10.1002/(SICI)1096-9101(1997)20)) and PS-BM-related a significant reduction in scarring, enhanced angiogenesis and functional improvement in pigs after myocardial infarction (*Photomedicine and Laser Surgery* 34(11). DOI:10.1089/pho.2015.3988). We did not find the results suggesting the PS effects on the skull bone marrow and on the RBC traffic

in the meninges. However, there is evidence that PS-mediated effects on BM can stimulate the clearance of toxins from the brain (*BBA Clinical* 6 (2016) 113–124. doi: 10.1016/j.bbacli.2016.09.002). Moreover, Oron and co-workers [*J. Mol. Neurosci.* 55 (2015) 430–436. doi: 10.1007/s12031-014-0354-z] have shown that delivering infra-red light to the mouse tibia (using either surface illumination or a fiber optic) resulted in an improvement in a transgenic mouse model of Alzheimer's disease (AD). Light was delivered weekly for 2 months, starting at 4 months of age (progressive stage of AD). They showed improved cognitive capacity and spatial learning, as compared to sham-treated AD mice. They proposed that the mechanism of this effect was to stimulate c-kit-positive mesenchymal stem cells (MSCs) in autologous BM to enhance the capacity of MSCs to infiltrate the brain, and clear β -amyloid plaques [*Photomed. Laser Surg.* (2016). doi: 10.1089/pho.2015.4072.].

Comment: The human data does not convincingly demonstrate the presence of RBCs in the meningeal lymphatics. Why are the rendering different in the control and IVH group? Better more annotated images would be required to validate the presence of RBC in human meninges.

Response: We added the new results demonstrating the RBCs presence in the human meninges of patients died after IVH (Page 4, Lines 98-101, Figure 1 e).

Comment: In the survival graph, are the IVH and IVH+PS group significantly different? The authors should indicate for each graph what statistical test was used in order to provide clarity.

Response: We used Kaplan-Meier method for the study of survival in mice in the IVH and the IVH+PS groups. Our results demonstrate that PS significantly improves the survival of mice. Indeed, the mortality rate of adult mice was 36.6 % (11 of 30) in the IVH group, while the mortality rate of adult mice in the IVH+PS group was 23.3% (7 of 30). The number of surviving mice by day 21 of observation was significantly higher in the IVH+PS group vs. the IVH group (p=0.009, X2 test Log Rank (Mantel-Cox) = 9,391, Kaplan-Meier method) (Page 8, Lines 195-200, Figure 3a). No animals died in the sham groups (n=30 of 30).

Comment: This reviewer feels that the use of "emotional state" is anthropomorphic and misleading. It would be better if the authors just labelled it mouse behavior to not create meaning to these tasks.

Response: Thank you so much for good comment. We changed “emotional state” to “behavior state” and we performed also additional behavioral outcome assessment, which we included in our manuscript (Page 11, Lines 266-290).

Reviewer 4

This is an interesting and potentially clinically relevant studies pertaining to intraventricular hemorrhage (IVH) cleanup, showing that the non-invasive transcranial low level infra-red (1267 nm) photostimulation help in RBC clearance from the cerebral ventricles, which coincides with lowering intracranial pressure (ICP) and reduced emotional distress. The overall hypothesis is that photostimulation (PS) by the mean of affecting brain lymphatic vessels and its draining systems, through (nitric oxide) NO involving mechanism, helps improving post IVH outcome. Overall paper is interesting and well written.

Comments: Nitric oxide (NO) is normally effectively sequestered by hemoglobin. Thus, in presence of high levels of local hemoglobin a byproduct of IVH, it is counterintuitive that NO may generate such strong effect on relaxation of lymphatic vasculature. Please consider discussing this issue. Also, the in vitro evidence suggesting increase in NO production by PS is indirect and weak. Additional in vivo data confirming the role of NO is needed.

Response: The authors would like to sincerely thank the referee for the in-depth analysis of our results, constructive advice, and important suggestions for improving our paper. We performed additional *in vivo* experiments using L-NAME, a blocker of NOS. Our new results clearly demonstrate that NO blocker significantly attenuates the PS-mediated increase of the MLVs diameter and the RBCs evacuation from the brain into dcLNs suggesting an important role of NO in the PS effects on MLVs and the lymphatic clearance of RBCs. (Pages 15-17, Lines 330-376, Figure 5a,b,e, Page 24/25, Lines 518-559).

Indeed, NO is effectively sequestered by hemoglobin. However, the hemoglobin as a byproduct of blood degradation can stimulate the NO generation and thus contributes vasorelaxation (Pages 24/25, Lines 528-542).

Comment: The behavioral assays used by the investigators probe general sickness level and have limited specificity to measure IVH outcome. Broader range of behavioral outcome assessment should be used.

Response: We carried out an additional behavior assessment of mice with IVH without and after the PS-course. The new results are presented in Pages 11-13, Lines 266-290, Figure 4).

Comment: Presence of heparin in the blood used to generate SAH could influence interpretation of the outcome. E.g. heparinized blood used to produce experimental intracerebral hemorrhage dramatically inhibits edema formation as compared to non-heparinized blood. Could the heparin-inhibited RBC aggregation be essential for allowing the PS-facilitated cleanup process?

Response: Thank you so much for this the interesting question. Here situation is ambiguous and can be different in adults and newborns. Indeed, the IVH is associated with RBC aggregation, especially in pre-term newborns, which creates additional difficulties in the therapy of such babies (*Biomedical Journal* Volume 43, Issue 3, June 2020, Pages 268-276 <https://doi.org/10.1016/j.bj.2020.03.006>). The basic anatomic structure of the germinal matrix of the premature infants may predispose them to IVH. The veins in this region make a 180° turn at the caudate nuclei and drain via internal cerebral veins. This anatomical arrangement may predispose to turbulence in blood flow and promotes platelet aggregation and vascular instability. The heparin-inhibited RBC aggregation can be essential for promoting the PS-mediated cleanup process. Also, it is important to emphasize, that only non-phagocytized RBCs arrive from the brain into dcLNS, where the extremely rapid ingestion and digestion of RBCs by lymph node macrophages is observed (*M. Oehmichen, H. Wietholter, H. Gruninger, and M. Gencic, "Destruction of intracerebrally applied red blood cells in cervical lymph*

nodes. *Experimental investigations,* *Forensic Sci Int* 21(1), 43-57 (1983)). From this point of view, the heparin-related preventing RBCs aggregation may also contribute to the rapid removal of red blood cells from the brain.

At the same time, in our experiments on adult mice we observed that injection of heparinized blood into the right lateral ventricle was associated with the penetration of blood into the brain tissues and with vasogenic edema 3 days after IVH suggesting an increase permeability of the cerebral blood vessels. We added additional histological data presented in Figure 3 d,e,h-n. The role of heparin in the permeability of the cerebral blood vessel is not fully understood [*Immunity*. 2011 Feb 25;34(2):258-68. doi: 10.1016/j.immuni.2011.02.008; *ICMx* 4, 33 (2016). <https://doi.org/10.1186/s40635-016-0104-3>; *Exp Neurol*. 2020 Aug;330:113320. doi: 10.1016/j.expneurol.2020.113320]. There is evidence that heparin increases the vascular permeability to the blood [*Immunity*. 2011 Feb 25;34(2):258-68. doi: 10.1016/j.immuni.2011.02.008; *ICMx* 4, 33 (2016). <https://doi.org/10.1186/s40635-016-0104-3>]. Hemoglobin and heme are potent cytotoxic chemicals capable of causing death to many brain cells [*Stroke*. 2011;42(6):1781-1786. doi:10.1161/STROKEAHA.110.596718; *J Cereb Blood Flow Metab*. 2003;23:629–652. doi: 10.1097/01.WCB.0000073905.87928.6D]. Prominently, the mechanism of hemoglobin toxicity is via generating free radicals (mainly through Fenton-type mechanism) and massive oxidative damage to proteins, nucleic acids, carbohydrates and lipids [*Stroke*. 2011;42(6):1781-1786. doi:10.1161/STROKEAHA.110.596718; *J Cereb Blood Flow Metab*. 2003;23:629–652. doi: 10.1097/01.WCB.0000073905.87928.6D; *Stroke*. 2003;34:2964–2969. doi: 10.1161/01.STR.0000103140.52838.45; *Dev Neurosci*. 2002;24(2-3):154-60. doi: 10.1159/000065703; *Brain Res*. 2005 Mar 28;1039(1-2):30-6. doi: 10.1016/j.brainres.2005.01.036]. This can explain high mortality rate in adult mice with IVH.

However, the injection of blood into the right later ventricle in newborn animals was not accompanied by formation perivascular hematoma as we observed in adult mice (Fig. 3 e and Fig. 8 e). This fact can be explained by the higher resistance capacity of the blood-brain barrier to injuries in newborn rodents than adult animals [*J Neurosci*. 2012 Jul 11; 32(28): 9588–9600. doi:10.1523/JNEUROSCI.5977-11.2012].

Comment: For decades and through hundreds of research papers, PS has been shown to produce highly pleiotropic biological effects (other than lymphatic engagement) with many of them having high relevance to pathobiology of IVH – both systemic and cellular. Thus, a stronger direct evidence for causal relationship between lymphatic system and PS in post IVH outcome (to exclude other beneficial effect of PS), could be highly beneficial.

Response: Our experiments with photodynamic injuries of MLVs clearly demonstrate that the loss of MLVs network is accompanied by dramatically decrease in the RBCs removing from the right later ventricle into dcLNs and PS does not affect these process, i.e. the PS effects on lymphatic clearance of RBCs do not appear in mice with ablation of MLVs (Pages 15-17, Lines 352-376, Figure 6). These facts give evidence to conclude about direct evidence for relationship between PS and lymphatic removing of RBCs in post IVH period.

Comment: Certainly, as mentioned by the Authors, photostimulation in humans may not be feasible due to scattering/extinction of trans-cranially applied energy. However, the method could be potentially useful for newborns, as rightfully pointed by the Authors. As such it is unclear why experiment with model of IVH in newborn animals was not tested.

Response: We performed the additional experiments on newborn rats. The new data demonstrating the therapeutic effects on rat pups are presented in Pages 17-23, Lines 378-476, Figures 7 and 8, Table S5, Figures S8 and S9.

Comment: The Authors demonstrate the increased number of RBC in dcLN only at one hour post IVC RBC injection. To understand the dynamic of the PS effect a timecourse (including RBC count in dcLN at both earlier and later timepoint) could be useful.

Response: We investigated the effects of PS course on the RBCs evacuation from the brain and their presence in dcLNs from 1 to 7 days of observation (Page 10, Lines 235-243, Figures 3 f and g).

Comment: Also, to excluded possibility that the influx of RBC is increased by photostimulation it needs to be determined that the RBC efflux from the nodes is not influenced by the photostimulation.

Response: This is an interesting line of thought. We investigated the delivery of liposomes and different dyes from dcLNs into the meninges as well as the PS stimulation of this process [*Nanophotonics*, vol. 10, no. 12, 2021, pp. 3215-3227. <https://doi.org/10.1515/nanoph-2021-0212>]. We added a discussion about it (Page 23, Lines 479-494).

Comment: Fig 1. Quantitative analysis for RBC in dcLNs should be provided.

Response: The quantitative analysis for RBC in dcLNs is presented in Figure 2 and is described in manuscript (Pages 6 and 7, Lines 170-187).

Comment: Are RBC present in lymph nodes free or insight of phagocytes? Lyve-1 is known to recognize some of the subtypes of macrophages. The images provided in their work do not exclude possibility that RBC are internalized by macrophages and as such are in the lymphatic system. Also, on representative images from panels 2 and 3, please include nuclear stain (e.g. DAPI) to show what is the overall population of cells on the micrograph.

Response: We did not include the human data of RBCs accumulation in dcLNs in our manuscript. However, we have the human results of hemosiderin presence in dcLNs after death of adult patients with IVH (Figure 1 is below). This means that the some RBCs are destroyed in the brain and are arrived into dcLNs as a product of blood degradation. We do not exclude the same scenario for rodents. Nevertheless, we did not find hemosiderin in dcLNs in adult mice. Oehmichen et al. discovered on rabbits that only non-phagocytized RBCs arrive from the brain into dcLNs, where is observed the extremely rapid ingestion and digestion of RBCs by lymph node macrophages [*M. Oehmichen, H. Wietholter, H. Gruninger, and M. Gencic, "Destruction of intracerebrally applied red blood cells in cervical lymph nodes. Experimental investigations," Forensic Sci Int 21(1), 43-57 (1983)*].

Indeed, Lyve-1 is known to recognize some of the subtypes of macrophages. However, the LVs were labeled with Lyve-1/Prox1, i.e. we used two markers for identification of LVs in dcLNs and in the meninges (Figure 1). In new version of manuscript, we changed the Figure 2, where we added DAPI (Page 7, Lines 175-184).

Figure 1 – The histological analysis of hemosiderin (Perls' Prussian blue stain) presence in the human dcLNs in the normal state (a) and after IVH (b), bar – 64.6 x.

Comment: Is the presence of RBC/hemoglobin remaining in the ventricles/CSF reduced significantly with PS? A total number of RBC cleared by the lymphatic system could be small and not essential in the overall blood clearance process. If so, it could mean that PS works through some other process.

Response: We discuss the alternative pathways of RBCs removing from the brain (Page 23, Lines 479-494).

Comment: Fig 5 legend – correct IHV to IVH.

Response: We made correction of Fig. 5 legend.

Comment: Please indicate (especially for experiments described in Fig 4) when after IVH photostimulation was initiated. Also, regarding therapeutic implications, is the therapeutic window used in this study clinically relevant?

Response: We added the schematic illustration of design of experiments, which is presented in Figures 3c and 8c. Regarding to therapeutic window, the employment of laser wavelengths within the so-called tissue transparency window (600–1300 nm) ensures low light absorption in superficial tissue layers, thus providing deeper light penetration into the treated volume [Tuchin, V.V. Volume 1: Light-Tissue Interaction. In Handbook of Optical Biomedical Diagnostics, 2nd ed.; SPIE: Bellingham, WA, USA, 2016; J Invest Dermatol. 1981 Jul;77(1):45-50. doi: 10.1111/1523-1747.ep12479235]. The infra-red light of 800-1100 nm widely used for the PS therapy of brain diseases [M. R. Hamblin, "Low-Level Light Therapy," (2018). <https://spie.org/Publications/Book/2295637?SSO=1>; Int J Stroke 8(5), 315-320 (2013).18-23. doi: 10.1111/j.1747-4949.2011.00754.x; Acta Neurochir Suppl . 2016;121:7-12. doi: 10.1007/978-3-319-184975_2; J Neurosci Res . 2018 Apr;96(4):731-743. doi: 10.1002/jnr.24190. Epub 2017 Nov 13; J Biophotonics . 2015 Jun;8(6):502-11. doi: 10.1002/jbio.201400069. Epub 2014 Sep 8; Neuropsychiatr Dis Treat . 2015 Aug 20;11:215975. doi: 10.2147/NDT.S65809].

However, the infra-red PS has a significant limitation, such as limited penetration into the brain due to light scattering and heating effect [Lasers Surg Med . 2015 Apr;47(4):312-22. doi: 10.1002/lsm.22343. Epub 2015 Mar 13].

The light wavelength of 1300 nm has less scattering and can penetrate deeper into the brain [Nat Methods 15, 789–792 (2018). <https://doi.org/10.1038/s41592-018-0115-y>].

Therefore, we selected optimal PS (1267 nm) dose for the experiments studying the light transmission, the changes in the temperature on the skull and the surface of the brain after PS 3-69-18-27 J/cm² as well as the morphological changes in the brain tissues after the PS-course 9 J/cm².

Our data clearly demonstrate that PS 9 J/cm² vs. other PS doses was most effective for stimulating lymphatic clearance of GNRs and EBD without heating effect and any morphological changes in the brain tissues that determined our choice of PS 9 J/cm² as the optimal dose for further investigations.

We discuss this in manuscript (Page 24, Lines 502-512) as well as in our review [*Int. J. Mol. Sci.* 2022, 23(6), 2975; <https://doi.org/10.3390/ijms23062975>] and original articles [*Nanophotonics*, vol. 10, no. 12, 2021, pp. 3215-3227. <https://doi.org/10.1515/nanoph-2021-0212>; *Translational Biophotonics* 2(1-2), (2020). doi.org/10.1002/tbio.201900036; *Biomed Opt Express* 11(2), 725-734 (2020). <https://doi.org/10.1364/BOE.383390>].

Our findings of the PS-therapy of IVH in adults and newborns are pilot. Therefore, we cannot make comparison between our results and other data related to the time of PS application for the IVH treatment. In our study, the time window of PS treatment is at least 3 days, because we began the PS courses 3 days after IVH and could have significant improvement on the survival rate and behavioral tests.

Comment: Fig 4. When after the IVH onset behavioral tests were conducted? Because of the high mortality in IVH groups how many animals were ultimately subjected to the behavioral evaluation. Was mortality and behavior evaluated in separate cohort of animals? Finally, line 190 – the authors indicate that the mortality was “1.57-fold lesser” in the treated group. Based on Fig 4b there is no statistical difference indicated for IVH vs. IVH-PS groups. Please clarified use of “lesser” and provide p value.

Response: We added in Figure 3c the time points of behavioral outcome assessment as well as we added in manuscript explanation of high mortality among adult mice with the statistical analysis of mortality rate.

Comment: Some histological analyses comparing IVH vs. IVH+PS needs to be performed to further validate conclusions of the functional (behavioral) findings. Eg., hydrocephalous is one of the consequences of IVH. Was there any difference detected between ventricular sizes in animals from various treatment groups?

Response: We added the additional experiments and made the histological and quantitative analysis of the periventricular hematoma size, the PVS size, and the ventricle size in groups, including: 1) the sham, 3 days; 2) the IVH, 3 days; 3) the sham, 11 days; 4) the IVH, 11 days; 5) the sham+PS, 11 days; 6) the IVH+PS, 11 days after injection of saline (for the sham groups) or blood (for IVH groups) into the right lateral ventricle, which are presented in Figure 3 h-n and Figure 8 d-l (Pages 10/11, Lines 229-265; Page 20-22, Lines 429-455).

Reviewers' comments:

Reviewer #2 (Remarks to the Author):

The authors did not address my concern well. The main concern remains.

Reviewer #3 (Remarks to the Author):

First I would like to thank the authors for addressing my comments from the first round of revision.

I however still have some comments, particularly towards the new data that was added:

- My primary concern is regarding the neonatal rat experiment. As the authors specify, the meningeal lymphatic system, particularly its dorsal portion, fully develop postnatally in mice. Furthermore, some reports are suggesting that their draining function is present even later (around P15 in mice). Given the lack of information regarding the developmental timing and functionality of the meningeal lymphatics in the neonatal rats, it is hard to interpret the data from the PS treatment in P10 rats. If compared to mice lymphatic, they are still almost virtually absent from the dorsal region and not yet functional. Are the effects of PS still lymphatic dependent in neonatal rats ? through the same pathways ? I do not believe the neonatal model is adding to the story compared to the adult one, particularly as the mechanisms may be very different and would require substantial experiments to validate the neonatal model.

Reviewer #4 (Remarks to the Author):

I have no further comments.

Response to Reviewers

Reviewer 2.

Comment: The authors did not address my concern well. The main concern remains.

(noted by the editor: Before reaching a decision, I would like to ask that you provide further clarifications related to the concerns of Reviewer #2 on the high mortality rates that you observe with this modified protocol.)

Response: We would like to thank you so much for this reviewer's helpful comment. Since the research was carried out in three independent scientific groups (in China, the USA and Russia) and over a period of 4.5 years (since 2018), we analyzed the conduct of experiments in each group.

The study of PS effects for therapy of IVH started in early 2018 in Russia. To understand the efficacy of therapeutic PS effects, we started by looking at mortality in aged mice. For this, a large group of 30 aging mice (12-14 months old) was recruited. We selected aged mice because older age is the risk factor for IVH (Neurosurgery. 2006; 59(4):767–773. doi: 10.1227/01.NEU.0000232837.34992.32) as well as because turnover and drainage of CSF decrease with aging (Nature. 2018 Aug;560(7717):185-191. doi: 10.1038/s41586-018-0368-8; Nat Rev Neurol. 2015 Aug;11(8):457-70. doi: 10.1038/nrneuro.2015.119; Nat Commun. 2017 Nov 10;8(1):1434. doi: 10.1038/s41467-017-01484-6). The aged mice show regression of dorsal MLVs, playing a crucial role in drainage and clearing of the brain tissues, compared with young mice (Nature. 2019 Aug;572 (7767):62-66. doi: 10.1038/s41586-019-1419-5). The injury of the brain drainage system is a crucial mechanism of ICH and can be an important reason for high mortality after IVH (Nat Commun 11, 3159 (2020). <https://doi.org/10.1038/s41467-020-16851-z>; Acta Neuropathol Commun 8, 16 (2020). <https://doi.org/10.1186/s40478-020-0888-y>). Thus, the mortality rate of 36.6 % (11 of 30) in the IVH group can be explained by a decrease in the lymphatic evacuation of RBCs from the ventricles in aged mice that leads to blockage of CSF pathways outflow in particularly narrow places, such as Monroe's foramina, cerebral aqueduct, foramen of Luschka and Magendie.

Since our studies have shown the high efficacy of PS in terms of survival of mice, we decided to continue studying the mechanisms of phototherapy of IVH in three independent groups (in China, in the USA and in Russia), which were performed from 2019 to 2022 on mice 2-3 months old. The choice of 2–3-month-old mice for further experiments was associated with the identification of experimental conditions, lineage, and age of mice in the three research groups, as well as a reduction in the risk of high mortality among mice in our long-term experiments, which we observed in aged animals. However, their morphology in the brain tissues and ICP, as well as behavioral outcomes, were still remarkably influenced by IVH, where such a situation could be improved by PS.

Thanks for the reviewer's comment; we checked our original experimental record and realized that we used the mortality rate of aged mice instead of 2-3-month-old mice in our manuscript.

We corrected this methodological inaccuracy and performed additional survival experiments in mice 2-3 months of age. Mortality among these mice is 10% (2 out of 20), which corresponds to the literature data (PLoS One. 2014 May 15;9(5): e97423. doi: 10.1371/journal.pone.0097423). We have made changes to the survival rate in Fig. S10a.

As for the heparin, we are sorry for the misleading method description. Blood was not physically mixed with heparin. We used a sterile Eppendorf pre-flushed with heparin to avoid coagulation during blood sampling and injection. Therefore, we did not use the sham group (PBS + heparin). It has been shown that high concentrations of heparin can increase the vascular permeability to the blood (Immunity. 2011 Feb 25;34(2):258-68. doi: 10.1016/j.immuni.2011.02.008; ICMx 4, 33 (2016). <https://doi.org/10.1186/s40635-016-0104-3>); however, the role of heparin in cerebral blood vessels permeability is not fully understood. Theoretically, the residual micro-concentration of heparin from the walls of eppendorf might enter into the blood and affect the cerebral vessels BBB permeability. However, in our case, the residual heparin concentration in a sterile eppendorf flushed with heparin was negligible and unlikely to alter the BBB permeability.

In the revised manuscript, we have moved the results on the study of mortality rate, morphological changes in the brain tissues, ICP as well as the behavior outcomes (Fig. 3 and 4 became Fig. S10 and S12) obtained on adult mice to SI to make clearer the applied aspect of our study. We demonstrate in adult mice the successful application of PS for the stimulation of RBCs clearance

from the brain as well as the possible mechanisms underlying PS of IVH and the optimal dose of PS therapy. However, the main obstacle to PS application is limited laser penetration into the brain due to the scattering effects of the skull. Nevertheless, PS-acceleration of RBCs clearance from the brain via MLVs can be clinically significant for the therapy of intracranial hemorrhages in newborns, where PS can be applied through the fontanelles in the neonatal skull, which is also a window into the MLVs localized along the sagittal and transverse sinuses. Using this optimal PS-dose established on adult mice, we show the clinical significance of PS-therapy of IVH in newborn rats by increasing their survival and behavior outcomes.

Accordingly, we have added the following Figures and paragraphs in the revised manuscript:

Pages 18-20, Lines 385-461: In the final step, we studied the therapeutic effects of PS on adult mice with IVH. The schematic diagram of the time points of experiments and the number of animals in the experimental groups are presented in Fig. S10 c. Our results demonstrate completely other PS effects on adult mice compared with newborn rats after IVH. The adult mice (2-3 months old) did not show statistically significant differences in the mortality between the IVH and IVH+PS group ($p=0,161$, X2 test Log Rank (Mantel-Cox) = 1,961, Kaplan-Meier method) because only 2 mice died (2 of 20) in the IVH group and no dead mice were in the IVH+PS group ($n=20$) (Fig. S10 a). No animals died in the sham groups ($n=20$).

However, despite the fact that mortality among adult mice (2-3-month-old) was low, they demonstrated significant IVH-mediated changes in brain morphology and behavior that were improved after the PS course.

The IVH was accompanied by the formation of periventricular hematoma, which was reduced after 11 days of recovery. The PS course significantly improved brain recovery after IVH. Indeed, the size of periventricular hematoma 11 days vs. 3 days after IVH was 2.51-fold lesser in the IVH+PS group ($0.27\pm 0.05 \text{ mm}^2$ vs. $0.68\pm 0.12 \text{ mm}^2$, $n=10$ in each group, $p=0,000001$, the Welch's test) and 1.58-fold lesser in the IVH group ($0.43\pm 0.12 \text{ mm}^2$ vs. $0.68\pm 0.12 \text{ mm}^2$, $n=10$ in each group, $p=0,000299$, the Welch's test) (Fig. S10 b and l).

The faster-reducing hematoma after the PS course was accompanied by faster RBCs removal from the brain. Fig. S10 f and g clearly demonstrate that the number of RBCs in dcLNs gradually decreased from 1 to 7 days of observation that was more pronounced in the IVH+PS group vs. the IVH group without PS ($(5.3\pm 0.4)\times 10^5$ per mm^3 vs. $(2.0\pm 0.4)\times 10^5$ per mm^3 , $p<0.001$, $n=6$, independent-samples T-test for 1 hour after IVH; $(3.1\pm 0.3)\times 10^5$ per mm^3 vs. $(2.3\pm 2.6)\times 10^5$ per mm^3 , $p<0.01$, $n=6$, independent-samples T-test for 1 day after IVH; $(1.6\pm 0.1)\times 10^5$ per mm^3 vs. $(2.6\pm 0.1)\times 10^5$ per mm^3 , $p<0.001$, $n=6$, independent-samples T-test for 3 days after IVH; $(0.9\pm 0.2)\times 10^5$ per mm^3 vs. $(1.6\pm 0.2)\times 10^5$ per mm^3 , $p<0.01$, $n=6$, independent-samples T-test for 7 days after IVH, respectively).

The PS course effectively reduced the vasogenic edema observed in mice after IVH (Fig. S10 h-k and m). So, the size of the perivascular space (PVS) increased by 3.37 times in mice 3 days after IVH vs. sham mice ($0.027\pm 0.007 \text{ Jim}$ vs. $0.008\pm 0.001 \text{ Jim}$, $n=10$ in each group, $p=0.0001$, the Welch's test) and decreased by day 11 of recovery. In this recovery time compared with 3 days after IVH, the PVS size was 2.25-fold lesser in the IVH+PS group and 1.42-fold lesser in the IVH group without PS-course ($0.012\pm 0.001 \text{ Jim}$ vs. $0.027\pm 0.007 \text{ Jim}$, $n=10$ in each group, $p=0,000009$ and $0.019\pm 0.004 \text{ Jim}$ vs. $0.027\pm 0.007 \text{ Jim}$, $n=10$ in each group, $p=0.006$, the Welch's test, respectively).

The PS course promoted a faster recovery of the ventricles, which dilated after IVH (Fig. S10 n and Fig. S11). Indeed, the right lateral ventricle size was 3.5-fold higher in mice 3 days after IVH

vs. sham mice ($0.91 \pm 0.02 \text{ mm}^2$ vs. $0.26 \pm 0.06 \text{ mm}^2$, $n=10$ in each group, $p=0.00001$, the Welch's test). By 11 days, the ventricle size was decreased (Fig. S10n and S11). However, during this time, the size of the right later ventricle was 2.16-fold lesser in the IVH+PS group and 1.44-fold lesser in the IVH group without the PS course ($0.42 \pm 0.01 \text{ mm}^2$ vs. $0.91 \pm 0.02 \text{ mm}^2$, $n=10$ in each group, $p=0.0001$ and $0.63 \pm 0.01 \text{ mm}^2$ vs. $0.91 \pm 0.02 \text{ mm}^2$, $n=10$ in each group, $p=0.0001$, the Welch's test, respectively).

Figure S10 o shows clear evidence that IVH is accompanied by a dramatic rise in ICP that was significantly improved by PS (single laser dose 9 J/cm^2). The blood injection into the right lateral ventricle caused an immediate increase in ICP ($75.5 \pm 12.5 \text{ mm Hg}$ vs. $10.2 \pm 2.1 \text{ mm Hg}$, $p<0.01$ for the IVH group and $74.2 \pm 11.4 \text{ mm Hg}$ vs. $10.2 \pm 2.1 \text{ mm Hg}$, $p<0.01$ for the IVH+PS group, $n=10$ in each group, the Welch's test). Afterward, ICP gradually decreased but remained to be high by the end of 60 min monitoring. The PS significantly reduced the time of ICP recovery. Indeed, the recovery of ICP in the IVH+PS group was faster than in the IVH group ($29.0 \pm 3.1 \text{ mm Hg}$ vs. $19.6 \pm 3.2 \text{ mm Hg}$ at 60 min of monitoring, respectively, $p<0.05$, $n=10$ in each group, the Welch test).

The development of IVH was accompanied by a neurological deficit with changes in locomotor and memory functions that were significantly improved after the PS course (Fig. S12 a-e). All behavior tests were performed 11 days after the injection of blood (the IVH and the IVH+PS groups) or saline (the sham groups) into the right later ventricle (Fig. S10 c). The locomotor deficits were evaluated by a 24-point neurologic scoring system, where higher scores indicate a more significant deficiency, and by the wire-hanging test, which measures gripping and forelimb strength (Fig. S12 c and d). Mice in the IVH group had a more substantial motor deficit than did mice in the sham group ($n=10$ in each group, all $p<0.001$, the Wilcoxon test). Gripping and forelimb strength were significantly impaired in the IVH group compared with the sham group ($n=10$ in each group, $p<0.001$, the Wilcoxon test). Mice after the PS course demonstrated a significant decrease in neurological deficit score ($n=10$ in each group, $p<0.001$, the Wilcoxon test) as well as higher gripping and forelimb strength than mice in the IVH group without the PS-course ($n=10$ in each group, $p<0.001$, the Wilcoxon test) (Fig. S12 c and d).

The immobility time in the forced swim and tail suspension tests was longer in the IVH than in the sham group ($n=10$ in each group, $p<0.001$; Fig. S12 a and b, the Wilcoxon test). Mice after the PS course improved the time of immobility in both tests ($n=10$ in each group, $p<0.001$; Fig. S12 a and b).

In the novel object recognition test, sham mice spent more time exploring the novel object than the old ones ($n=10$ in each group, $p<0.001$; the Wilcoxon test, Fig. S12 e), but mice with IVH spent a similar amount of time exploring the new and old objects ($n=10$ in each group, $p<0.001$; the Wilcoxon test, Fig. S12 e) suggesting the recognition memory deficit. After the PS course, mice performed this test similarly to the sham group and spent more time exploring the novel object than the old object, suggesting memory function improvement.

Thus, this series of experiments demonstrated that the PS course significantly reduces the mortality rate and brain injuries as well as improves the recovery of ICP, locomotor and memory functions after IVH.

Figure S10 – Therapeutic effects of the PS-9 J/cm² course in adult mice with IVH: (a) Kaplan-Meier overall survival plots in the tested groups with and without PS-course; the number of surviving mice by day 21 of observation was significantly higher in the IVH+PS group than in the

IVH group (n=30 in each group, p=0.009, X2 test Log Rank (Mantel-Cox) = 9,391); **(b)** – Representative 2D images of the normal brain, 3 and 11 days after IVH with and without the PS-course, n=10 in each group; **(c)** The schematic diagram of time points of experiments and the number of animals in the experimental groups; **(d and e)** – Histological imaging of the normal brain tissues around the right later ventricle and the periventricular hematoma formed 3 days after IVH, n=10 in each group; **(f)** Representative confocal images of presence of RBC in dcLNs of mice without and after the PS-course, n=6 in each group; **(g)** The number of RBCs in dcLN 1 hour, a day, 3 days and 7 day after IVH with/without PS, ** p<0.01, *** p<0.001, n=6 in each group, independent-samples T test; **(h-k)** Histological images illustrating the normal brain tissues without vasogenic edema (h); the significant increase of the PVS size 3 days after IVH (i); the reducing the PVS size 11 days after IVH (j) that was more pronounced in the IVH+PS group (k) compared with the IVH without PS group (j), n=10 in each group; **(l)** The quantitative analysis of periventricular hematoma (mm²) formed on 3 days and 11 days after IVH with and without the PS-course, n=10 in each group, Mean ± SD, the Welch’s test); **(m)** The quantitative analysis of the PVS size (µm) on 3 days and 11 days after IVH with and without the PS-course, n=10 in each group, Mean ± SD, the Welch’s test); **(n)** The quantitative analysis of the ventricular area size (mm²) on 3 days and 11 days after IVH with and without the PS-course, n=10 in each group, Mean ± SD, the Welch’s test); **(o)** The continues monitoring of ICP in sham mice and in mice after IVH with and without PS (single laser dose 9 J/cm²).

Figure S11 – 2D Images of the size of the right lateral ventricle (three projections) in adult mice in the control (sham) group, 3 days and 11 days after IVH without and after PS-course 9 J/cm².

Figure S12 – The PS effects on the locomotor and memory functions: (a and b) The immobility time (sec) in the forced (a) and tail (b) tests in sham mice and in mice with IVH without and after the PS-course, n=10 in each group, Mean ± SD, *** - p<0.001 in all comparisons, the Wilcoxon test; (c) The falling latency time (sec) in the wire hanging test in sham mice and in mice with IVH without and after the PS-course, n=10 in each group, Mean ± SD, *** - p<0.001 between the sham and the IVH groups, between the IVH and the IVH+PS groups; ** - p<0.05 between the sham+PS and the IVH+PS groups; the Wilcoxon test; (d) The neurologic deficit score measured by a 24-point scoring system in sham mice and in mice with IVH without and after the PS-course, n=10 in each group, Mean ± SD, *** - p<0.001 in all comparisons, the Wilcoxon test; (e) The time of exploring of the new and old objects in the novel object recognition test in sham mice and in mice with IVH without and after the PS-course, n=10 in each group, Mean ± SD, *** - p<0.001 in all comparisons).

Reviewer 3.

Comment: First, I would like to thank the authors for addressing my comments from the first round of revision. I, however, still have some comments, particularly regarding the new data that was added:

My primary concern is regarding the neonatal rat experiment. As the authors specify, the meningeal lymphatic system, particularly its dorsal portion, fully develops postnatally in mice. Furthermore, some reports suggest that their draining function is present even later (around P15 in mice). Given the lack of information regarding the developmental timing and functionality of the meningeal lymphatics in neonatal rats, it is hard to interpret the data from the PS treatment in P10 rats. If compared to mice lymphatics, they are still almost virtually absent from the dorsal region and not yet functional. Are the effects of PS still lymphatic-dependent in neonatal rats? through the same pathways? I do not believe the neonatal model is adding to the story compared to the adult one, particularly as the mechanisms may be very different and would require substantial experiments to validate the neonatal model.

Response: Thank you so much for the important remarks. We completely agree with the referee that there is no ideal neonatal model for comparison the adult animals and, especially, to be relevant to humans. However, we did not make a comparison between adult and newborn animals. We present these age groups separately and interpret them differently because adult mice and newborn rats have different resistance to IVH and recovery after IVH. It was not a trivial task to select the appropriate newborn model for our study. Our choice was based on a large number of data suggesting that rat brain maturation and human neonates have similarities in anatomical and functional aspects⁵¹⁻⁵⁶. There is evidence that PD 7-10 pups are equivalent to the near-term human brain⁵²⁻⁵⁵.

We agree with the referee that there is very limited information about the maturation of the lymphatic vessels. Antila et al. presented the results of an anatomical network of MLVs in mice in the early and late stages of ontogeny (J Exp Med 214, 3645-3667 (2017). doi: 10.1084/jem.20170391). The MLVs appear along the main cerebral veins in the dorsal aspect of the brain only by the 16th day and later. However, in the basal aspect of the brain, MLVs mature very early, by the 2nd day - in the cribriform plate, by the 4th day – along the pterygopalatine artery, and by the 8th day – along the middle meningeal artery.

Accordingly, we have performed extra experiments and added the following paragraphs in the revised manuscript:

Pages 12-13, Lines 297-309: Since there is no information about the anatomical and functional maturation of the lymphatic drainage and clearing in newborn rats, we performed the experimental analysis of lymphatic removal of FITC-dextran 70 kDa (i.c.v.) from the brain of 10-days old pups without and after PS using the published protocol⁵⁷. Our data clearly demonstrate the effective lymphatic removal of tracer from the brain into the dcLNs in pups (Fig. S7). The intensity of the fluorescent signal from FITC-dextran was more significant in basal vs. dorsal aspects of the brain, suggesting that the tracer moved from the ventricle predominantly to the ventral area of the brain, where the basal MLVs are localized, playing an essential role in the brain drainage and clearance³⁶. We observed the presence of FITC-dextran in the area of the cribriform plate, which is the main route of brain lymphatic drainage and clearance^{46, 58}. The PS significantly increased the evacuation of dye from the brain and its accumulation in dcLNs. Thus, 10-days old rat pups demonstrate the clearance of FITC-dextran from the brain into the peripheral lymphatic system suggesting that they have enough effective lymphatic mechanisms of drainage and clearance of the brain.

Figure S7 – The lymphatic clearance of FITC-dextran 70 kDa (i.c.v.) from the brain of 10-days old rats without and after PS. The cerebral blood vessels were filled by 1% Evans Blue (red color): (a) distribution of the tracer in the ventral aspect of the brain without (v no PS) and after (v PS) PS; (b) distribution of the tracer in the dorsal aspect of the brain without (d no PS) and after (d PS) PS; (c) accumulation of tracer in dcLNs without (dcLNs no PS) and after (dcLns PS) PS; (d) distribution of the tracer in area of the cribriform plate; (e) and (f) the quantitative analysis of intensity signal from FITC-dextran in the brain (e) and in dcLNs (f); n=5 in each group, the Mann-Whitney U-test (Wilcoxon rank sum test with continuity correction (Mann–Whitney–Wilcoxon) was performed in all comparisons.

- [36] Ahn JH, et al. Meningeal lymphatic vessels at the skull base drain cerebrospinal fluid. *Nature* 572, 6266 (2019).
- [46] Koh L, Zakharov A, Johnston M. Integration of the subarachnoid space and lymphatics: is it time to embrace a new concept of cerebrospinal fluid absorption? *Cerebrospinal fluid research* 2, 6-16 (2005).
- [57] Semyachkina-Glushkovskaya O, et al. Night Photostimulation of Clearance of Beta-Amyloid from Mouse Brain: New Strategies in Preventing Alzheimer's Disease. *Cells* 10, (2021).
- [58] Helen F. Cserr, Christine J. Harling-Berg, Knopf PM. Drainage of Brain Extracellular Fluid into Blood and Deep Cervical Lymph and its Immunological Significance. *Brain Pathol* 2, 269-276 (1992).

REVIEWER COMMENTS

Reviewer #3 (Remarks to the Author):

First I would like to thank the authors for answering my comments from the previous round of revision.

However, there is some inconsistency in some of the presentation that make the article difficult to follow.

The authors are focusing their studies on the meningeal lymphatic vasculature of the dorsal and ventral part of the meninges, target them with photostimulation to facilitate red blood cell drainage. However, in the neonatal model, but also in the conclusion, the authors are stating that the cribriform plate drainage is the main route for CSF drainage. There is some discrepancy in the statement. How is the effect of P/S on adult cribriform plate drainage ? The statement that cribriform plate route is the main route out for CSF is also highly controversial and debated in the field with numerous studies now questioning the importance of this pathway in human drainage.

Furthermore, it still remains unclear how photo stimulation of the mostly dorsal region of the skull would affect the lymphatics of the cribriform plate to stimulate drainage.

Reviewer #5 (Remarks to the Author):

Li et al. explore the capacity of transcranial low-level infra-red photostimulation in red blood cells evacuation from the ventricles of newborn and adult mice and rats. Authors explain that low-level infra-red photostimulation reduces intracranial pressure, reduces mortality and improves neurological outcomes. Parallel studies are performed in human tissue and authors also analyze the role of nitric oxide in observed effects.

The study is relevant and provides an interesting venue for patients in need of new therapeutic options. Nevertheless the manuscript also raises some relevant concerns.

Concerns

1. Abstract should be rewritten to include more detailed and relevant information of the manuscript. General information is given but specific results to support the provided statements are missing.

Information regarding explored mechanisms or human studies are also missing (both in the abstract and the introduction).

2. Bibliography should be carefully revised and updated. The manuscript largely relies in bibliography that has over 40 or even 70 years (eg. 1952, 1979, 1983). While there might be seminal studies worth to be cited, they should be accompanied by recent studies confirming these data. The fact that no recent bibliography supports these outcomes should be taken into consideration. Also some sentences should be rewritten to avoid misinterpretations and errors related to the provided bibliographic support. Eg:

3 “Brain hemorrhage in adults and premature infants has the highest morbidity and mortality worldwide”. In premature infants, IVH is a major complication associated to prematurity but it is not responsible for the highest mortality rates in this population. The referred bibliography in the manuscript is not recent and does not support these asseverations (eg. Beek P, et al. Arch Dis Child Fetal Neonatal Ed 2021; Juul JAMA Network Open.2022;5(2):e2146404. doi:10.1001/jamanetworkopen.2021.46404).

4. “Intracranial hemorrhages occur in 45% of premature infants and 46-50% of mature babies”. While IVH is the most common hemorrhage in the preterm newborn, the manuscript focuses on IVH and therefore preterm data should refer to IVH specifically. The data provided for mature babies relies on a pilot study from 1999. Again, the bibliography is far from recent and does not support the provided information.

5. “It has been shown that RBCs can be cleared from the brain”. The manuscript largely depends on this asseveration; nevertheless, this sentence relies on studies from 1952 and 1979. Bibliographical support should be updated to more recent studies showing described information.

6. The authors estate “However, the existing therapy of IVH has dramatically impacted the natural history of the disease”. It is not clear what they refer to with “impacting the natural history of the disease”.

7. Methodology should be revised and detailed information should be provided in order to allow the reproduction of the experiments. Some experiments are not even included in the methods section.

8. L-NAME experiments are not described.

9. Detailed information regarding the patients under study (apart from age) should be provided. Exact brain regions should be also specified. Specific Bioethical information consents and approvals for human studies should be provided.

10. The age of adult mice, and not only the weight, should be provided since adult mice show limited differences in weight, but results in 3 or 8 month-old mice may significantly differ (as authors also state). Also, specifying ages and sex guarantee that the studies can be reproduced by others.

11. 10 days old rats can be hardly considered preterm. The manuscript used as reference induces IVH to P4 rats. Younger rats should be used to reproduce the IVH of the preterm newborn, as a major point of the manuscript (already stated in the introduction), and given the fact that IVH are common in preterm infants but not in full-term kids.

12. Was blood used for IVH treated in anyway? Coagulation seems difficult to avoid but no specs are provided in the methods. On the other hand, figure 6 includes "hepanized blood".

13. How was sacrifice performed?

14. The specific regions, number of sections, animals... used in each experiment should be detailed. Similarly, the number, regions and selection of samples for human meninges studies should be specified.

15. Given the nature of the performed approaches and the feasible damage of the vasculature and BBB, how was this aspect taken into consideration? These experimental procedures might interfere with the actual outcomes. Similarly, FITC-dextran experiments largely rely on an intact BBB.

16. Statistical approaches are confusing. Non-parametric tests (Mann Whitney and Wilconxon tests) are performed when 2 populations are under study; however parametric (ANOVA) are used when more populations are under study. Even though Shapiro Wilks test is used to assess normality, this should be clarified. Complete data for ANOVA should be provides (degrees of freedom, f values).

17. The possibility of intranasal low-level infra-red photostimulation seems a relevant possibility, and authors consider it a relevant issue to the point of being included in the conclusions section. Nevertheless, this idea is only considered for the first time in the last paragraph of the manuscript.

Minor points

-Scattered plots in figure 1 and graphs in figure 6 are too small and difficult to decipher.

-Some expressions should be revises: "Rats are commonly employed employed.." "2.51-fold lesser..."

-Marks in figure 6 (l, j, k,l) should be more clear. They are very hard to see.

-Figures and figure legends should be autoexplicative. Ej Fig S2, S5, S8 lack information regarding staining procedure, what exact "histological analysis" is performed?

Response to Reviewers

Reviewer #3

First I would like to thank the authors for answering my comments from the previous round of revision. However, there is some inconsistency in some of the presentation that make the article difficult to follow.

Comments: The authors are focusing their studies on the meningeal lymphatic vasculature of the dorsal and ventral part of the meninges, target them with photostimulation to facilitate red blood cell drainage. However, in the neonatal model, but also in the conclusion, the authors are stating that the cribriform plate drainage is the main route for CSF drainage. There is some discrepancy in the statement. How is the effect of P/S on adult cribriform plate drainage? The statement that cribriform plate route is the main route out for CSF is also highly controversial and debated in the field with numerous studies now questioning the importance of this pathway in human drainage. Furthermore, it still remains unclear how photo stimulation of the mostly dorsal region of the skull would affect the lymphatics of the cribriform plate to stimulate drainage.

Response: The authors would like to express their gratitude for the constructive advice that helps to significantly improve our article. We agree that the cribriform plate as the main route for CSF exit, especially for RBC evacuation, is debated and needs further research. The pathways between the meningeal and peripheral networks remain in the process of active study, including our and other searches for the presence of lymphatic vessels directly in human brain tissues (Proc Natl Acad Sci U S A. 2021 Jan 19; 118(3):e2002574118. doi: 10.1073/pnas.2002574118; bioRxiv 2021.09.05.458990; doi: <https://doi.org/10.1101/2021.09.05.458990>). In our recent studies, we demonstrate intranasal delivery of liposomes to glioblastoma by photostimulation of the lymphatic system (Pharmaceutics 2023 15(1), 36; <https://doi.org/10.3390/pharmaceutics15010036>). We also show photomodulation of lymphatic delivery of liposomes to the brain bypassing the blood-brain barrier (Nanophotonics, vol. 10, no. 12, 2021, pp. 3215-3227. <https://doi.org/10.1515/nanoph-2021-0212>). We discover that the BBB opening is associated with activation of lymphatic drainage of the brain tissues and clearance of tracers from the brain to the deep cervical lymph nodes (Comput Struct Biotechnol J. 2022 Dec 15; 21:758-768. doi: 10.1016/j.csbj.2022.12.019). We discuss that sleep and photostimulation stimulate lymphatic communication between the central nervous system and peripheral lymphatics (Int J Mol Sci. 2023 Feb 6, 24(4):3221. doi: 10.3390/ijms24043221; 2022 Nov 26, 14(12):2612. doi: 10.3390/pharmaceutics14122612; IEEE Journal of Selected Topics in Quantum Electronics, vol. 27, no. 4, pp. 1-13, 2021, Art no. 7400313, doi: 10.1109/JSTQE.2020.3045834; Int J Mol Sci. 2020 Aug 31, 21(17):6293. doi: 10.3390/ijms21176293). However, a generally accepted concept of the mechanisms of lymphatic clearance of metabolites, toxins, and cells (e.g. RBCs) from brain tissues does not yet exist.

We apologize for the inconsistency in the description of the results. In fact, in the neonatal model, we studied the PS effects on lymphatic clearance of RBCs and FITCD from the brain, rather than targeting cribriform plate drainage.

To make the paper more comprehensible and focused, we have removed the description of cribriform plate drainage in the revised manuscript, we also improved the conclusion. All changes in the manuscript are marked in yellow.

Pages 13-14, Lines 315-333: Since there is no information about the anatomical and functional maturation of the lymphatic drainage and clearing in newborn rats, we performed this study of lymphatic removal of FITC-dextran 70 kDa (FITCD) from the right lateral ventricle to dcLNs of PD4 pups using the published protocol. Suppression of the brain lymphatic functions after the development of subdural and subarachnoid hemorrhages in adult mice has been reported in several publications. Based on these facts, we hypothesized that IVH will be accompanied by a decrease in lymphatic drainage and clearing of the brain tissues in newborn rats. Indeed, Figures 5 a-e show that the IVH group vs. the control (sham) group demonstrated suppression of lymphatic evacuation of FITCD from the right lateral ventricle leading to reduced spreading of tracer in the dorsal and ventral aspects of the brain as well as causing a decrease in its accumulation in dcLNs. Our detailed quantitative analysis showed that the intensity of the fluorescent signal from FITCD in the IVH group vs. the control group was decreased to 52.7% and 57.4% in the dorsal and ventral aspects of the brain as well as decreased to 13.9% in dcLNs (0.29 ± 0.02 a.u. vs. 0.55 ± 0.12 a.u. $p=0.016$, in the dorsal part of the brain; 0.35 ± 0.05 a.u. vs. 0.61 ± 0.11 a.u. $p=0.032$, in the ventral part of the brain; 1.19 ± 0.27 a.u. vs. 8.58 ± 1.10 a.u. $p=0.029$, in dcLNs; $n=5$ in each group). These results clearly confirm our hypothesis that IVH causes significant suppression of lymphatic functions in newborn rats.

Thus, PD4 rat pups demonstrate the clearance of FITCD from the brain into the peripheral lymphatic system suggesting that they have enough effective lymphatic mechanisms of drainage and clearance of the brain that is suppressed after the development of IVH.

Figure 5 - The PS 4 J/cm² stimulation of lymphatic clearance of RBCs and macromolecules from the brain of PD4 pups: (a) and (b) distribution of FITCD (green color) in the ventral and dorsal aspects of the brain without and after PS in the control and IVH groups, respectively. The cerebral blood vessels were filled by 1% Evans Blue (red color); (c) accumulation of FITCD in dCLNs without and after PS in the control and IVH groups;

(d) and (e) the quantitative analysis of the intensity signal from FITCD in the brain (d) and in dcLNs (e); n=5 in each group, the Mann–Whitney–Wilcoxon; (f) Representative fluorescent images of EBD clearance from the right lateral ventricle into dcLNs with and without PS in healthy newborn rats; (g) Quantitative analysis of fluorescence intensity of EBD accumulation in dcLNs in healthy newborn rats with and without PS, n=5 in each group; (h) the OCT data of GNRs accumulation in dcLNs after its intraventricular injection with and without PS in healthy newborn rats, n=10 in each group; (i-k) Representative confocal images of dcLN 1 hour after the injection of saline (i), blood without PS (j) and blood+PS (k) into the right later ventricle; (l-n) 3D rendering of dcLN 1 hour after the injection of saline (l), blood without PS (m) and blood+PS (n) into the right later ventricle (the volume of dcLN was $135 \times 135 \times 40 \mu\text{m}^3$); (o) the number of RBCs in dcLN 1 hour after the blood injection into the right later ventricle with and without PS, n=6 in each group. The LVs were labeled by LYVE-1 (green color) and PROX-1 (yellow color), and RBCs were imaged by their autofluorescence (red color). All values are presented as Mean \pm SEM. Statistical significance in d, e, g and h was assessed by the Wilcoxon test; and statistical significance in o ($F(2,15)=169.597$) was assessed using one-way ANOVA with Turkey's multiple-comparison test (*P < 0.05, **P < 0.01 and ***P < 0.001).

Pages 25, Lines 557-580: Our findings shed light on the therapeutic effects of low-level infra-red PS on brain recovery after IVH and clearly demonstrate that MLVs drain RBCs from the ventricle into dcLNs that can be modulated by PS via activation of lymphatic drainage and clearing mechanisms. The main obstacle to PS application is limited laser penetration into the brain due to the scattering effects of the skull. However, a PS-acceleration of RBCs clearance from the brain via MLVs can be clinically significant for the therapy of brain hemorrhages in premature newborns, who are at highest risk for IVH and in whom PS can be applied through the fontanel. Indeed, we found that the PS course in PD4 rats (a brain maturation level in this age of rats is identical to a preterm human brain of 23 - 28 weeks gestation) improves neurological outcomes, accelerates the RBCs evacuation from the ventricles and provides faster recovery of the BBB permeability and the brain drainage. We believe that a PS-related stimulation of the lymphatic clearance of RBCs and macromolecules from the brain may offer innovative therapeutic approaches to alleviate IVH severity in humans that needs further detailed clinical investigations. We found that PS also had the therapeutic effects in adult mice. We suggest that PS can be an effective treatment for IVH in adults, as has already been shown for treatment of stroke and traumatic brain injury. However, given the loss of laser energy when passing through the skull, it can be assumed that that PS during deep sleep, when the brain's drainage system is activated itself, can significantly increase the therapeutic efficacy of PS, as has been demonstrated for the night therapy of Alzheimer's disease. Further studies of optimal doses and wavelengths of PS, and advantages and limitations of PS-stimulation of the brain's drainage and MLVs in patients of different ages and with various brain hemorrhages could significantly help in the development of guidelines for the safe use of PS in humans. If our preclinical results are confirmed in further clinical trials, non-invasive PS-mediated stimulation of lymphatic clearance of RBCs can be a novel bedside, readily applicable, commercially attractive and viable technology for effective routine treatment of IVH and other types of brain bleedings.

Reviewer #5

Li et al. explore the capacity of transcranial low-level infra-red photostimulation in red blood cells evacuation from the ventricles of newborn and adult mice and rats. Authors explain that low-level infra-red photostimulation reduces intracranial pressure, reduces mortality and improves neurological outcomes. Parallel studies are performed in human tissue and authors also analyze the role of nitric oxide in observed effects.

The study is relevant and provides an interesting venue for patients in need of new therapeutic options. Nevertheless the manuscript also raises some relevant concerns.

Concerns

Comment 1: Abstract should be rewritten to include more detailed and relevant information of the manuscript. General information is given but specific results to support the provided statements are missing. Information regarding explored mechanisms or human studies are also missing (both in the abstract and the introduction).

Response: The authors would like to express their sincere gratitude to the referee for their great help in improving our paper and for the important remark. We rewrote the abstract and the introduction. All changes and additions in the text are highlighted in yellow.

Pages 1-2, Lines 30-47: Intraventricular hemorrhage (IVH) is the most fatal form of brain injury in premature infants and adults. The intracerebral hematoma is toxic to neighboring cells. However, the therapy of IVH is very limited, and new strategies are needed to reduce hematoma expansion. In this pilot study, we have clearly shown that the meningeal lymphatic vessels (MLVs) of rodents and humans play a crucial role in the removal of red blood cells (RBCs) from the brain after IVH. We have discovered that the transcranial low-level infra-red photostimulation (PS) significantly increases the lymphatic excretion of RBCs from the ventricular system of the adult and neonatal mouse brain and have uncovered the clinical significance of the PS therapy of IVH in PD4 rat pups that have the brain similar to a preterm human brain. The PS course in newborn rats significantly improves behavioral outcomes and provides fast reduction of the hematoma size that is associated with full recovery of the ventricular size and the permeability of blood-brain barrier leading to disappearance of vasogenic edema compared with pups not receiving PS and demonstrating long-term recovery after IVH. Our study of the mechanisms of the therapeutic PS effects revealed that PS stimulates a production of nitric oxide in the lymphatic endothelium that improves the brain drainage through alternating phases of relaxation and contraction of lymphatic vessels. These findings shed light on our fundamental knowledge about the PS effects on the brain recovery after IVH and suggest that PS can be a novel bedside, readily applicable, and commercially viable technology for routine treatment of IVH and other types of brain bleedings.

Pages 2-3, Lines 50-97: Intraventricular hemorrhage (IVH) is a bleeding into the brain's ventricular system, where the cerebrospinal fluid (CSF) is produced. The IVH is one of the most common types of brain injury in preterm infants born before the 30th gestational week. The IVH in preterm infants occurs when a germinal matrix hemorrhage ruptures through the ependyma into the lateral ventricle. The incidence of IVH in such infants is approximately 25–30%. Most IVH occurs within the first 72 hours after birth and progresses rapidly within one week. Despite survival rates increasing to approximately 70%, about 45–85% of premature infants with moderate-to-

severe IVH develop significant cognitive deficits, and approximately 75% of such babies need special education in school. In adults, about 30% of IVHs primarily result from trauma, and 70% are secondary, i.e., originate from spontaneous intracranial hemorrhage (ICH). The common associations include hypertension, arteriovenous malformations, aneurysms, moyamoya disease, coagulopathy, and arteriovenous fistula. It is established that IVH extension is independently associated with high mortality and poor functional outcome in adult patients.

Blood in the ventricular system contributes to morbidity in a variety of ways. Pressure from the leaked blood damages brain cells, disabling the proper function of the injured area. Besides, red blood cell (RBC) lysis after IVH results in a release of blood breakdown products (hemoglobin, iron, and bilirubin). Such products have been implicated in post-hemorrhagic hydrocephalus and increased intracerebral pressure (ICP) due to impaired CSF circulation and brain drainage system. Therefore, the conventional therapy for IVH, including surgery and fibrinolysis in combination with extraventricular drainage, is aimed at the blood evacuation from the ventricles, both to reduce ICP induced by mass effects of blood clots on the ventricular walls and the secondary damage caused by blood cell lysis. However, the existing therapy directed at ameliorating intraventricular clot has been limited, and new strategies are needed to reduce hematoma expansion and improve the drainage system of the brain.

Photostimulation (PS) can be an innovative technology targeted for the therapy of IVH. Transcranial PS is a non-pharmacological and non-invasive therapy for stroke and traumatic brain injuries. It is believed that PS increases the metabolic activity of brain tissues and microcirculation, which increases recovery resources. There is strong evidence that PS can regulate the relaxation and permeability of the lymphatic vessels (LVs), activate the movement of immune cells in the lymph, and effectively manage lymphedema. Our preliminary work demonstrated that near-infrared PS (1267 nm) stimulates the clearance of tracers from the brain via modulation of lymphatic tone and contraction. We also showed that PS (1267 nm) effectively stimulates lymphatic clearance of beta-amyloid from the brain.

In this pilot study performed on adult mice and newborn rats, we found that MLVs drain RBCs from the right lateral ventricle into the deep cervical lymph nodes (dcLNs). Using the meninges of humans which died due to IVH, we found RBCs in MLVs. Our animal and human results provide strong support for the lymphatic pathway of RBCs evacuation from the brain. We have in particular uncovered that PS has therapeutic effects on IVH in both adult and newborn rodents, improving neurological outcomes, accelerating the RBCs evacuation from the ventricles, providing faster recovery of permeability of the blood-brain barrier (BBB) and the brain's drainage. These findings shed light on our fundamental knowledge about the effects of low-level infra-red PS on mature and neonatal brain recovery after IVH. The PS-acceleration of RBCs clearance from the brain via MLVs can be clinically significant for the therapy of brain hemorrhages in newborns, where PS can be applied through the fontanelles and where MLVs are localized along the sagittal and transversal sinuses. If our preclinical results will be confirmed in further clinical trials, noninvasive PS-mediated stimulation of lymphatic clearance of RBCs can become a novel bedside, readily applicable, and commercially viable technology for the effective routine treatment of IVH and other types of brain bleeding.

Comments 2: Bibliography should be carefully revised and updated. The manuscript largely relies in bibliography that has over 40 or even 70 years (eg. 1952, 1979, 1983). While there might be seminal studies worth to be cited, they should be accompanied by recent studies confirming these data. The fact that no recent bibliography supports these outcomes should be taken into consideration. Also some sentences should be rewritten to avoid misinterpretations and errors related to the provided bibliographic support. Eg:

“Brain hemorrhage in adults and premature infants has the highest morbidity and mortality worldwide”. In premature infants, IVH is a major complication associated to prematurity but it is not responsible for the highest mortality rates in this population. The referred bibliography in the manuscript is not recent and does not support these asseverations (eg. Beek P, et al. Arch Dis Child Fetal Neonatal Ed 2021; Juul JAMA Network Open.2022;5(2):e2146404. doi:10.1001/jamanetworkopen.2021.46404).

“Intracranial hemorrhages occur in 45% of premature infants and 46-50% of mature babies”. While IVH is the most common hemorrhage in the preterm newborn, the manuscript focuses on IVH and therefore preterm data should refer to IVH specifically. The data provided for mature babies relies on a pilot study from 1999. Again, the bibliography is far from recent and does not support the provided information”.

“It has been shown that RBCs can be cleared from the brain. The manuscript largely depends on this asseveration; nevertheless, this sentence relies on studies from 1952 and 1979. Bibliographical support should be updated to more recent studies showing described information”.

The authors estate “However, the existing therapy of IVH has dramatically impacted the natural history of the disease”. It is not clear what they refer to with “impacting the natural history of the disease”

Response: We highly appreciate this reviewer’s helpful comments. As suggested, we rewrote the introduction and improved bibliography.

Pages 2-3, Lines 50-97: Intraventricular hemorrhage (IVH) is a bleeding into the brain’s ventricular system, where the cerebrospinal fluid (CSF) is produced. The IVH is one of the most common types of brain injury in preterm infants born before the 30th gestational week. The IVH in preterm infants occurs when a germinal matrix hemorrhage ruptures through the ependyma into the lateral ventricle. The incidence of IVH in such infants is approximately 25–30%. Most IVH occurs within the first 72 hours after birth and progresses rapidly within one week. Despite survival rates increasing to approximately 70%, about 45–85% of premature infants with moderate-to-severe IVH develop significant cognitive deficits, and approximately 75% of such babies need special education in school. In adults, about 30% of IVHs primarily result from trauma, and 70% are secondary, i.e., originate from spontaneous intracranial hemorrhage (ICH). The common associations include hypertension, arteriovenous malformations, aneurysms, moyamoya disease, coagulopathy, and arteriovenous fistula. It is established that IVH extension is independently associated with high mortality and poor functional outcome in adult patients.

Blood in the ventricular system contributes to morbidity in a variety of ways. Pressure from the leaked blood damages brain cells, disabling the proper function of the injured area. Besides, red blood cell (RBC) lysis after IVH results in a release of blood breakdown products (hemoglobin,

iron, and bilirubin). Such products have been implicated in post-hemorrhagic hydrocephalus and increased intracerebral pressure (ICP) due to impaired CSF circulation and brain drainage system. Therefore, the conventional therapy for IVH, including surgery and fibrinolysis in combination with extraventricular drainage, is aimed at the blood evacuation from the ventricles, both to reduce ICP induced by mass effects of blood clots on the ventricular walls and the secondary damage caused by blood cell lysis. However, the existing therapy directed at ameliorating intraventricular clot has been limited, and new strategies are needed to reduce hematoma expansion and improve the drainage system of the brain.

Photostimulation (PS) can be an innovative technology targeted for the therapy of IVH. Transcranial PS is a non-pharmacological and non-invasive therapy for stroke and traumatic brain injuries. It is believed that PS increases the metabolic activity of brain tissues and microcirculation, which increases recovery resources. There is strong evidence that PS can regulate the relaxation and permeability of the lymphatic vessels (LVs), activate the movement of immune cells in the lymph, and effectively manage lymphedema. Our preliminary work demonstrated that near-infrared PS (1267 nm) stimulates the clearance of tracers from the brain via modulation of lymphatic tone and contraction. We also showed that PS (1267 nm) effectively stimulates lymphatic clearance of beta-amyloid from the brain.

In this pilot study performed on adult mice and newborn rats, we found that MLVs drain RBCs from the right lateral ventricle into the deep cervical lymph nodes (dcLNs). Using the meninges of humans which died due to IVH, we found RBCs in MLVs. Our animal and human results provide strong support for the lymphatic pathway of RBCs evacuation from the brain. We have in particular uncovered that PS has therapeutic effects on IVH in both adult and newborn rodents, improving neurological outcomes, accelerating the RBCs evacuation from the ventricles, providing faster recovery of permeability of the blood-brain barrier (BBB) and the brain's drainage. These findings shed light on our fundamental knowledge about the effects of low-level infra-red PS on mature and neonatal brain recovery after IVH. The PS-acceleration of RBCs clearance from the brain via MLVs can be clinically significant for the therapy of brain hemorrhages in newborns, where PS can be applied through the fontanelles and where MLVs are localized along the sagittal and transversal sinuses. If our preclinical results will be confirmed in further clinical trials, non-invasive PS-mediated stimulation of lymphatic clearance of RBCs can become a novel bedside, readily applicable, and commercially viable technology for the effective routine treatment of IVH and other types of brain bleeding.

Pages 4, Lines 116-121: These results suggest the lymphatic pathway of RBCs clearance from the adult mouse brain, which is consistent with the oldest and latest animal data on the lymphatic efflux of RBCs from the brain and the meninges. Our human results on the meninges are supported by the data presented by Caversaccio et al., who found an intense accumulation of iron, as an essential element of RBCs, in dcLNs in patients died due to an intracranial hemorrhage that indicates on a lymphatic pathway of clearance of blood products from the brain to the peripheral lymphatics.

Comment 3: Methodology should be revised and detailed information should be provided in order to allow the reproduction of the experiments. Some experiments are not even included in the

methods section.

Response: Thanks very much for your comment. We revised the methods section, and added the missing description of methods. Accordingly, we have added following paragraphs in the revised manuscript.

Page 26, Lines 594-604: The human dura was collected in accordance with the Declaration of Helsinki as a statement of ethical principles for medical research involving human subjects, including research on identifiable human material and data. The species of human meninges at autopsy were obtained from the Department of Pathological Anatomy at the Saratov Medical State University. The present study was performed according to a protocol approved by the Committee of Science and Research Ethics, the Saratov Medical State University. All personal data are stored in strict ethical control, and samples were coded before the analyses of tissue. All obtained samples were fixed and stored in a 10% formalin solution for prolonged periods. The human studies were performed on 3 patients (average age 42) died from IVH (parenchymatous-ventricular hemorrhage in the right cerebral hemisphere with formation of subdural and intracerebral hematomas with blood rupture into ventricles and subarachnoid space).

Page 28, Lines 662-664: At the end of each experiment, animals were immediately sacrificed by cervical dislocation or by decapitation within seconds under deep anesthesia with isoflurane and using the guillotine (Stoelting Co, Wheat Lane, USA) and the DecapiCone® (Braintree Scientific Inc., Braintree, USA).

Page 29, Lines 685-687: The sections of whole meninges from mice as well as approximately 10 slices of dcLN per adult animal and 5 slices for PD4 newborn rats were imaged using a confocal microscope (LSM 710, Zeiss, Jena, Germany) with a ×20 objective (0.8 NA) or a ×60 oil immersion objective (1.46 NA).

Page 29, Lines 692-694: For confocal visualization of MLVs from patients (average age 42) died after IVH (n=3), the meninges (5 cm x 5 cm) from the region of the junction of the anterior sagittal and transverse sinuses were fixed with 4% PFA.

Page 30-31, Lines 722-751:

The analysis of the BBB permeability to FITCD

FITCD (1 mg/25 g mouse, 0.5% solution in saline, Sigma-Aldrich) was injected intravenously via the tail vein and allowed to circulate in the blood during 30 min in mice from the following group: 3 and 11 days after injection of saline/blood as well as after the PS course in sham mice and with IVH. Afterward, mice were decapitated, their brains were quickly removed and fixed in 4% paraformaldehyde (PFA) for 24 h. For confocal imaging of mouse brain slices, we used the protocol for the IHC analysis with the markers for endothelial cell adhesion by Claudin 5 (CLDN5) and for astrocytes by glial fibrillary acidic protein (GFAP). Brain sections were processed according to the standard IHC protocol with the corresponding primary and secondary antibodies. The brain tissues were fixed for 48 hours in a 4% saline solution-buffered formalin, then sections of the brain with a thickness of 40-50 μm were cut on a vibrotome (Leica Microsystems GmbH, Germany). Confocal microscopy of mouse brain sections was performed using a confocal microscope (Nikon A1R MP, Nikon Instruments Inc.). The nonspecific activity was blocked by 2-hour incubation at room temperature with 10% BSA in a solution of 0.2% Triton X-100 in PBS.

Solubilization of cell membranes was carried out during 1-hour incubation at room temperature in a solution of 1% Triton X-100 in PBS. Incubation with primary antibodies in a 1:500 dilution was performed overnight at 4 °C with rabbit antibodies to CLDN 5 (1:500; ab131259; Abcam, Biomedical Campus Cambridge, Cambridge, UK), mouse antibodies to GFAP (1:500; ab279290; Abcam, Biomedical Campus Cambridge, Cambridge, UK). At all stages, the samples were washed 3-4 times with 5-minute incubation in a washing solution. Afterward, the corresponding secondary antibodies were applied (goat anti-rabbit IgG (H+L) Alexa Four 555; goat anti-mouse IgG (H+L) Alexa Four 647; Invitrogen, Molecular Samples, Eugene, Oregon, USA). At the final stage, the sections were transferred to the glass and 15 µl of mounting liquid (50% glycerin in PBS with DAPI at a concentration of 2 µg/ml) was applied to the section. The preparation was covered with a cover glass and confocal microscopy was performed.

Approximately 10 slices per animal from cortical and subcortical (excepting hypothalamus and choroid plexus where BBB is leaky) regions were imaged. The FITCD leakage in arbitrary units was evaluated by measuring changes in the fluorescence intensity of FITCD in the perivascular area in confocal images of the cortex taken 50 and 150 µm depth in 30 min after FITCD intravenous injection using Fiji software (Open-source image processing software).

Pages 38-39, Lines 962-989:

Photo-Damages of MLVs

To ablate the MLVs, visudyne treatment was carried out according to a previous publication. Briefly, mice from the visudyne+laser group were anesthetized with ketamine hydrochloride and visudyne (APExBIO, Cat. No. A8327, 5 µL) was injected into the cisterna magna at a speed of 1 µL/min. Fifteen minutes later, with a 689-nm wave-length laser (Changchun Laser Technology, a dose of 50 J/cm²) was applied through the skull in different places, including the cisterna magna, the left and right transverse sinuses, the superior sagittal sinus, and the junction of all sinuses. For the control groups (sham and visudyne only), mice were injected (5 µL, into the cisterna magna) by physiological saline or visudyne, respectively; the laser (control) group included mice treated by laser (689 nm) irradiation without visudyne. The eyes of mice were protected during photoablation of the MLVs. In all groups, 7 days after photoablation of the MLVs, blood (10 µl) was injected into the right lateral ventricle 1 h before removal of the meninges and confocal microscopy.

Laser speckle contrast imaging

Mice were anesthetized by isoflurane, an incision was done along the midline to separate the skin of the skull, and laser speckle contrast imaging (RWD Life Science Co., Ltd) was used to detect mice cerebral blood flow. Laser speckle blood flow images were recorded and used to identify the regions of interest (ROIs). Within these ROIs, the mean blood flow index was calculated in real time.

Hematoxylin & Eosin (H&E) staining

To analyze the morphological changes in brain tissue, H&E staining was performed. The entire brain of each animal was fixed in 4% neutral paraformaldehyde for 24 h and embedded in paraffin. After that, the samples were cut into 5-µm-thick sections and stained with hematoxylin & eosin. Finally, the sections were magnified and scanned with white light.

L-NAME treatment

To block the release of NO, L-NAME (the blocker of NOS) treatment was carried out according to previous publications. Briefly, mice were anesthetized with ketamine hydrochloride and their heads were fixed in a stereotactic instrument. L-NAME (Sigma, Cat. No. N5751) was reconstituted according to the manufacturer's instructions, and 5 μ l (100 mg/mL) was injected into the cisterna magna at a speed of 1 μ L/min. Four hours later, the mice were used to produce the IVH model.

Comment 4: L-NAME experiments are not described.

Response: We added the description of "L-NAME treatment" in the section of methods in the revised manuscript.

Page 39, Lines 984-989:

L-NAME treatment

To block the release of NO, L-NAME (the blocker of NOS) treatment was carried out according to previous publications^{111, 112}. Briefly, mice were anesthetized with ketamine hydrochloride and their heads were fixed in a stereotactic instrument. L-NAME (Sigma, Cat. No. N5751) was reconstituted according to the manufacturer's instructions, and 5 μ l (100 mg/mL) was injected into the cisterna magna at a speed of 1 μ l/min. Four hours later, the mice were used to produce the IVH model.

Comment 5: Detailed information regarding the patients under study (apart from age) should be provided. Exact brain regions should be also specified. Specific Bioethical information consents and approvals for human studies should be provided.

Response: We added the detailed information regarding the meninges obtained from patients died from IVH and used for our study.

Page 26, Lines 594-604: The human dura was collected in accordance with the Declaration of Helsinki as a statement of ethical principles for medical research involving human subjects, including research on identifiable human material and data. The species of human meninges at autopsy were obtained from the Department of Pathological Anatomy at the Saratov Medical State University. The present study was performed according to a protocol approved by the Committee of Science and Research Ethics, the Saratov Medical State University. All personal data are stored in strict ethical control, and samples were coded before the analyses of tissue. All obtained samples were fixed and stored in a 10% formalin solution for prolonged periods. The human studies were performed on 3 patients (average age 42) died from IVH (parenchymatous-ventricular hemorrhage in the right cerebral hemisphere with formation of subdural and intracerebral hematomas with blood rupture into ventricles and subarachnoid space).

Comment 6: The age of adult mice, and not only the weight, should be provided since adult mice show limited differences in weight, but results in 3 or 8 month-old mice may significantly differ (as authors also state). Also, specifying ages and sex guarantee that the studies can be reproduced

by others.

Response: We highly appreciate the reviewer's helpful comments. As suggested, we have added the information about age of animals used in the experiments in the revised manuscript.

Page 26, Lines 586-587: The experiments were performed on male BALB/c mice (25-28 g, 2-3 months) and PD4 newborn Wistar rats (7-8 g), which during the study reached 14 days of age (1820 g).

Comment 7: 10 days old rats can be hardly considered preterm. The manuscript used as reference induces IVH to PD4 rats. Younger rats should be used to reproduce the IVH of the preterm newborn, as a major point of the manuscript (already stated in the introduction), and given the fact that IVH are common in preterm infants but not in full-term kids.

Response: Thanks very much for this constructive advice. We performed the additional experiments using PD4 rats to demonstrate clinical significance of the PS therapy of IVH. We used PD4 rat pups because their brain maturation level is similar to a preterm human brain of 23 - 28 weeks gestation. In the manuscript, we discuss that the main obstacle to PS application is limited laser penetration into the brain due to the scattering effects of the skull. However, a PS-acceleration of RBCs clearance from the brain via MLVs can be clinically significant for the therapy of brain hemorrhages in premature newborns, who are at highest risk for IVH and in whom PS can be applied through the fontanel. We found that the PS course in PD4 rats improves neurological outcomes, accelerates the RBCs evacuation from the ventricles and provides faster recovery of the BBB permeability and the brain drainage. The new results are presented in the Session "Study of the clinical significance of PS-mediated lymphatic clearance of RBCs from the brain". We have added following paragraphs in the revised manuscript.

Pages 13-14, Lines 294-333:

Study of the clinical significance of PS-mediated lymphatic clearance of RBCs from the brain

The main obstacle to PS application is limited laser penetration into the brain due to the scattering effects of the skull. However, PS-acceleration of the RBCs clearance from the brain via MLVs can be clinically significant for the therapy of intracranial hemorrhages in newborns, where PS can be applied through the fontanelles in the neonatal skull, which is also a window into MLVs localized along the sagittal and transverse sinuses.

Importantly, the lymphatic system can only be one pathway for the RBCs clearance from the brain during the first days of life. There are two pathways for CSF drainage, including the extracerebral lymphatics and the arachnoid villi of the cerebral venous system. However, the arachnoid villi or granulations do not exist prenatally. They only start to become visible in the dura in infants, increasing their density with age, and in adults, they exist in abundance. In recent rodent studies has been discovered that MLVs appear very early in embryos. After postnatal days (PD) 4, MLVs grow along the middle meningeal artery reaching the transversal and the sagittal sinuses to the PD 8 and 28, respectively. At PD3, MLVs are presented in dcLNs

Rats are commonly used as experimental models of neonatal studies due to the similarity of brain maturation in newborn rat pups and human neonates, based on gross anatomical analysis and brain

function studies. Since IVH is common in preterm infants born before the 30th gestational week 1-3, newborn rat pups PD4 were selected for the study of PS effects on the brain lymphatic functions. The newborn rats PD4 have been suggested to reach a brain maturation level that is similar to a preterm human brain of 23 - 28 weeks gestation.

Since there is no information about the anatomical and functional maturation of the lymphatic drainage and clearing in newborn rats, we performed the study of lymphatic removal of FITC-dextran 70 kDa (FITCD) from the right lateral ventricle to dcLNs of PD4 pups using the published protocol. Suppression of the brain lymphatic functions after the development of subdural and subarachnoid hemorrhages as well as the traumatic brain trauma in adult mice has been reported in several publications. Based on these facts, we hypothesized that IVH will be accompanied by a decrease in lymphatic drainage and clearing of the brain tissues in newborn rats. Indeed, Figures 5a-e show that the IVH group vs. the control (sham) group demonstrated suppression of lymphatic evacuation of FITCD from the right lateral ventricle leading to reduced spreading of tracer in the dorsal and ventral aspects of the brain as well as causing a decrease of its accumulation in dcLNs. The quantitative analysis showed that the intensity of fluorescent signal from FITCD in the IVH group vs. the control group was significantly decreased in the dorsal and ventral aspects of the brain as well as in dcLNs (0.29 ± 0.02 a.u. vs. 0.55 ± 0.12 a.u. $p=0.01587$, in the dorsal part of the brain; 0.35 ± 0.05 a.u. vs. 0.61 ± 0.11 a.u. $p=0.03175$, in the ventral part of the brain; 1.19 ± 0.27 a.u. vs. 8.58 ± 1.10 a.u. $p=0.02857$, in dcLNs; $n=5$ in each group). These results confirm our hypothesis that IVH causes significant suppression of lymphatic functions in newborn rats.

Thus, PD4 rat pups demonstrate the clearance of FITCD from the brain into the peripheral lymphatic system suggesting that they have enough effective lymphatic mechanisms of drainage and clearance of the brain that is suppressed after the development of IVH.

Figure 5 - The PS 4 J/cm² stimulation of lymphatic clearance of RBCs and macromolecules from the brain of PD4 pups: (a) and (b) distribution of FITCD (green color) in the ventral and dorsal aspects of the brain without and after PS in the control and IVH groups, respectively. The cerebral blood vessels were filled by 1% Evans Blue (red color); (c) accumulation of FITCD in dCLNs without and after PS in the control and IVH groups;

(d) and (e) the quantitative analysis of the intensity signal from FITCD in the brain (d) and in dcLNs (e); n=5 in each group, the Mann–Whitney–Wilcoxon; (f) Representative fluorescent images of EBD clearance from the right lateral ventricle into dcLNs with and without PS in healthy newborn rats; (g) Quantitative analysis of fluorescence intensity of EBD accumulation in dcLNs in healthy newborn rats with and without PS, n=5 in each group; (h) the OCT data of GNRs accumulation in dcLNs after its intraventricular injection with and without PS in healthy newborn rats, n=10 in each group; (i-k) Representative confocal images of dcLN 1 hour after the injection of saline (i), blood without PS (j) and blood+PS (k) into the right later ventricle; (l-n) 3D rendering of dcLN 1 hour after the injection of saline (l), blood without PS (m) and blood+PS (n) into the right later ventricle (the volume of dcLN was $135 \times 135 \times 40 \mu\text{m}^3$); (o) the number of RBCs in dcLN 1 hour after the blood injection into the right later ventricle with and without PS, n=6 in each group. The LVs were labeled by LYVE-1 (green color) and PROX-1 (yellow color), and RBCs were imaged by their autofluorescence (red color). All values are presented as Mean \pm SEM. Statistical significance in d, e, g and h was assessed by the Wilcoxon test; and statistical significance in o ($F(2,15)=169.597$) was assessed using one-way ANOVA with Turkey’s multiple-comparison test (* $P < 0.05$, ** $P < 0.01$ and *** $P < 0.001$).

Pages 16-17, Lines 334-364:

PS stimulates lymphatic clearance of RBCs and macromolecules from the brain of newborn rats

Since, in previous experiments on adult mice, the PS dose 9 J/cm^2 was found as optimal, we used this PS dose in newborn rats that was 4 J/cm^2 after adaptation of the PS dose to their thin skull transparency (See Session “Laser radiation scheme and dose calculation” in Methods). There were no changes in the temperature on the cortex surface (single PS dose) and no morphological changes (the PS course) in the brain tissues after PS 4 J/cm^2 (Table S5, Fig. S7).

Table S5 - Temperature ($^{\circ}\text{C}$) at the external surface of skull and the brain cortex before and after PS in newborn rats

Number of mouse/ Thermocouple positioning	1	2	3	4	5	6	7	8	9	10	Mean	SEM
no PS												
The skull external surface	35.82	36.00	36.04	36.12	36.08	35.71	36.03	36.01	35.73	36.19	35.973	0.052
Under the skull on the cortex surface	37.18	37.15	37.12	37.21	37.12	37.17	37.10	37.16	37.19	37.14	37.154	0.011
4 J/cm^2 PS												
The skull external surface	36.06	36.11	36.16	36.00	36.17	36.21	36.10	36.18	36.14	36.11	36.124	0.020
Under the skull on the cortex surface	37.07	37.10	37.12	37.18	37.06	37.20	37.16	37.15	37.08	37.07	37.119	0.016

Wilcoxon signed rank test (no PS & 4 J/cm^2 PS)	
Thermocouple positioning	p-value
The skull external surface	0.0414
Under the skull on the cortex surface	0.0850

Figure S7 – The histological analysis (H&E) of the cortex; (a) before and (b) after the PS course 4 J/cm², n=5 in each group.

In *in vivo* experiments on healthy pups, we demonstrated that PS application in a single PS dose of 4 J/cm² significantly stimulated the removal of EBD, GNRs and RBCs from the right lateral ventricle into dcLNs (Fig. 5 f-o). Indeed, the intensity of EBD fluorescence in dcLNs on 60 min of observation was higher in pups after PS compared with pups without PS (2.35 ± 0.24 a.u. vs. 1.66 ± 0.21 a.u., $p=0.037$, $n=5$ in each group) (Fig. 5 f and g). The rate of GNRs accumulation in dcLNs was also higher in pups after PS vs. pups without PS (0.24 ± 0.10 a.u. vs. 0.15 ± 0.17 a.u., $p=0.005$, $n=6$ for the control group (no PS) and $n=7$ for the PS group) (Fig. 5 h).

The results presented in Figures 5 i-o show that the number of RBCs in dcLNs was higher in the IVH+PS group vs. the IVH group without PS ($(0.23 \pm 0.02) \times 10^5$ per mm³ vs. $(0.15 \pm 0.03) \times 10^5$ per mm³, $p=0.000052$, $n=5$ in each group).

In the next step, we answered the question whether PS could improve lymphatic functions in PD4 pups after IVH. Our results clearly demonstrate that the FITCD removal from the brain to dcLNs was significantly improved by PS in newborn rats with IVH and increased in the control group (Fig. 5 a-e). So, in the IVH+PS vs. the IVH no PS groups, the intensity of signal from FITCD was significantly higher in the dorsal and ventral parts of the brain as well as in dcLNs (0.53 ± 0.07 a.u. vs. 0.29 ± 0.02 a.u. $p=0.00793$, in the dorsal part of the brain; 1.04 ± 0.22 a.u. vs. 0.35 ± 0.05 a.u. $p=0.00793$, in the ventral part of the brain; 7.77 ± 1.07 a.u. vs. 1.19 ± 0.27 a.u. $p=0.01587$, in dcLNs; $n=5$ in each group). In the control group+PS vs. the control group no PS, the intensity of signal from FITCD was higher in the dorsal and ventral parts of the brain as well as in dcLNs (1.47 ± 0.35 a.u. vs. 0.55 ± 0.12 a.u. $p=0.03175$, in the dorsal part of the brain; 1.46 ± 0.21 a.u. vs. 0.61 ± 0.11 a.u. $p=0.01587$, in the ventral part of the brain; 19.87 ± 3.23 a.u. vs. 8.58 ± 1.10 a.u. $p=0.02857$, in dcLNs; $n=5$ in each group).

Thus, these series of *in vivo* and *ex vivo* experiments demonstrated that PS effectively improves lymphatic removal of RBCs from the brain of PD4 pups with IVH and increases lymphatic clearance of macromolecules (GNRs, FITCD and EBD) from the brain of healthy newborn rats.

Pages 17-20, Lines 365-414:

Therapeutic effects of the PS-course on recovery of newborn rats and adult mice after IVH

The PS course significantly reduced the intraventricular hemorrhage size in PD4 pups

(Fig. 6 a). Indeed, the size of intraventricular hemorrhage 11 days vs. 3 days after IVH was decreased by 72.5% the IVH+PS (0.08 ± 0.02 mm² vs. 0.29 ± 0.07 mm², $p=0.00037$, $n=7$ in each group) and was decreased only by 34.5% in the IVH without PS (0.19 ± 0.04 mm² vs. 0.29 ± 0.07 mm², $p=0.1608$, $n=7$ in each group). Thus, the size of hematoma was reduced 2.3 times in the IVH+PS group vs. the IVH without PS on the 11th days after IVH (0.08 ± 0.02 mm² vs. 0.19 ± 0.04 mm², $p=0.0014$, $n=7$ in each group).

The PS-course promoted the entire recovery of the ventricles, which were dilated after IVH (Fig. 6 b and Fig. S8). Indeed, 3 days after IVH, the size of the right lateral ventricle was higher in the IVH vs. the sham group (0.98 ± 0.12 mm² vs. 0.21 ± 0.07 mm², $p=0.00001$, $n=7$ in each group). After the PS course, the size of the right lateral ventricle returned to the normal values and did not differ between the IVH+PS and the sham+PS groups (0.23 ± 0.02 mm² and 0.22 ± 0.03 mm², respectively, NS ($p=0.9992$), $n=7$ in each group). However, in the IVH group without PS, the size of the right lateral ventricle remained large vs. the sham group (0.43 ± 0.06 mm² vs. 0.17 ± 0.12 mm², $p=0.00007$, $n=7$ in each group), despite the trend towards recovery compared with pups 3 days after IVH (0.43 ± 0.06 mm² vs. 0.98 ± 0.12 mm², $p=0.00037$, $n=7$ in each group) (Fig. 6 b and Fig. S8). Thus, PS contributed the effective recovery of the ventricular system of brain after IVH.

After the PS course, rat pups demonstrated complete recovery from the vasogenic edema, which was observed in pups 3 days after IVH. Three days after IVH, the size of PVS increased and was higher in the IVH group vs. the sham group (0.0198 ± 0.0045 μ m vs. 0.0015 ± 0.0007 μ m, $p=0.00006$, $n=7$ in each group) (Fig. 6 c,d,g). Eleven days after the IVH, the size of PVS was completely restored after the PS course (0.0014 ± 0.0007 μ m in the IVH+PS group and 0.0012 ± 0.0005 μ m in the sham+PS group, NS ($p=0.9999$), $n=7$ in each group) (Fig. 6 c,f,i). However, the size of PVS remained large on 11th day of observation in the IVH without PS and was higher compared with the sham group (0.0098 ± 0.0021 μ m vs. 0.0017 ± 0.0010 μ m, $p=0.0002$, $n=7$ in each group) (Fig. 6 c,e,h).

Figure 6 – Therapeutic effects of the PS-4 J/cm² course in PD4 newborn rats with IVH: (a) The quantitative analysis of intraventricular hemorrhage size (mm²) formed on 3 days and 11 days after IVH with and without the PS-course, n=7 in each group; (b) The quantitative analysis of the right lateral ventricle size (mm³) on 3 days and 11 days after IVH with and without the PS-course, n=7 in each group; (c) The quantitative analysis of the

PVS size (μm) on 3 days and 11 days after IVH with and without PS-course, $n=7$ in each group; **(d-i)** Histological images illustrating the normal brain tissues without vasogenic edema in the sham group 3 **(d)** and 11 **(e)** days after the injection of physiological saline into the right lateral ventricle as well as in the sham group treated by the PS course **(f)**; the significant increase of the PVS size 3 days after IVH **(g)**; the reducing the PVS size 11 days after IVH without the PS course **(h)** and the completely recovery of PVS in the IVH+PS group **(i)**, $n=7$ in each group; **(j-o)** Representative confocal images of the intact BBB in the sham group 3 **(j)** and 11 **(k)** days after the injection of physiological saline into the right lateral ventricle as well as in the sham group treated by the PS course **(l)**; the FITCD leakage in mice 3 **(m)** and 11 **(n)** days after IVH without PS; completely recovery of BBB in mice with IVH 11 days after IVH with the PS course **(o)**; **(p and q)** – the score of motor tests, including the forelimb test **(p)** and hindlimb test **(q)**; **(r and s)** – the score of neurodevelopment reflex tests, including the cliff avoidance test **(r)** and the gait test **(s)**; $n=7$ in each group. All values are presented as Mean \pm SEM; statistical significance in **a-c** was assessed using the Welch's test; and statistical significance in **p-s** was assessed using the Wilcoxon test (NS represents not significant, * $P < 0.05$, ** $P < 0.01$ and *** $P < 0.001$).

The PS-mediated improvement of brain drainage was associated with better recovery of the blood-brain barrier (BBB) after IVH (Fig. 6 j-o). Indeed, the FITCD leakage was observed 3 and 11 days after IVH vs. the sham groups and was quantified by measuring the fluorescence intensity of FITCD in the perivascular area (Fig.6 j, k and m, n) (1.073 ± 0.016 a.u. vs. 0.069 ± 0.007 a.u. 3 days after IVH, $p=0.000583$ and 0.763 ± 0.090 a.u. vs. 0.057 ± 0.007 a.u. 11 days after IVH, $p=0.000583$; $n=7$ in each group, The Mann-Whitney test). However, there was no the FITCD leakage in the IVH+PS group and the sham+PS group (Fig. 6 l and o) (0.071 ± 0.005 a.u. in the IVH+PS group and 0.080 ± 0.010 a.u. in the sham+PS group, $n=7$ in each group, $p=0.106770$).

Figure S8 – 2D confocal images of fresh brains illustrating size of the right lateral ventricle (three projections) in PD4 newborn rats in the control (sham) group, 3 days and 11 days after IVH without and after PS-course 4 J/cm^2 .

The PS-course 4 J/cm^2 improved the neurological status after IVH in newborn rats. The timeline of motor and neurodevelopment reflex testing is presented in Figure S9. So, the score of the motor tests, such as hind-limb and front-limb suspension, decreased in the IVH group vs. the sham group

suggesting the impairment of motor function after IVH (n=7 in each group, p=0.000168). However, the scores of these tests were essentially increased after PS compared with pups without the PS therapy (n=7 in each group, p=0.000156 for the hind-limb test and p=0.000121 for the front-limb test) (Fig. 6 p and q). The neurodevelopment reflexes were impaired in pups with IVH as evidenced by a decrease in score of the cliff avoidance and the gait tests in the IVH group compared with the sham group (n=7 in each group, p=0.000168 for the cliff avoidance test and p=0.000174 for the gait test). After the PS course, pups performed these tests much better than animals without the PS therapy (n=7 in each group, p=0.000173 for the cliff avoidance test and p=0.000201 for the gait test) (Fig. 6 r and s).

Our results demonstrate that no animals died in the IVH+PS group (10 of 10) and the sham group (10 of 10), while 3 of 10 pups died in the IVH no PS (p=0.092, X2 test Log Rank (Mantel-Cox) = 2,832, Kaplan-Meier method).

Comment 8: Was blood used for IVH treated in anyway? Coagulation seems difficult to avoid but no specs are provided in the methods. On the other hand, figure 6 includes “heparinized blood”.

Response: Sorry for the misleading method description. Blood was not physically mixed with heparin. We used a sterile eppendorf pre-flushed with heparin to avoid coagulation during blood sampling and injection. We have removed the words “heparinized blood” from Fig. S10 and added following paragraph in the revised manuscript.

Page 26, Lines 609-611: The blood was taken from the tail vein of the same mouse and collected in a sterile eppendorf pre-flushed with heparin to avoid coagulation during blood sampling and injection.

Figure S10 – Therapeutic effects of the PS-4 J/cm² course in adult mice with IVH: (a) Kaplan-Meier overall survival plots in the tested groups with and without PS-course; the number of surviving mice by day 21 of observation was significantly higher in the IVH+PS group than in the IVH group (n=30 in each group, p=0.009, X2 test Log Rank (Mantel-Cox) = 9,391); (b) – Representative 2D images of the normal brain, 3 and 11 days after IVH with and without the PS-course, n=10 in each group; (c) The schematic diagram of time points of experiments and the number of animals in the experimental groups; (d and e) – Histological imaging of the normal brain tissues around the right later ventricle and the periventricular hematoma formed 3 days after IVH, n=10 in each group; (f) Representative confocal images of presence of RBC in dCLNs of mice without and after the PS-course, n=6 in each group; (g) The number of RBCs in dCLN 1 hour, a day, 3 days and 7 day after IVH with/without PS, ** p<0.01, *** p<0.001, n=6 in each group, independent-samples T test; (h-k) Histological images illustrating the normal brain tissues without vasogenic edema (h); the significant increase of the PVS size 3 days after IVH (i); the reducing of the PVS size 11 days after IVH (j) that was more pronounced in the IVH+PS group (k) compared with the IVH without PS group (j), n=10 in each group; (l) The quantitative analysis

of periventricular hematoma (mm²) formed on 3 days and 11 days after IVH with and without the PS-course, n=10 in each group, Mean ± SEM, the Welch's test); (**m**) The quantitative analysis of the PVS size (µm) on 3 days and 11 days after IVH with and without the PS-course, n=10 in each group, Mean ± SEM, the Welch's test); (**n**) The quantitative analysis of the ventricular area size (mm²) on 3 days and 11 days after IVH with and without the PS-course, n=10 in each group, Mean ± SEM, the Welch's test).

Comment 9: How was sacrifice performed?

Response: We highly appreciate this reviewer's helpful comments. As suggested, we have added following sentences in the revised manuscript.

Page 28, Lines 662-664: At the end of each experiment, animals were immediately sacrificed by cervical dislocation or by decapitation within seconds under deep anesthesia with isoflurane and using the guillotine (Stoelting Co, Wheat Lane, USA) and the DecapiCone® (Braintree Scientific Inc., Braintree, USA).

Comment 10: The specific regions, number of sections, animals... used in each experiment should be detailed. Similarly, the number, regions and selection of samples for human meninges studies should be specified.

Response: We highly appreciate this reviewer's helpful comments. The number of sections, animals used in each experiment were described in the legend of each figure. We added in the method the detailed information regarding the samples of meninges obtained at autopsy from 3 patients (average age 42) died after IVH as well as about the number of sections of the brains and dcLNs obtained from animals.

Page 26, Lines 594-604: The human dura was collected in accordance with the Declaration of Helsinki as a statement of ethical principles for medical research involving human subjects, including research on identifiable human material and data. The species of human meninges at autopsy were obtained from the Department of Pathological Anatomy at the Saratov Medical State University. The present study was performed according to a protocol approved by the Committee of Science and Research Ethics, the Saratov Medical State University. All personal data are stored in strict ethical control, and samples were coded before the analyses of tissue. All obtained samples were fixed and stored in a 10% formalin solution for prolonged periods. The human studies were performed on 3 patients (average age 42) died from IVH (parenchymatous-ventricular hemorrhage in the right cerebral hemisphere with formation of subdural and intracerebral hematomas with blood rupture into ventricles and subarachnoid space).

Page 29, Lines 693-695: For confocal visualization of MLVs from patients (average age 42) died after IVH (n=3), the meninges (5 cm x 5 cm) from the region of the junction of the anterior sagittal and transverse sinuses were fixed with 4% PFA

Page 29, Lines 685-687: The sections of whole meninges from mice as well as approximately 10 slices of dcLN per adult animal and 5 slices for PD4 newborn rats were imaged using a confocal microscope (LSM 710, Zeiss, Jena, Germany) with a ×20 objective (0.8 NA) or a ×60 oil immersion objective (1.46 NA).

Page 31, Lines 747-751: Approximately 10 slices per animal from cortical and subcortical (excepting hypothalamus and choroid plexus where BBB is leaky) regions were imaged. The FITCD leakage in arbitrary units was evaluated by measuring changes in the fluorescence intensity of FITCD in the perivascular area in confocal images of the cortex taken 50 and 150 μm depth in 30 min after FITCD intravenous injection using Fiji software (Open-source image processing software).

Comment 11: Given the nature of the performed approaches and the feasible damage of the vasculature and BBB, how was this aspect taken into consideration? These experimental procedures might interfere with the actual outcomes. Similarly, FITC-dextran experiments largely rely on an intact BBB.

Response: We added the new results demonstrating the suppression of FITCD evacuation from the right lateral ventricle to the peripheral lymphatic system in PD4 pups with IVH vs. the control (Figure 5 a-e). We also show that PS significantly improves lymphatic clearance of FITCD, EBD and RBCs from the brain in newborn rats. We presented the additional data of the changes in the FITCD 70 kDa leakage in newborn rats with IVH receiving and not the PS course (Figure 6 j-o). The results revealed the FITCD leakage 3 and 11 days after IVH without PS, while the PS course provided fully restoration of the BBB permeability that was accompanied by fast recovery of ventricular size and disappearance of vasogenic edema (Figure 6 a-i). Accordingly, we have added following paragraphs in the revised manuscript.

Pages 13-14, Lines 294-333:

Study of the clinical significance of PS-mediated lymphatic clearance of RBCs from the brain

The main obstacle to PS application is limited laser penetration into the brain due to the scattering effects of the skull. However, PS-acceleration of the RBCs clearance from the brain via MLVs can be clinically significant for the therapy of intracranial hemorrhages in newborns, where PS can be applied through the fontanelles in the neonatal skull, which is also a window into MLVs localized along the sagittal and transverse sinuses.

Importantly, the lymphatic system can only be one pathway for the RBCs clearance from the brain during the first days of life. There are two pathways for CSF drainage, including the extracerebral lymphatics and the arachnoid villi of the cerebral venous system. However, the arachnoid villi or granulations do not exist prenatally. They only start to become visible in the dura in infants, increasing their density with age, and in adults, they exist in abundance. In recent rodent studies has been discovered that MLVs appear very early in embryos. After postnatal days (PD) 4, MLVs grow along the middle meningeal artery reaching the transversal and the sagittal sinuses to the PD 8 and 28, respectively. At PD3, MLVs are presented in dcLNs.

Rats are commonly used as experimental models of neonatal studies due to the similarity of brain maturation in newborn rat pups and human neonates, based on gross anatomical analysis and brain function studies. Since IVH is common in preterm infants born before the 30th gestational week¹⁻³, newborn rat pups PD4 were selected for the study of PS effects on the brain lymphatic functions. The newborn rats PD4 have been suggested to reach a brain maturation level that is similar to a preterm human brain of 23 - 28 weeks gestation.

Since there is no information about the anatomical and functional maturation of the lymphatic drainage and clearing in newborn rats, we performed the study of lymphatic removal of FITC-dextran 70 kDa (FITCD) from the right lateral ventricle to dcLNs of PD4 pups using the published protocol. Suppression of the brain lymphatic functions after the development of subdural and subarachnoid hemorrhages as well as the traumatic brain trauma in adult mice has been reported in several publications. Based on these facts, we hypothesized that IVH will be accompanied by a decrease in lymphatic drainage and clearing of the brain tissues in newborn rats. Indeed, Figures 5a-e show that the IVH group vs. the control (sham) group demonstrated suppression of lymphatic evacuation of FITCD from the right lateral ventricle leading to reduced spreading of tracer in the dorsal and ventral aspects of the brain as well as causing a decrease of its accumulation in dcLNs. The quantitative analysis showed that the intensity of fluorescent signal from FITCD in the IVH group vs. the control group was significantly decreased in the dorsal and ventral aspects of the brain as well as in dcLNs (0.29 ± 0.02 a.u. vs. 0.55 ± 0.12 a.u. $p=0.01587$, in the dorsal part of the brain; 0.35 ± 0.05 a.u. vs. 0.61 ± 0.11 a.u. $p=0.03175$, in the ventral part of the brain; 1.19 ± 0.27 a.u. vs. 8.58 ± 1.10 a.u. $p=0.02857$, in dcLNs; $n=5$ in each group). These results confirm our hypothesis that IVH causes significant suppression of lymphatic functions in newborn rats.

Thus, PD4 rat pups demonstrate the clearance of FITCD from the brain into the peripheral lymphatic system suggesting that they have enough effective lymphatic mechanisms of drainage and clearance of the brain that is suppressed after the development of IVH.

Figure 5 - The PS 4 J/cm² stimulation of lymphatic clearance of RBCs and macromolecules from the brain of PD4 pups: (a) and (b) distribution of FITCD (green color) in the ventral and dorsal aspects of the brain without and after PS in the control and IVH groups, respectively. The cerebral blood vessels were filled by 1% Evans Blue (red color); (c) accumulation of FITCD in dCLNs without and after PS in the control and IVH groups; (d) and (e) the quantitative analysis of the intensity signal from FITCD in the brain (d) and in dCLNs (e); n=5 in

each group, the Mann–Whitney–Wilcoxon; (f) Representative fluorescent images of EBD clearance from the right lateral ventricle into dcLNs with and without PS in healthy newborn rats; (g) Quantitative analysis of fluorescence intensity of EBD accumulation in dcLNs in healthy newborn rats with and without PS, n=5 in each group; (h) the OCT data of GNRs accumulation in dcLNs after its intraventricular injection with and without PS in healthy newborn rats, n=10 in each group; (i-k) Representative confocal images of dcLN 1 hour after the injection of saline (i), blood without PS (j) and blood+PS (k) into the right later ventricle; (l-n) 3D rendering of dcLN 1 hour after the injection of saline (l), blood without PS (m) and blood+PS (n) into the right later ventricle (the volume of dcLN was $135 \times 135 \times 40 \mu\text{m}^3$); (o) the number of RBCs in dcLN 1 hour after the blood injection into the right later ventricle with and without PS, n=6 in each group. The LVs were labeled by LYVE-1 (green color) and PROX-1 (yellow color), and RBCs were imaged by their autofluorescence (red color). All values are presented as Mean \pm SEM. Statistical significance in **d, e, g** and **h** was assessed by the Wilcoxon test; and statistical significance in **o** ($F(2,15)=169.597$) was assessed using one-way ANOVA with Turkey’s multiple-comparison test (* $P < 0.05$, ** $P < 0.01$ and *** $P < 0.001$).

Pages 16-17, Lines 334-364:

PS stimulates lymphatic clearance of RBCs and macromolecules from the brain of newborn rats

Since, in previous experiments on adult mice, the PS dose 9 J/cm^2 was found as optimal, we used this PS dose in newborn rats that was 4 J/cm^2 after adaptation of the PS dose to their thin skull transparency (See Session “Laser radiation scheme and dose calculation” in Methods). There were no changes in the temperature on the cortex surface (single PS dose) and no morphological changes (the PS course) in the brain tissues after PS 4 J/cm^2 (Table S5, Fig. S7).

Table S5 - Temperature ($^{\circ}\text{C}$) at the external surface of skull and the brain cortex before and after PS in newborn rats

Number of mouse/ Thermocouple positioning	1	2	3	4	5	6	7	8	9	10	Mean	SEM
no PS												
The skull external surface	35.82	36.00	36.04	36.12	36.08	35.71	36.03	36.01	35.73	36.19	35.973	0.052
Under the skull on the cortex surface	37.18	37.15	37.12	37.21	37.12	37.17	37.10	37.16	37.19	37.14	37.154	0.011
$4 \text{ J/cm}^2 \text{ PS}$												
The skull external surface	36.06	36.11	36.16	36.00	36.17	36.21	36.10	36.18	36.14	36.11	36.124	0.020
Under the skull on the cortex surface	37.07	37.10	37.12	37.18	37.06	37.20	37.16	37.15	37.08	37.07	37.119	0.016

Wilcoxon signed rank test (no PS & $4 \text{ J/cm}^2 \text{ PS}$)	
Thermocouple positioning	p-value
The skull external surface	0.0414
Under the skull on the cortex surface	0.0850

Figure S7 – The histological analysis (H&E) of the cortex; (a) before and (b) after the PS course 4 J/cm², n=5 in each group.

In *in vivo* experiments on healthy pups, we demonstrated that PS application in a single PS dose of 4 J/cm² significantly stimulated the removal of EBD, GNRs and RBCs from the right lateral ventricle into dcLNs (Fig. 5 f-o). Indeed, the intensity of EBD fluorescence in dcLNs on 60 min of observation was higher in pups after PS compared with pups without PS (2.35 ± 0.24 a.u. vs. 1.66 ± 0.21 a.u., $p=0.037$, $n=5$ in each group) (Fig. 5 f and g). The rate of GNRs accumulation in dcLNs was also higher in pups after PS vs. pups without PS (0.24 ± 0.10 a.u. vs. 0.15 ± 0.17 a.u., $p=0.005$, $n=6$ for the control group (no PS) and $n=7$ for the PS group) (Fig. 5 h).

The results presented in Figures 5 i-o show that the number of RBCs in dcLNs was higher in the IVH+PS group vs. the IVH group without PS ($(0.23 \pm 0.02) \times 10^5$ per mm³ vs. $(0.15 \pm 0.03) \times 10^5$ per mm³, $p=0.000052$, $n=5$ in each group).

In the next step, we answered the question whether PS could improve lymphatic functions in PD4 pups after IVH. Our results clearly demonstrate that the FITCD removal from the brain to dcLNs was significantly improved by PS in newborn rats with IVH and increased in the control group (Fig. 5 a-e). So, in the IVH+PS vs. the IVH no PS groups, the intensity of signal from FITCD was significantly higher in the dorsal and ventral parts of the brain as well as in dcLNs (0.53 ± 0.07 a.u. vs. 0.29 ± 0.02 a.u. $p=0.00793$, in the dorsal part of the brain; 1.04 ± 0.22 a.u. vs. 0.35 ± 0.05 a.u. $p=0.00793$, in the ventral part of the brain; 7.77 ± 1.07 a.u. vs. 1.19 ± 0.27 a.u. $p=0.01587$, in dcLNs; $n=5$ in each group). In the control group+PS vs. the control group no PS, the intensity of signal from FITCD was higher in the dorsal and ventral parts of the brain as well as in dcLNs (1.47 ± 0.35 a.u. vs. 0.55 ± 0.12 a.u. $p=0.03175$, in the dorsal part of the brain; 1.46 ± 0.21 a.u. vs. 0.61 ± 0.11 a.u. $p=0.01587$, in the ventral part of the brain; 19.87 ± 3.23 a.u. vs. 8.58 ± 1.10 a.u. $p=0.02857$, in dcLNs; $n=5$ in each group).

Thus, these series of *in vivo* and *ex vivo* experiments demonstrated that PS effectively improves lymphatic removal of RBCs from the brain of PD4 pups with IVH and increases lymphatic clearance of macromolecules (GNRs, FITCD and EBD) from the brain of healthy newborn rats.

Pages 17-20, Lines 365-414:

Therapeutic effects of the PS-course on recovery of newborn rats and adult mice after IVH

The PS course significantly reduced the intraventricular hemorrhage size in PD4 pups

(Fig. 6 a). Indeed, the size of intraventricular hemorrhage 11 days vs. 3 days after IVH was decreased by 72.5% the IVH+PS ($0.08\pm 0.02 \text{ mm}^2$ vs. $0.29\pm 0.07 \text{ mm}^2$, $p=0.00037$, $n=7$ in each group) and was decreased only by 34.5% in the IVH without PS ($0.19\pm 0.04 \text{ mm}^2$ vs. $0.29\pm 0.07 \text{ mm}^2$, $p=0.1608$, $n=7$ in each group). Thus, the size of hematoma was reduced 2.3 times in the IVH+PS group vs. the IVH without PS on the 11th days after IVH ($0.08\pm 0.02 \text{ mm}^2$ vs. $0.19\pm 0.04 \text{ mm}^2$, $p=0.0014$, $n=7$ in each group).

The PS-course promoted the entire recovery of the ventricles, which were dilated after IVH (Fig. 6 b and Fig. S8). Indeed, 3 days after IVH, the size of the right lateral ventricle was higher in the IVH vs. the sham group ($0.98\pm 0.12 \text{ mm}^2$ vs. $0.21\pm 0.07 \text{ mm}^2$, $p=0.00001$, $n=7$ in each group). After the PS course, the size of the right lateral ventricle returned to the normal values and did not differ between the IVH+PS and the sham+PS groups ($0.23\pm 0.02 \text{ mm}^2$ and $0.22\pm 0.03 \text{ mm}^2$, respectively, NS ($p=0.9992$), $n=7$ in each group). However, in the IVH group without PS, the size of the right lateral ventricle remained large vs. the sham group ($0.43\pm 0.06 \text{ mm}^2$ vs. $0.17\pm 0.12 \text{ mm}^2$, $p=0.00007$, $n=7$ in each group), despite the trend towards recovery compared with pups 3 days after IVH ($0.43\pm 0.06 \text{ mm}^2$ vs. $0.98\pm 0.12 \text{ mm}^2$, $p=0.00037$, $n=7$ in each group) (Fig. 6 b and Fig. S8). Thus, PS contributed the effective recovery of the ventricular system of brain after IVH.

After the PS course, rat pups demonstrated complete recovery from the vasogenic edema, which was observed in pups 3 days after IVH. Three days after IVH, the size of PVS increased and was higher in the IVH group vs. the sham group ($0.0198\pm 0.0045 \mu\text{m}$ vs. $0.0015\pm 0.0007 \mu\text{m}$, $p=0.00006$, $n=7$ in each group) (Fig. 6 c,d,g). Eleven days after the IVH, the size of PVS was completely restored after the PS course ($0.0014\pm 0.0007 \mu\text{m}$ in the IVH+PS group and $0.0012\pm 0.0005 \mu\text{m}$ in the sham+PS group, NS ($p=0.9999$), $n=7$ in each group) (Fig. 6 c,f,i). However, the size of PVS remained large on 11th day of observation in the IVH without PS and was higher compared with the sham group ($0.0098\pm 0.0021 \mu\text{m}$ vs. $0.0017\pm 0.0010 \mu\text{m}$, $p=0.0002$, $n=7$ in each group) (Fig. 6 c,e,h).

Figure 6 – Therapeutic effects of the PS-4 J/cm² course in P4 newborn rats with IVH: (a) The quantitative analysis of intraventricular hemorrhage size (mm²) formed on 3 days and 11 days after IVH with and without the PS-course, n=7 in each group; (b) The quantitative analysis of the right lateral ventricle size (mm³) on 3 days and 11 days after IVH with and without the PS-course, n=7 in each group; (c) The quantitative analysis of the

PVS size (μm) on 3 days and 11 days after IVH with and without PS-course, $n=7$ in each group; **(d-i)** Histological images illustrating the normal brain tissues without vasogenic edema in the sham group 3 **(d)** and 11 **(e)** days after the injection of physiological saline into the right lateral ventricle as well as in the sham group treated by the PS course **(f)**; the significant increase of the PVS size 3 days after IVH **(g)**; the reducing the PVS size 11 days after IVH without the PS course **(h)** and the completely recovery of PVS in the IVH+PS group **(i)**, $n=7$ in each group; **(j-o)** Representative confocal images of the intact BBB in the sham group 3 **(j)** and 11 **(k)** days after the injection of physiological saline into the right lateral ventricle as well as in the sham group treated by the PS course **(l)**; the FITCD leakage in mice 3 **(m)** and 11 **(n)** days after IVH without PS; completely recovery of BBB in mice with IVH 11 days after IVH with the PS course **(o)**; **(p and q)** – the score of motor tests, including the forelimb test **(p)** and hindlimb test **(q)**; **(r and s)** – the score of neurodevelopment reflex tests, including the cliff avoidance test **(r)** and the gait test **(s)**; $n=7$ in each group. All values are presented as Mean \pm SEM; statistical significance in **a-c** was assessed using the Welch's test; and statistical significance in **p-s** was assessed using the Wilcoxon test (NS represents not significant, * $P < 0.05$, ** $P < 0.01$ and *** $P < 0.001$).

The PS-mediated improvement of brain drainage was associated with better recovery of the blood-brain barrier (BBB) after IVH (Fig. 6 j-o). Indeed, the FITCD leakage was observed 3 and 11 days after IVH vs. the sham groups and was quantified by measuring the fluorescence intensity of FITCD in the perivascular area (Fig.6 j, k and m, n) (1.073 ± 0.016 a.u. vs. 0.069 ± 0.007 a.u. 3 days after IVH, $p=0.000583$ and 0.763 ± 0.090 a.u. vs. 0.057 ± 0.007 a.u. 11 days after IVH, $p=0.000583$; $n=7$ in each group, The Mann-Whitney test). However, there was no the FITCD leakage in the IVH+PS group and the sham+PS group (Fig. 6 l and o) (0.071 ± 0.005 a.u. in the IVH+PS group and 0.080 ± 0.010 a.u. in the sham+PS group, $n=7$ in each group, $p=0.106770$).

Figure S8 – 2D confocal images of fresh brains illustrating size of the right lateral ventricle (three projections) in PD4 newborn rats in the control (sham) group, 3 days and 11 days after IVH without and after PS-course 4 J/cm^2 .

The PS-course 4 J/cm^2 improved the neurological status after IVH in newborn rats. The timeline of motor and neurodevelopment reflex testing is presented in Figure S9. So, the score of the motor tests, such as hind-limb and front-limb suspension, decreased in the IVH group vs. the sham group

suggesting the impairment of motor function after IVH (n=7 in each group, p=0.000168). However, the scores of these tests were essentially increased after PS compared with pups without the PS therapy (n=7 in each group, p=0.000156 for the hind-limb test and p=0.000121 for the front-limb test) (Fig. 6 p and q). The neurodevelopment reflexes were impaired in pups with IVH as evidenced by a decrease in score of the cliff avoidance and the gait tests in the IVH group compared with the sham group (n=7 in each group, p=0.000168 for the cliff avoidance test and p=0.000174 for the gait test). After the PS course, pups performed these tests much better than animals without the PS therapy (n=7 in each group, p=0.000173 for the cliff avoidance test and p=0.000201 for the gait test) (Fig. 6 r and s).

Our results demonstrate that no animals died in the IVH+PS group (10 of 10) and the sham group (10 of 10), while 3 of 10 pups died in the IVH no PS (p=0.092, X2 test Log Rank (Mantel-Cox) = 2,832, Kaplan-Meier method).

Comment 12: Statistical approaches are confusing. Non-parametric tests (Mann Whitney and Wilcoxon tests) are performed when 2 populations are under study; however parametric (ANOVA) are used when more populations are under study. Even though Shapiro Wilks test is used to assess normality, this should be clarified. Complete data for ANOVA should be provided (degrees of freedom, f values).

Response: We highly appreciate this reviewer's helpful comments. As suggested, we have modified the statistical analysis approaches in the revised manuscript.

Page 39, Lines 992-1002: Statistical analysis was performed using the SPSS software. Data are presented as mean \pm SEM. The Shapiro-Wilk test, a method for small sample sizes, was used to assess the normality of data distribution in each experiment. The heterogeneity of variance was evaluated using the Levene test, a stable method for both normally and non-normally distributed data. The significance of differences between means was evaluated by unpaired Student's T test (normality distribution, variance homogeneity), Welch's test (normality distribution, variance non-homogeneity) or non-parametric tests (Mann-Whitney-Wilcoxon test, non-normality distribution) for two independent group comparisons. And ANOVA with Turkey's multiple-comparison test (variance homogeneity) or Dunnett's T3 multiple-comparison test (variance non-homogeneity) was used for comparisons of more than two groups. In this study, $P < 0.05$ was considered significant (* $P < 0.05$, ** $P < 0.01$, and *** $P < 0.001$).

Complete data for ANOVA (degrees of freedom, f values) were indicated in the legend of each figure.

By definition the analysis of variance works correctly for the case of a normal distribution [Yu, Zhaoxia, et al. "Beyond t test and ANOVA: applications of mixed-effects models for more rigorous statistical analysis in neuroscience research." *Neuron* 110.1 (2022): 21-35.; Lantz, Björn "The impact of sample non - normality on ANOVA and alternative methods" *British Journal of Mathematical and Statistical Psychology*. 66.2 (2013): 224-244; Zhang, Jin-Ting. "Analysis of variance for functional data. CRC press (2013)". The Kolmogorov-Smirnov and Shapiro-Wilk tests were used to test samples for the normality of statistical distributions and in some cases showed significant deviations from Gaussian statistics (p<0.01). Since the main task was an

independent pairwise comparison of the medians of the samples (for example, IVH and IVH+PS groups), the Mann-Whitney-Wilcoxon test was used, which is well applicable in the case of a non-Gaussian distribution and a small sample sizes ($N \leq 10$) [McKnight, Patrick E., and Julius Najab. "Mann-Whitney U Test. "The Corsini encyclopedia of psychology. (2010): 1-1]. At the same time, the populations were dependent within each pair, but were compared separately from other pairs.

Comment 13: The possibility of intranasal low-level infra-red photostimulation seems a relevant possibility, and authors consider it a relevant issue to the point of being included in the conclusions section. Nevertheless, this idea is only considered for the first time in the last paragraph of the manuscript.

Response: We presented in our recent results intranasal delivery of liposomes to glioblastoma by photostimulation of the lymphatic system” (*Pharmaceutics* 2023, 15(1), 36; <https://doi.org/10.3390/pharmaceutics15010036>). However, in the case of IVH, laser exposure must be applied to the area of the MLVs (along the main venous sinuses) in order to stimulate the removal of blood from the brain tissues. We expect that PS during deep sleep, when the brain's drainage is activated itself, can significantly increase the therapeutic efficacy of PS (Brain Waste Removal System and Sleep: Photobiomodulation as an Innovative Strategy for Night Therapy of Brain Diseases / *Int J Mol Sci.* 2023 Feb 6;24(4):3221. doi: 10.3390/ijms24043221)

Since we didn't perform relevant experiments in this work, we have modified the conclusion and removed the description of intranasal low-level infra-red photostimulation in the revised manuscript.

Page 25, Lines 557-580: Our findings shed light on the therapeutic effects of low-level infra-red PS on brain recovery after IVH and clearly demonstrate that MLVs drain RBCs from the ventricle into dcLNs that can be modulated by PS via activation of lymphatic drainage and clearing mechanisms. The main obstacle to PS application is limited laser penetration into the brain due to the scattering effects of the skull. However, a PS-acceleration of RBCs clearance from the brain via MLVs can be clinically significant for the therapy of brain hemorrhages in premature newborns, who are at highest risk for IVH and in whom PS can be applied through the fontanel. Indeed, we found that the PS course in PD4 rats (a brain maturation level in this age of rats is identical to a preterm human brain of 23 - 28 weeks gestation) improves neurological outcomes, accelerates the RBCs evacuation from the ventricles and provides faster recovery of the BBB permeability and the brain drainage. We believe that a PS-related stimulation of the lymphatic clearance of RBCs and macromolecules from the brain may offer innovative therapeutic approaches to alleviate IVH severity in humans that needs further detailed clinical investigations. We found that PS also had the therapeutic effects in adult mice. We suggest that PS can be an effective treatment for IVH in adults, as has already been shown for treatment of stroke and traumatic brain injury. However, given the loss of laser energy when passing through the skull, it can be assumed that that PS during deep sleep, when the brain's drainage system is activated itself, can significantly increase the therapeutic efficacy of PS, as has been demonstrated for the night therapy of Alzheimer's disease. Further studies of optimal doses and wavelengths of PS, and advantages and limitations of PS-stimulation of the brain's drainage and MLVs in patients of different ages and with various brain

hemorrhages could significantly help in the development of guidelines for the safe use of PS in humans. If our preclinical results are confirmed in further clinical trials, non-invasive PS-mediated stimulation of lymphatic clearance of RBCs can be a novel bedside, readily applicable, commercially attractive and viable technology for effective routine treatment of IVH and other types of brain bleedings.

Minor points

Comment 1: Scattered plots in figure 1 and graphs in figure 6 are too small and difficult to decipher.

Response: We have modified figure 1 and figure 6 (we presented the new data in Fig.6) to make them easy to decipher.

Figure 1 - Lymphatic clearance of RBCs from mouse brain into dLNs. (a) and (b) Representative confocal images of LVs of dCLN stained for LYVE-1 (green color), PROX-1 (yellow color) and RBCs (red color, autofluorescence) without (a) and after IVH (b). 3D rendering images illustrate LVs of dCLN in intact mice on the top in (a) and LVs filled with RBCs in mice with IVH on the bottom in (b). MIP: Maximum intensity projection. PCC: Pearson's correlation coefficient, which is between 1 and -1. 1 represents perfect correlation, -1 represents an entirely negative correlation, and 0 represents a random relationship. Manders' M1: The proportion of the red/yellow fluorescence regions co-located with green fluorescence. Scatter plots indicate that the distribution of PROX-1 is positively correlated with LYVE-1, and almost all PROX-1 coincide with LYVE-1. In addition, the red signals, which represent RBCs are negatively correlated with the distribution of LYVE-1 without IVH, suggesting there were no RBCs inside LVs of dCLNs in intact mice. (c and d) Representative confocal images of MLVs stained for LYVE-1 (green color) and PROX-1 (yellow color), the blood vessels stained with CD31 (red, color) and RBCs (red color, autofluorescence) in intact mice (c) and in mice with IVH (d). 3D rendering images illustrate MLVs in intact mice on the top in (c) and MLVs filled with RBCs (red color, autofluorescence) in mice with IVH on the bottom in (d). In (d) 3D rendering

images are larger view of the frame areas in (1) and (2), which clearly show that RBCs (red color) around MLVs (1) or inside MLVs (2); (e) Representative confocal images of LVs in the human meninges stained for LYVE-1 (green color), the cerebral vessels stained for CD31 (red color), RBCs labeled with GPA (red color), DAPI (blue color).

Figure 6 – Therapeutic effects of the PS-4 J/cm² course in P4 newborn rats with IVH: (a) The quantitative analysis of intraventricular hemorrhage size (mm²) formed on 3 days and 11 days after IVH with and without the PS-course, n=7 in each group; (b) The quantitative analysis of the right lateral ventricle size (mm³) on 3 days and 11 days after IVH with and without the PS-course, n=7 in each group; (c) The quantitative analysis of the PVS size (μm) on 3 days and 11 days after IVH with and without PS-course, n=7 in each group; (d-i) Histological images illustrating the normal brain tissues without vasogenic edema in the sham group 3 (d) and 11 (e) days after the injection of physiological saline into the right lateral ventricle as well as in the sham group treated by the PS course (f); the significant increase of the PVS size 3 days after IVH (g); the reducing the PVS size 11 days after IVH without the PS course (h) and the completely recovery of PVS in the IVH+PS group (i), n=7 in each group; (j-o) Representative confocal images of the intact BBB in the sham group 3 (j) and 11 (k) days after the injection of physiological saline into the right lateral ventricle as well as in the sham group treated by the PS course (l); the FITCD leakage in mice 3 (m) and 11 (n) days after IVH without PS; completely recovery of BBB in mice with IVH 11 days after IVH with the PS course (o); (p and q) – the score of motor tests, including the forelimb test (p) and hindlimb test (q); (r and s) – the score of neurodevelopment reflex tests, including the cliff avoidance test (r) and the gait test (s); n=7 in each group. All values are presented as Mean ± SEM; statistical significance in a-c was assessed using the Welch’s test; and statistical significance in p-s was assessed using the Wilcoxon test (NS represents not significant, *P < 0.05, **P < 0.01 and ***P < 0.001).

Comment 2: Some expressions should be revises: “Rats are commonly employed employed..” “2.51-fold lesser...”

Response: We corrected grammatical and type errors in all text.

Page 13, Lines 309-311: Rats are commonly used as experimental models of neonatal studies due to the similarity of brain maturation in newborn rat pups and human neonates, based on gross anatomical analysis and brain function studies.

Page 17, Lines 368-371: Indeed, the size of intraventricular hemorrhage 11 days vs. 3 days after IVH was decreased by 72.5% the IVH+PS and was decreased only by 34.5% in the IVH without PS...

Pages 20-21, Lines 425-429: Indeed, the size of periventricular hematoma 11 days vs. 3 days after IVH reduced by 60.3% in the IVH+PS group, while this parameter was decreased only by 36.6% in the IVH group without PS...

Page 21, Lines 439-442: In this recovery time compared with 3 days after IVH, the PVS size was decreased to 55.6% in the IVH+PS group and to only 29.6% in the IVH group without PS-course...

Page 21, Lines 446-449: However, during this time, the size of the right later ventricle was decreased to 53.8% in the IVH+PS group and 30.8% in the IVH group without the PS course...

Comment 3: Marks in figure 6 (i, j, k,l) should be more clear. They are very hard to see. Figures and figure legends should be autoexplicative.

Response: We added the new results presented in Figure 6 and modified the marks to make them clear.

Figure 6 – Therapeutic effects of the PS-4 J/cm² course in P4 newborn rats with IVH: (a) The quantitative analysis of intraventricular hemorrhage size (mm²) formed on 3 days and 11 days after IVH with and without the PS-course, n=7 in each group; (b) The quantitative analysis of the right lateral ventricle size (mm³) on 3 days and 11 days after IVH with and without the PS-course, n=7 in each group; (c) The quantitative analysis of the PVS size (µm) on 3 days and 11 days after IVH with and without PS-course, n=7 in each group; (d-i) Histological

images illustrating the normal brain tissues without vasogenic edema in the sham group 3 (d) and 11 (e) days after the injection of physiological saline into the right lateral ventricle as well as in the sham group treated by the PS course (f); the significant increase of the PVS size 3 days after IVH (g); the reducing the PVS size 11 days after IVH without the PS course (h) and the completely recovery of PVS in the IVH+PS group (i), n=7 in each group; (j-o) Representative confocal images of the intact BBB in the sham group 3 (j) and 11 (k) days after the injection of physiological saline into the right lateral ventricle as well as in the sham group treated by the PS course (l); the FITCD leakage in mice 3 (m) and 11 (n) days after IVH without PS; completely recovery of BBB in mice with IVH 11 days after IVH with the PS course (o); (p and q) – the score of motor tests, including the forelimb test (p) and hindlimb test (q); (r and s) – the score of neurodevelopment reflex tests, including the cliff avoidance test (r) and the gait test (s); n=7 in each group. All values are presented as Mean \pm SEM; statistical significance in a-c was assessed using the Welch's test; and statistical significance in p-s was assessed using the Wilcoxon test (NS represents not significant, *P < 0.05, **P < 0.01 and ***P < 0.001).

Comment 4: Fig S2, S5, S8 lack information regarding staining procedure, what exact “histological analysis” is performed?

Response: We added the information about staining procedure in Fig. S2, S7 and S8 as well as we added the methods used for measurement of the cerebral blood flow in S5 and S6. As suggested, we have added following sentences in the revised manuscript.

Pages 38-39, Lines 974-978:

Hematoxylin & Eosin (H&E) staining

To analyze the morphological changes in brain tissue, H&E staining was performed. The entire brain of each animal was fixed in 4% neutral paraformaldehyde for 24 h and embedded in paraffin. After that, the samples were cut into 5- μ m-thick sections and stained with hematoxylin & eosin. Finally, the sections were magnified and scanned with white light.

Pages 39, Lines 979-983:

Laser speckle contrast imaging

Mice were anesthetized by isoflurane, an incision was done along the midline to separate the skin of the skull, and laser speckle contrast imaging (RWD Life Science Co., Ltd) was used to detect mice cerebral blood flow. Laser speckle blood flow images were recorded and used to identify the regions of interest (ROIs). Within these ROIs, the mean blood flow index was calculated in real time.

Figure S2 – The histological analysis (H&E) of the cortex (a) before and after PS course (b) 3 J/cm²; (c) 6 J/cm²; (d) 9 J/cm²; (e) 18 J/cm² and (f) 27 J/cm² (on the skull surface) in adult mice, n=5 in each group.

Figure S5 - The changes in the cerebral blood flow (CBF) detected by laser speckle contrast imaging before and after PS in single dose 9 J/cm² in adult healthy mice (mean±SEM), n=5 in each group, the Wilcoxon test.

Figure S6 – The changes in the CBF detected by laser speckle imaging in the sham group and after

photodynamic ablation of MLVs in the Visudyne+Laser group. There were no changes in CBF after photodynamic effects, n=5, the independent-samples T test.

Figure S7 – The histological analysis (H&E) of the cortex before (a) and after PS single 4 J/cm² dose (b) and the PS course 4 J/cm², n=5 in each group.

Figure S8 – 2D confocal images of fresh brains illustrating size of the right lateral ventricle (three projections) in PD4 newborn rats in the control (sham) group, 3 days and 11 days after IVH without and after PS-course 4 J/cm².

The authors thank the referee for the opportunity to improve our paper with the important suggestions for its possible publication in Nature Communications.

REVIEWERS' COMMENTS

Reviewer #3 (Remarks to the Author):

First I would like to thank the authors for addressing the comments from the previous round of revision.

I have no further scientific question but the manuscript still present some typos and syntax errors (page 13, line 292: ".. CSF drainage via the cranial nerves and the cranial nerves"

Response to Reviewer

We highly appreciate the great assistance with our manuscript (NCOMMS-20-39996D) and the constructive comments of the reviewers. We enclose the revised manuscript, which addresses reviewer' remarks point-by-point.

Reviewer #3

Comments: First I would like to thank the authors for addressing the comments from the previous round of revision.

I have no further scientific question but the manuscript still present some typos and syntax errors (page 13, line 292: "... CSF drainage via the cranial nerves and the cranial nerves"

Response: We are very grateful to this reviewer for the recognition of our work and the time and energy put into the review. We also thank for this helpful comment. As suggested, we have read through the text and fixed all the typos in the revised manuscript.